# A unified mechanism for mitochondrial damage sensing in PINK1-Parkin–mediated mitophagy

Julia A Thayer[1], Jennifer D Petersen[1,7], Xiaoping Huang[1,7], Luiza M Gruel Budet[1,2], James Hawrot[2,3], Daniel M Ramos[4], Shiori Sekine [5], Yan Li[6], Michael E Ward[3] & Derek P Narendra [1✉]

## Abstract

**Damaged mitochondria can be cleared from the cell by mitophagy, using a pathway formed by the recessive Parkinson's disease genes PINK1 and Parkin. Whether the pathway senses diverse forms of mitochondrial damage via a common mechanism, however, remains uncertain. Here, using a novel Parkin reporter in genome-wide screens, we identified that diverse forms of mitochondrial damage converge on loss of mitochondrial membrane potential (MMP) to activate PINK1. Loss of MMP, but not the presequence translocase-associated import motor (PAM), blocked progression of PINK1 import through the translocase of the inner membrane (TIM23), causing it to remain bound to the translocase of the outer membrane (TOM). Ablation of TIM23 was sufficient to arrest PINK1 within TOM, irrespective of MMP. Meanwhile, TOM (including subunit TOMM5) was required for PINK1 retention on the mitochondrial surface. The energy state outside of the mitochondria further modulated the pathway by controlling the rate of new PINK1 synthesis. Together, our findings point to a convergent mechanism of PINK1–Parkin activation by mitochondrial damage: loss of MMP stalls PINK1 import during its transfer from TOM to TIM23.**

**Keywords** Autophagy; Glycolysis; Parkinson's Disease; Unfolded Protein Response
**Subject Categories** Autophagy & Cell Death; Membranes & Trafficking; Organelles

## Introduction

Mitochondria use oxidative phosphorylation (OXPHOS) to generate most of the cell's energy but suffer oxidative damage in the process (Suomalainen and Battersby, 2018). These damaged mitochondria accumulate with age, especially in long-lived neurons and myocytes—unless they can be recognized and degraded through a quality control process, such as the selective form of autophagy called "mitophagy" (Onishi et al, 2021). One major mitophagy pathway eliminating damaged mitochondria is formed by the Parkinson's disease (PD) genes PINK1 and Parkin (Narendra and Youle, 2024). A robust understanding of how the PINK1–Parkin pathway is regulated may spur the development of new therapies for neurodegenerative disorders and other diseases resulting from mitochondrial damage.

In the PINK1–Parkin pathway, damaged mitochondria are first recognized by the ubiquitin kinase PINK1, using an elegant mechanism that relies on differential sorting of PINK1 by healthy and damaged mitochondria (reviewed in (Narendra and Youle, 2024)). Healthy mitochondria rapidly cleave PINK1 from their surface, by importing PINK1 along the TOM-TIM23 mitochondrial precursor path to the inner mitochondrial membrane (IMM). There, PINK1 is cleaved by the protease PARL and released from healthy mitochondria into the cytosol for degradation by the proteasome. Damaged mitochondria, by contrast, cannot fully import and cleave PINK1. Import arrested PINK1, instead, matures on the surface of the damaged mitochondria in a supercomplex with the outer membrane translocase TOM and inner membrane translocase TIM23 (Akabane et al, 2023; Eldeeb et al, 2024; Lazarou et al, 2012). There, PINK1 activates the E3 ubiquitin ligase Parkin in two steps. PINK1, first, phosphorylates ubiquitin on outer mitochondrial membrane (OMM) proteins, which Parkin binds, and PINK1, then, directly phosphorylates Parkin on its ubiquitin-like domain (UBL). Once activated, Parkin ubiquitinates several proteins on the OMM of the damaged mitochondrion. These, in turn, bind ubiquitin-dependent selective autophagy adapters, including Optineurin (Wong and Holzbaur, 2014; Lazarou et al, 2015), to initiate mitophagy. Some of Parkin's most efficient substrates, including the mitochondrial fusion protein Mitofusin-2 (MFN2), can also be degraded by the proteasome in the cytosol following their extraction from the OMM by AAA + -ATPase VCP and its adapters (Tanaka et al, 2010; Kim et al, 2013).

[1]Mitochondrial Biology and Neurodegeneration Unit, Neurogenetics Branch, National Institute of Neurological Disorders and Stroke, National Institutes of Health, Bethesda, MD 20892, USA. [2]Department of Neuroscience, Brown University, Providence, RI 02912, USA. [3]Inherited Neurodegenerative Diseases Section, Neurogenetics Branch, National Institute of Neurological Disorders and Stroke, National Institutes of Health, Bethesda, MD 20892, USA. [4]iPSC Neurodegenerative Disease Initiative, National Institute of Aging, National Institutes of Health, Bethesda, MD 20892, USA. [5]Aging Institute, Department of Cell Biology, School of Medicine, University of Pittsburgh, Pittsburgh, PA 15219, USA. [6]Proteomics Core Facility, National Institute of Neurological Disorders and Stroke, National Institutes of Health, Bethesda, MD 20892, USA. [7]These authors contributed equally: Jennifer D Petersen, Xiaoping Huang. ✉E-mail: derek.narendra@nih.gov

Although many details of the PINK1–Parkin pathway have been worked out, several key questions remain. One of the most pressing is understanding how the PINK1–Parkin response is shaped by individual components of the TOM-TIM23 import precursor pathway and the two driving forces for mitochondrial protein import: the mitochondrial membrane potential (MMP) and (for some but not all precursors) the ATP-dependent presequence translocase-associated motor (PAM) complex (Makki and Rehling, 2023). While it was recognized early that pharmacological disruption of the MMP activates the PINK1–Parkin pathway (Narendra et al, 2008), it remains unclear whether MMP loss is the main trigger of PINK1–Parkin activation in the context of more physiologically relevant sources of mitochondrial damage. For instance, it was recently proposed that protein misfolding in the mitochondrial matrix (e.g., due to loss of the quality control protease LONP1) may activate the PINK1–Parkin pathway by disrupting the PAM complex rather than the MMP (Michaelis et al, 2022). In this model, misfolded protein in the mitochondrial matrix competes away the chaperone component of PAM, HSPA9, presumably disrupting PINK1 import and cleavage. However, it has not been established whether the PAM complex is required for endogenous PINK1 import, as required by the model, and whether activation of PINK1–Parkin by matrix protein misfolding is independent of MMP loss when measured for individual cells and mitochondria. Finally, while recent studies have established that PINK1 is stabilized in a supercomplex with TOM and TIM23 translocases upon activation, it remains unclear which components of the TOM and TIM23 translocases contribute to PINK1 import and stability (Akabane et al, 2023; Eldeeb et al, 2024; Lazarou et al, 2012). Answers to these questions are critical for understanding how the PINK1–Parkin pathway senses mitochondrial damage. Additionally, an understanding of these mechanisms may reveal how the PINK1–Parkin mitophagy pathway can be tuned pharmacologically for therapeutic benefit.

To help address these questions, we developed a novel screening approach for genome-wide activators and facilitators of the PINK1–Parkin pathway. We endogenously tagged one of Parkin's preferred substrates, MFN2, with HaloTag (Los and Wood, 2006), allowing us to monitor its degradation following PINK1–Parkin activation at the single-cell level by flow cytometry. Notably, this strategy is complementary to existing single-cell reporters, such as the widely used mitophagy reporters mt-Keima and mito-QC (McWilliams et al, 2016; Katayama et al, 2011), allowing us to probe for PINK1–Parkin activators and facilitators that may have been missed by prior genome-wide screens.

Using this reporter in fluorescence-activated cell sorting (FACS)-based whole-genome CRISPRi screens (including in cells with endogenous Parkin for the first time), we uncovered several novel facilitators and activators of the PINK1–Parkin pathway. We found diverse forms of mitochondrial damage converge on the loss of the MMP to stabilize PINK1 on the surface of mitochondria. An exception was the disruption of TIM23, which stabilized PINK1 on the mitochondrial surface even in mitochondria with preserved MMP. Together, our results suggest a model in which PINK1–Parkin activation is primarily sensitive to loss of the MMP. The MMP, in this model, provides the main driving force for PINK1 import, as PINK1 is transferred from the TOM translocase

to the TIM23 translocase. In the absence of either the MMP or TIM23, PINK1 is stabilized in the TOM complex at the mitochondrial surface, where it can activate Parkin.

# Results

## MFN2-Halo is a sensitive single-cell reporter of PINK1–Parkin activation

As PINK1–Parkin activation robustly degrades MFN2 upon activation, we reasoned that whole-cell MFN2 levels might be used as a sensitive single-cell reporter of PINK1–Parkin activity (Fig. 1A). To test this idea, we first engineered a series of cell lines in which MFN2 was endogenously tagged with HaloTag (MFN2-Halo). These cell lines included HeLa cells, which lack Parkin, and HEK293 cells, which express endogenous Parkin at low levels (nTPM = 0.5, Human Protein Atlas proteinatlas.org). mCherry-Parkin (mCh-Parkin) was additionally added to HeLa cells to generate a line with high Parkin expression. To enable single and pooled knockdown of genes, these three cell lines (HeLa$^{MFN2-Halo}$, HeLa$^{MFN2-Halo+mCh-Parkin}$, and HEK293$^{MFN2-Halo}$) were additionally engineered for CRISPR interference (CRISPRi) by the stable integration of either dCas9-BFP-KRAB or dCas9-BFP-ZIM3 (Alerasool et al, 2020; Qi et al, 2013; Hsu et al, 2025).

We first evaluated whether MFN2-Halo levels could be detected at the single-cell level by confocal microscopy and flow cytometry. Examining HeLa$^{MFN2-Halo+mCh-Parkin}$ cells by confocal microscopy, we observed MFN2 fluorescence in an OMM pattern consistent with MFN2's known localization (Santel and Fuller, 2001) (Fig. 1B). This signal was reduced by guides directed against endogenous MFN2, confirming its specificity (Fig. 1B). Similarly, by flow cytometry MFN2-Halo expression appeared as a single peak, indicating homogenous expression of MFN2-Halo in single HeLa cells, and decreased approximately fivefold following MFN2 knockdown (KD) (Fig. 1C). Together these findings demonstrate that MFN2 protein levels can be determined in single cells over a wide dynamic range. Similar flow cytometry results were obtained from fixed cells, showing stability of the reporter to fixation (Appendix Fig. S1A).

Next, we tested whether MFN2-Halo could be degraded by Parkin activation in HeLa cells. We first examined the MFN2-Halo in lysates from HeLa cells by immunoblotting. MFN2 appeared as two bands corresponding to one tagged and one untagged copy of endogenous MFN2. The expression level of tagged and untagged MFN2 was similar at baseline, and both were degraded following Parkin activation by OXPHOS inhibitors antimycin and oligomycin (AO) (Fig. 1D). Notably, MFN2 degradation was blocked by PINK1 knockdown (KD), demonstrating that it is mediated by the PINK1–Parkin pathway in HeLa cells (Fig. 1D).

We next evaluated Parkin degradation of MFN2-Halo at a single-cell level by flow cytometry. Activation of Parkin by AO greatly reduced MFN2-Halo levels, as expected. The reduction was largely blocked by PINK1 KD (Fig. 1E), demonstrating its dependence on the PINK1–Parkin pathway. Similarly, we found MFN2-Halo could be degraded by endogenous Parkin in HEK293 cells treated with 10 μM CCCP, albeit at a lower level and with slower kinetics than in HeLa cells with high exogenous Parkin

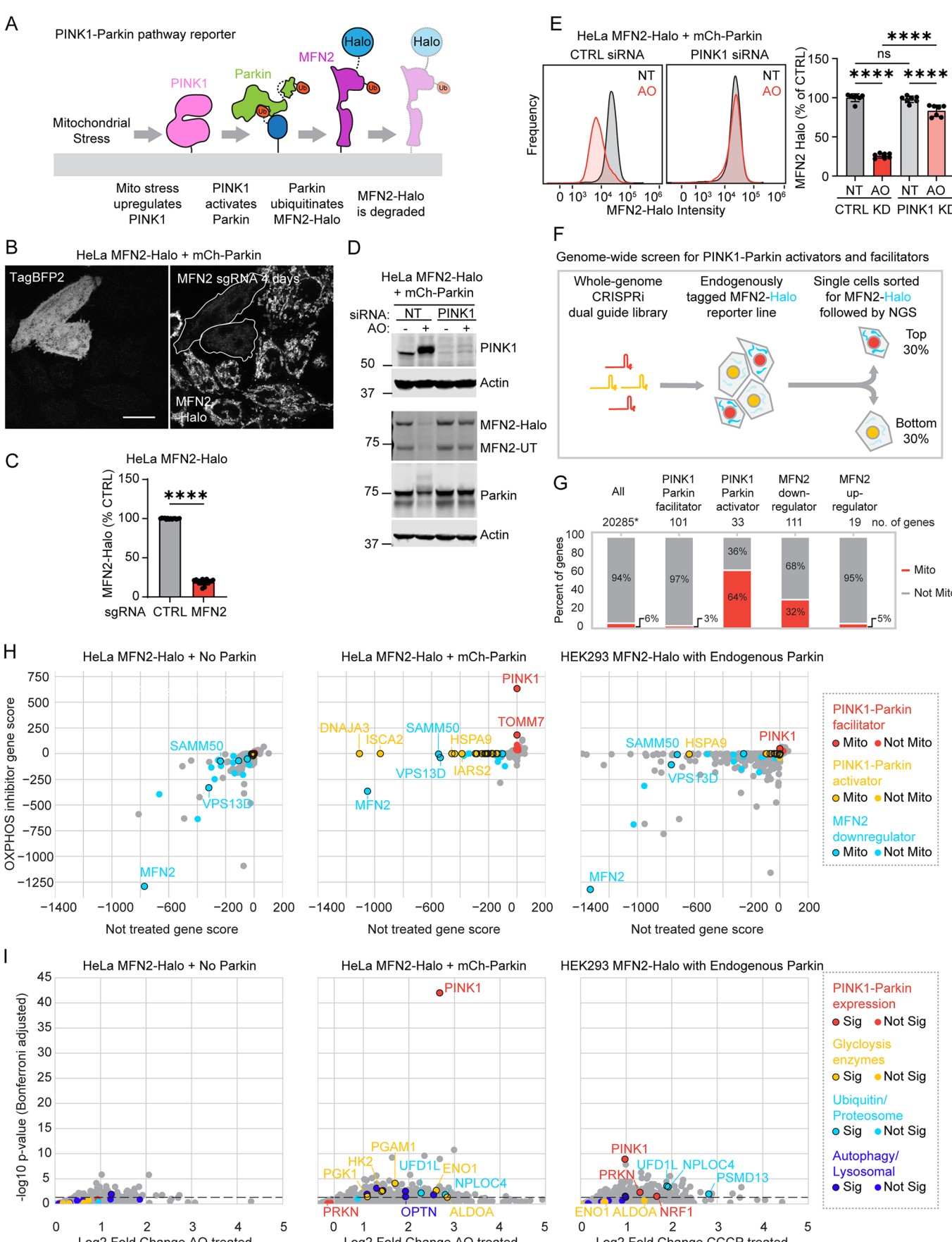

**Figure 1.   Genome-wide screens with the MFN2-Halo reporter identify activators and facilitators of the PINK1–Parkin mitophagy pathway.**

(**A**) Schematic illustrating the PINK1–Parkin pathway reporter MFN2-Halo. (**B**) Representative confocal images of HeLa[MFN2-Halo+mCh-Parkin] cells transduced with a cassette co-expressing a dual sgRNA targeting MFN2 and TagBFP2 to mark transduced cells (white outline). Scale bar = 20 μm. (**C**) Flow cytometry of HeLa[MFN2-Halo] cells demonstrating loss of MFN-Halo signal in the presence of MFN2 sgRNA. Unpaired *t* test two-tailed ****$P \le 0.0001$ (exact *P* value $P = 4.9e\text{-}07$). Error bars mean $+/-$ SD. $N = 3$ independent experiments. (**D**) Representative immunoblots of endogenously tagged and untagged MFN2 alleles (MFN2-Halo and MFN2-UT, respectively) of the same cells in (1B), showing that MFN2-Halo degradation following AO treatment is PINK1 dependent. $N = 5$ replicates on at least two occasions. (**E**) Representative histograms demonstrating the PINK1-dependent decrease in MFN2-Halo fluorescence following treatment with AO in HeLa[MFN2-Halo+mCh-Parkin] cells and quantification. ns = $P = 0.8883$, ****$P \le 0.0001$ (exact *P* values NT CTRL vs AO CTRL, $P = 1.8e\text{-}14$; AO CTRL vs AO PINK1, $P = 1.8e\text{-}14$; NT PINK1 vs AO PINK1, $P = 8.27e\text{-}5$) by two-way ANOVA with Tukey's multiple comparisons test. Error bars mean $+/-$ SD. $N = 3$ independent experiments. (**F**) Schematic illustrating the FACS-based genome-wide CRISPRi screening strategy. Six screens were performed in total, using three different cell lines with or without exposure to OXPHOS inhibitors. (**G**) Stacked bar graph represents the proportion of nuclear gene perturbations encoding mitochondrial (mito) vs. non-mitochondrial proteins (not mito). (**H**) Dot plots of gene scores (product of $\log_2$ fold change and adjusted $-\log_{10}$ *P* value) from screens described in (**F**). (**I**) One-sided volcano plots for screens described in (**F**). Source data are available online for this figure.

expression (Appendix Fig. S1B,C). Together, these results establish that MFN2-Halo is a robust single-cell reporter of PINK1–Parkin activity.

## Genome-wide CRISPRi screens with MFN2-Halo reporter identify facilitators and activators of the PINK1–Parkin pathway

Having established MFN2-Halo as a single-cell reporter of the PINK1–Parkin pathway, we next used the reporter to screen for novel activators and facilitators of the pathway in a pooled FACS-based format. The three MFN2-Halo reporter cell lines were transduced with a whole-genome dual guide CRISPRi library and left untreated or treated with OXPHOS inhibitors (AO or CCCP) (Fig. 1F).

Comparison of HeLa[MFN2-Halo] screens with and without mCh-Parkin distinguished PINK1–Parkin activators and facilitators from guides that alter MFN2-Halo levels independently of the PINK1–Parkin pathway (Fig. 1G–I; Dataset EV1). Activators were those perturbations that caused Parkin-dependent MFN2-Halo degradation in the untreated condition; facilitators were those perturbations that blocked Parkin-dependent MFN2-Halo degradation following OXPHOS inhibition. The top facilitators included genes known to be critical for activation of the PINK1–Parkin pathway, including PINK1; TOMM7, which stabilizes PINK1 on OMM (Hasson et al, 2013); and OPTN and the VCP adapters NPLOC4 and UFD1L, which degrade MFN2 following PINK1–Parkin activation (Wong and Holzbaur, 2014; Lazarou et al, 2015; Tanaka et al, 2010; Kim et al, 2013). Strikingly, most PINK1–Parkin activators encoded mitochondrial proteins, and many have not been previously reported as PINK1–Parkin activators (Fig. 1G). These included, surprisingly, TIMM23, a core subunit of the TIM23 complex, as explored further below. Overall, the gene scores were substantially higher in the HeLa[mCh-Parkin] screens than in the HEK293 screens with endogenous Parkin. This was expected given the more robust degradation of MFN2-Halo in the presence of high Parkin levels. Conversely, the HEK293 screen revealed genes, such as Parkin and NRF1, which are required to maintain endogenous Parkin expression (Lu et al, 2020). These were absent from the HeLa screen, in which Parkin was expressed from an exogenous promoter.

Together, these datasets provide a deep resource of genes modulating the PINK1–Parkin pathway, including (for the first time) the endogenous PINK1–Parkin pathway. They additionally provide a valuable resource of genome-wide regulators of MFN2 expression, independent of the PINK1–Parkin pathway. Mitochondrial pathways downregulating MFN2 expression included mitochondrial fission (DNM1L and MFF), phospholipid transfer and metabolism (VPS13D, TAZ1), PINK1–Parkin independent mitophagy (FBXL4, TMEM11), and cristae remodeling (SAMM50, MINOS, IMMT, ATP5H, ATP5O, ATPC1, ATP5L, ATP5I, ATP5A1, ATP5D) (Dataset EV1).

## Glycolysis facilitates activation of the PINK1–Parkin pathway following OXPHOS impairment

We initially focused our detailed investigations on facilitators of the PINK1–Parkin pathway. Notable among these were genes encoding five core glycolytic enzymes (ENO1, ALDOA, HK2, PGK1, and PGAM1), some of which were also identified in the endogenous screen (but did not reach genome-wide significance) (Fig. 1I). HK2 has previously reported as a PINK1 facilitator (McCoy et al, 2014; Heo et al, 2019), with a proposed mechanism that is independent of its role in glycolysis (Heo et al, 2019). However, identifying four other glycolytic enzymes spread throughout the glycolysis pathway suggested that glycolysis may serve a broader function in PINK1–Parkin activation, with similar dependence on early and late phases of glycolysis. This would be consistent with other prior studies that emphasized the general importance of glycolysis on PINK1 activation following OXPHOS inhibition (McLelland et al, 2018; Lee et al, 2015).

To explore this further, we focused first on ENO1, which had the strongest effect in the screen and catalyzes the penultimate reaction in the late phase of glycolysis (Fig. 2A). We first assessed Parkin activation using the MFN2-Halo reporter by flow cytometry. Consistent with the screen results, ENO1 KD blocked Parkin-dependent degradation of MFN2 to a similar extent as PINK1 KD, following activation with CCCP (Fig. 2B, top). To verify Parkin activation with an orthogonal approach, we measured Parkin mitophagy induction by flow cytometry using the mt-Keima reporter. As expected, ENO1 KD blocked PINK1–Parkin mitophagy activated by CCCP treatment (Fig. 2B, middle). Together, this confirmed that both early and late glycolytic enzymes are required for mitochondrial ubiquitination and mitophagy by the PINK1–Parkin pathway following OXPHOS inhibition.

We next assessed whether ENO1 has its effect up- or downstream of PINK1, by monitoring PINK1-YFP levels using flow cytometry, in HeLa PINK1 KO cells rescued with PINK1-YFP.

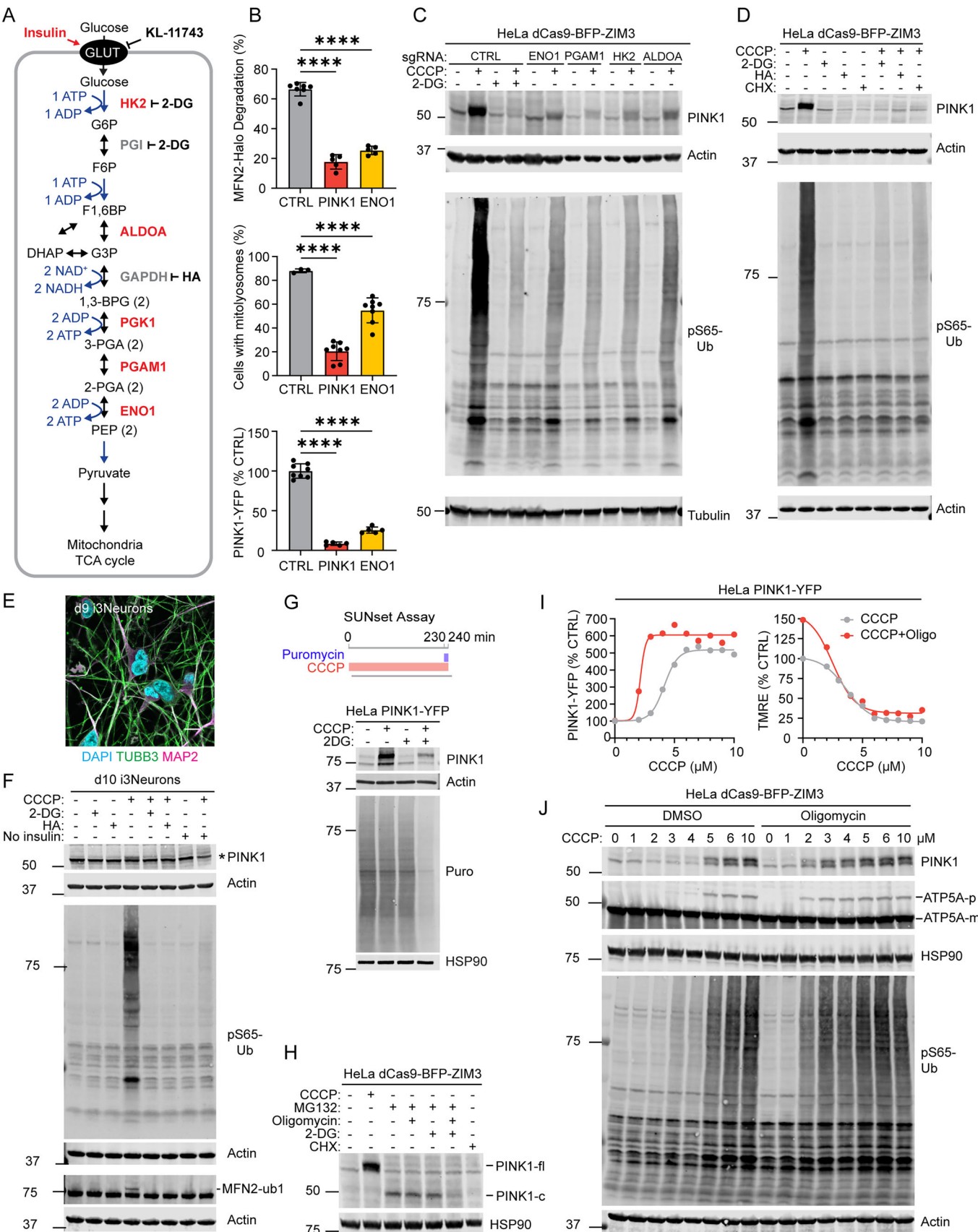

**Figure 2. Glycolysis is required for new PINK1 translation and activation following OXPHOS deficiency.**

(A) Schematic illustrating the core glycolysis pathway with screen hits in red and inhibitors tested in black. (B) Flow cytometry measurements in HeLa[MFN2-Halo+mCh-Parkin] cells (top), HeLa[mt-Keima] (middle), and HeLa[PINK1-YFP] (bottom) treated with CCCP 10 μM for 4 h, illustrating similar results between PINK1 and ENO1 sgRNA. ****$P \leq 0.0001$ by Brown–Forsythe and Welch ANOVA tests with Dunnett's T3 multiple comparisons test (exact $P$ values top graph—CTRL vs ENO1, $P = 6.07e-09$; CTRL vs PINK1, $P = 2.17e-07$; exact $P$ values middle graph—CTRL vs ENO1, $P = 4.55e-05$; CTRL vs PINK1, $P = 2.56e-08$; exact $P$ values bottom graph—CTRL vs ENO1, $P = 3.26e-09$; CTRL vs PINK1, $P = 6.46e-09$). Error bars mean $+/-$ SD. $N = 6$ independent experiments from two separate transductions. (C) Representative immunoblots of PINK1 stabilization and activity in HeLa[dCas9-BFP-ZIM3] cells with indicated glycolytic enzyme sgRNA treated with 10 μM CCCP $+/-$ 10 mM 2-DG for 4 h. $N = 3$ independent experiments. (D) Representative immunoblots of PINK1 stabilization and activity in HeLa[dCas9-BFP-ZIM3] cells treated with 10 μM CCCP, 10 μM HA, 10 mM 2-DG, and/or 50 μg/mL CHX for 4 h. $N = 3$ independent experiments. (E) Representative confocal image of i³Neurons immunostained for neuronal markers. Scale bar $= 10$ μm. (F) Representative immunoblot of lysates from i³Neurons, treated with 10 mM 2-DG, 10 μM HA, and/or 20 μM CCCP, with or without insulin withheld, probing for PINK1 stabilization and activity. *Denotes non-specific band. MFN2-ub1 $=$ monoubiquitinated MFN2. $N \geq 6$ total replicates from at least two independent differentiations. (G) Immunoblot SUNset assay (bottom) performed in HeLa[PINK1-YFP] cells as indicated in the scheme (top), demonstrating block in overall translation and loss of PINK1 accumulation in the presence of 10 μM CCCP and 10 mM 2-DG for 4 h. Puro puromycylated proteins. $N = 3$ independent experiments. (H) Immunoblot of HeLa[dCas9-BFP-ZIM3] cells illustrating cleaved PINK1 (PINK1-c) abundance, stabilized by proteasome inhibition (50 μM MG132), following inhibition of mitochondrial ATP (10 μg/mL oligomycin) and/or glycolytic ATP (10 mM 2-DG) production for 4 h. $N = 3$ replicates on at least two occasions. (I) Flow cytometry measurements of PINK1-YFP (left graph) and mitochondrial membrane potential with TMRE 20 nM (right graph), following drug treatment for 4 h. Graphs are representative of $N = 3$ independent experiments. (J) Immunoblot of HeLa[dCas9-BFP-ZIM3] cells treated with escalating doses of CCCP with or without $F_1F_O$-ATP synthase inhibition (oligomycin 10 μg/mL) for 4 h, probing for PINK1 stabilization and activation. Import block of ATP5A is monitored through the accumulation of uncleaved ATP5A (ATP5A-p) relative to mature ATP5A (ATP5A-m). $N = 3$ independent experiments. Source data are available online for this figure.

ENO1 KD dramatically blocked the increase in PINK1-YFP following CCCP treatment (Fig. 2B, bottom). This demonstrated that ENO1 is required at the first step of the PINK1–Parkin pathway, in which newly translated PINK1 is stabilized on the OMM (Narendra et al, 2010). PINK1-YFP accumulation following OXPHOS inhibition with either CCCP, AO, or rotenone (a complex I inhibitor) + oligomycin (RO) was also blocked by the HK2 inhibitor 2-deoxyglucose (2-DG) (Appendix Fig. S1D).

To assess if glycolysis inhibition also blocks activation of endogenous PINK1, we assayed endogenous PINK1 levels and activation by immunoblotting, focusing on the four strongest glycolysis hits (ENO1, PGAM1, HK2, and ALDOA) from the screen (Fig. 2C). Knockdown of each glycolysis gene blocked accumulation of endogenous PINK1 and PINK1's substrate phospho-S65 ubiquitin (pS65-Ub). Inhibition of HK2 with 2-DG (in the early phase of glycolysis) or GAPDH with heptelidic acid (HA) (in the late phase of glycolysis) similarly blocked accumulation of endogenous full-length PINK1 in HeLa cells (Fig. 2D). This was also phenocopied by introducing blocks upstream of glycolysis through glucose starvation or inhibition of glucose uptake (Appendix Fig. S1E,F). Bypassing glycolytic ATP production with galactose, which is oxidized to pyruvate without net ATP production, or supplementing glucose-free media with high pyruvate did not rescue the PINK1-YFP response to OXPHOS inhibition, similar to a prior report (McLelland et al, 2018; Lee et al, 2015) (Appendix Fig. S1F). This points to glycolytic ATP production as the critical requirement for PINK1 accumulation, as previously suggested (McLelland et al, 2018; Lee et al, 2015). In contrast to glycolysis inhibitors, SYNJ2, recently implicated in PINK1 mRNA transport (Harbauer et al, 2022; Hees et al, 2024), did not block PINK1 synthesis in HeLa cells (Appendix Fig. S1G).

We next tested glycolysis inhibitors in neuron-like cells induced from iPSCs by neurogenin-2 expression (i³Neurons). As in HeLa cells, inhibiting glycolysis in either the early phase (with 2-DG) or late phase (with HA) prevented PINK1 activation (Thoma et al, 2012) (Fig. 2E,F). Likewise, blocking neuronal glucose uptake by insulin withdrawal (Uemura and Greenlee, 2006), phenocopied the effect of glycolysis inhibitors (Fig. 2F, lane 8 vs. 4). This

demonstrates that PINK1 activation requires glucose uptake and glycolysis in i³Neurons.

Following mitochondrial stress, PINK1 is stabilized on the surface of mitochondria in complex with the TOM translocase. It was previously suggested that HK2 selectively blocks the incorporation of PINK1 into TOM translocase without affecting total PINK1 levels (Heo et al, 2019). To test this, we measured PINK1-YFP incorporation into the TOM complex in HeLa cells following HK2 and ENO1 KD, by clear native PAGE (CN-PAGE). Both HK2 and ENO1 KD blocked PINK1-YFP incorporation into the TOM complex in proportion with their effects on total PINK1 accumulation (Fig. 2C compared to Appendix Fig. S1H), arguing against a specific role for HK2 in promoting PINK1-YFP incorporation into the TOM complex. Together, these results demonstrate that glycolysis is required for PINK1 accumulation and activation following OXPHOS inhibition in HeLa cells and human i³Neurons, likely through the glycolytic production of ATP, as proposed previously (McLelland et al, 2018; Lee et al, 2015).

We next considered why glycolytic ATP is required for PINK1 accumulation following OXPHOS inhibition. Activation of the PINK1–Parkin pathway requires new PINK1 to be synthesized for accumulation on impaired mitochondria and can be blocked by the ribosome inhibitor cycloheximide (CHX) (Narendra et al, 2010) (Fig. 2D, lane 8 vs. 2). As protein synthesis is energy demanding (Dolfi et al, 2013), we hypothesized that glycolysis may be needed in the setting of OXPHOS inhibition to provide energy for new PINK1 synthesis, as was also suggested in a prior report (Lee et al, 2015). To test this hypothesis, we monitored global protein translation in the setting of glycolysis and OXPHOS inhibition, using the SUNset assay (Schmidt et al, 2009). Consistent with high energy requirements for translation, 10 mM 2-DG + 10 μM CCCP inhibited global protein translation in parallel with the block in full-length PINK1 accumulation (Fig. 2G, lane 4 vs. 2 and Appendix Fig. S1I). To test this another way, we monitored the accumulation of PARL-cleaved PINK1 (Jin et al, 2010), stabilized by the proteasome inhibitor MG132. Inhibition of both mitochondrial ATP production (using the $F_1F_O$-ATP synthase inhibitor oligomycin) and glycolytic ATP production with 2-DG dramatically decreased production of cleaved PINK1 (Fig. 2H, lane 6 vs. 3).

Together, these findings demonstrate that glycolytic ATP production is required for new PINK1 synthesis in the setting of reduced OXPHOS to activate the PINK1–Parkin pathway.

The mitochondrial ATP pool is partially separate from the cytosolic ATP pool and therefore may have distinct effects on PINK1-YFP stabilization in response to MMP lowering with CCCP. To test these effects, we inhibited mitochondrial ATP production with oligomycin in conjunction with increasing concentrations of CCCP, while simultaneously monitoring MMP and PINK1-YFP intensity by flow cytometry. Notably, oligomycin was not sufficient to stabilize PINK1-YFP on its own but lowered the MMP threshold for PINK1-YFP stabilization (Fig. 2I). As mitochondrial ATP contributes to import along the precursor path through cycling of HSPA9 (also known as, mt-HSP70) in the PAM import motor (Makki and Rehling, 2023), we hypothesized that oligomycin may sensitize mitochondria to MMP treatment by removing the other driving force, the PAM import motor, for precursor import. Consistent with this hypothesis, oligomycin co-treatment lowered the threshold for import block of both endogenous ATP5A and endogenous PINK1 by CCCP (Fig. 2J).

Together, these results demonstrate that the overall energetic state of the cell modulates PINK1–Parkin activation through its effects on PINK1 synthesis: ATP generated by glycolysis becomes limiting for new PINK1 synthesis in the setting OXPHOS dysfunction, attenuating the PINK1–Parkin response. Selectively lowering mitochondrial ATP, by contrast, reduces the MMP threshold for PINK1–Parkin pathway activation but is not sufficient to activate the PINK1–Parkin pathway on its own.

## Diverse forms of mitochondrial damage activate the PINK1–Parkin pathway by disrupting the MMP

We turned next to the PINK1–Parkin activators identified in the screen (Fig. 3A). To identify points of convergence among the functionally diverse activators, we assessed the impact of the top activators on the mitochondrial ultrastructure, the mitochondrial proteome, PINK1–Parkin activity, and the MMP (Fig. 3B).

We first considered their effects on mitochondrial ultrastructure, using transmission electron microscopy (TEM). To aid this analysis, we developed a strategy for identifying guide-transduced cells by TEM: we modified the dual guide vector to co-express ER-targeted APEX2, a genetically encoded reporter for TEM (Martell et al, 2012) (Fig. 3B, bottom). This allowed ready identification of guide transduced cells at the TEM level by the darkly stained osmophilic polymer in the ER lumen produced by the APEX2-DAB reaction (Figs. 3C and EV1A,B). Comparison of mitochondrial ultrastructure among the top activators demonstrated diverse ultrastructural damage to mitochondria. These often reflected the known mitochondrial function of the targeted genes. For instance, matrix protein inclusions were observed following loss of the matrix mt-HSP70 co-chaperone DNAJA3 (Figs. 3C and EV1C; Dataset EV2). In addition, disruption of the boundary IMM was observed following loss of the scaffolding protein PHB2 (Figs. 3C and EV1C; Dataset EV2). A point of convergence among the gene perturbations was disruption of mitochondrial cristae, the enfolded region of the IMM that houses the OXPHOS machinery. Cristae were affected in at least a portion of the mitochondrial network following each perturbation. Notably, protein densities that were frequently observed on the well-formed cristae in control cells,

plausibly formed by the $F_1F_O$-ATP synthase, were rare or absent on the disrupted cristae following knockdown of activator genes (Appendix Fig. S2).

A subset of mitochondria in several of the knockdowns (including NDUFAB1 and IARS2) formed abundant ring or cup shapes (hereafter, cupped mitochondria) (Fig. 3C, yellow closed arrowheads in upper right and lower right images, and Fig. EV1C). Cupped mitochondria have typically been associated with severe loss of membrane potential following treatment with CCCP (Ding et al, 2012; Miyazono et al, 2018). The cupped mitochondria observed following knockdown of NDUFAB1 were often adjacent to intact mitochondria (Fig. 3C, yellow open arrowheads). This heterogeneity may reflect a stochastic mechanism of mitochondrial damage or a compensatory mechanism, such as mitochondrial dynamics, for sorting the limited functional OXPHOS machinery into a subset of mitochondria. Strikingly, PINK1-YFP was found to accumulate selectively on cupped mitochondria (Fig. 3D) by correlated light and electron microscopy (CLEM) following KD of NDUFAB1. Consistent with these representing depolarized mitochondria, analysis of live cells demonstrated that mitochondria with low MMP had higher PINK1-YFP than mitochondria with high MMP in the same cell (Fig. 3E,F). For these experiments, we used the MMP dye MitoLite NIR, which requires a MMP for uptake similar to TMRE but emits in the far-red region (Murata et al, 2020) (Appendix Fig. S3A). Analyzed another way, we found PINK1-YFP correlated with MTS-mCh but poorly with MMP, measured by a Pearson correlation within the whole-cell region (Fig. 3G). Consistently, PINK1-YFP was negatively correlated with MMP in the mitochondrial region defined by MTS-mCh expression (Appendix Fig. S3B–D). Considered together, the mitochondrial ultrastructure was diverse among the activators. One shared ultrastructural feature, however, was disruption of the mitochondrial cristae, the MMP-generating compartment of mitochondria. In those cells with a heterogenous mitochondrial population, PINK1 preferentially targeted the de-energized, cupped mitochondria.

We next tested whether these gene perturbations had common or diverse effects on the mitochondrial proteome. For this purpose, we selected five activators with different mitochondrial functions (IARS2, ISCA2, DNAJA3, PMPCB, and PHB2) for knockdown in HeLa cells (Fig. 3H; Dataset EV3). Knockdown of IARS2, a mitochondrial tRNA synthetase, caused a severe loss of proteins related to mitochondrial translation, including those comprising the mitochondrial ribosome; loss of ISCA2 destabilized complex I subunits and other proteins containing Fe-S clusters; disrupting the matrix co-chaperone DNAJA3 destabilized matrix and IMM proteins *en mass*, while PMPCB and PHB2 knockdowns were minimally disruptive to mitochondrial protein abundance. Notably, all five activated the OMA1-DELE1-ATF4 integrated stress response (mt-ISR), evidenced by increased expression of ATF4 target gene products ASNS, PSAT1, and/or PSPH (Quirós et al, 2017; Guo et al, 2020; Fessler et al, 2020) (Fig. 3H). This is consistent with stress sensed at the IMM by the OMA1-DELE1 pathway (Fessler et al, 2020; Guo et al, 2020), a pathway that is often activated in parallel with the PINK1–Parkin mitophagy pathway. To assess evidence of Parkin activation, we compared the proteomes of HeLa cells with and without Parkin following each KD. We hypothesized that if Parkin was activated there should a global decrease in mitochondrial proteins when comparing cells

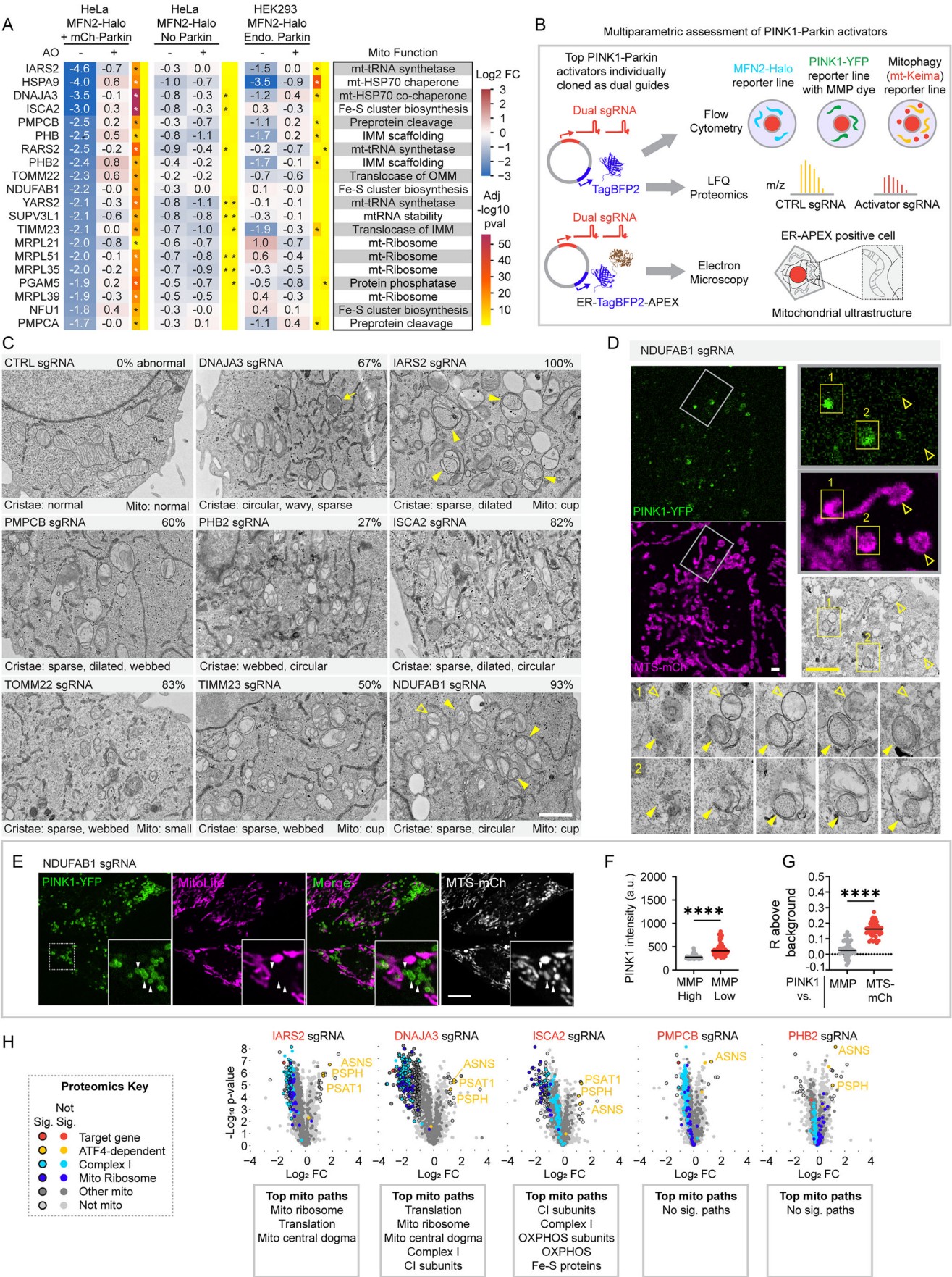

Figure panels A–H as shown.

◀ **Figure 3. PINK1–Parkin activators reflecting diverse mitochondrial functions converge on damage to the inner mitochondrial membrane.**

(A) Heatmaps show $\log_2$ fold changes for the top 20 mitochondrial PINK1–Parkin activators in each of the six screens described in (Fig. 1F). (B) Schematic summarizing the methods used to characterize the top PINK1–Parkin activators. (C) Representative TEM micrographs show abnormalities in mitochondrial ultrastructure induced by knockdown of the top PINK1–Parkin activators in HeLa$^{dCas9-BFP-ZIM3}$ cells transduced with vector co-expressing sgRNA and APEX-ER. Transduced cells were readily identified by dark staining of the osmophilic polymer in the ER lumen produced by the APEX-DAB reaction. In some samples, cupped mitochondria (closed yellow arrowheads), indicative of membrane potential collapse (Ding et al, 2012; Miyazono et al, 2018), were found adjacent to intact mitochondria (open arrowheads). A fluffy aggregate in the matrix of mitochondria (arrow) was observed in DNAJA3 sgRNA cells. The percentage of cells with abnormal mitochondria out of ten or more cells imaged per sample is indicated in the upper right of each image. The text below the images describes the predominant abnormal features of the cristae and cup-shaped or small mitochondrial morphology if observed. Control cells from two transductions were analyzed, while single transductions were performed and analyzed for non-control cells. Scale bar = 1 μm. (D) CLEM images from a NDUFAB1 KD HeLa$^{PINK1-YFP+MTS-mCh}$ cell. PINK1-YFP selectively accumulates on cupped mitochondria (closed arrowheads) while sparing adjacent intact mitochondria (open arrowheads). Serial sections of two boxed areas (1 and 2) are shown at magnification. Scale bars = 2 μm. (E) Representative confocal image of live HeLa$^{PINK1-YFP+MTS-mCh}$ cells transduced with a guide targeting NDUFAB1, showing PINK1-YFP accumulation on a subset of mitochondria that have not accumulated the MMP-dependent dye MitoLite NIR (arrowheads). Scale bar = 10 μm. (F) Graph of cells in (E) comparing PINK1-YFP intensity on mitochondria with and without MMP within the same cell, measured from confocal images as in (E). ****$P \leq 0.0001$ (exact $P$ value, $P = 7e{-}15$) by two-tailed Wilcoxon matched-pairs signed rank test. $N = 48$ cells were analyzed in total from five wells and two independent transductions. (G) Graph of cells in (E) comparing PINK1-YFP co-localization to MTS-mCh and the MMP dye MitoLite NIR. The mean Pearson coefficient for 20 pixel-randomized images (Rrand) above the background for each image. ****$P \leq 0.0001$ (exact $P$ value $P < 1e{-}15$) by two-tailed Mann–Whitney test. $N = 50$ cells were analyzed in total from five wells and two independent transductions. (H) Volcano plots show changes to whole-cell protein abundance following transduction with the indicated guide vs. a non-targeting guide in HeLa cells without Parkin (HeLa$^{dCas9-BFP-ZIM3}$). Boxes show the top MitoCarta3.0 mitochondrial pathways identified in enrichment analysis of significantly down-regulated proteins. Two-sided Student's $t$ tests were performed. Values were corrected for multiple comparisons by calculating an FDR with the Benjamini–Hochberg procedure. Proteins were annotated as significant if they had an FDR < 0.05 and an absolute $\log_2$ fold change of >1 (black outline). $N = 4$ replicates/sgRNA on one occasion. Source data are available online for this figure.

with and without exogenous Parkin expression. Consistently, all five PINK1–Parkin activators caused a global decrease in mitochondrial proteins in the comparison between Parkin-expressing and non-expressing HeLa cells (Appendix Fig. S3E). Thus, the PINK1–Parkin activators, though varied in their perturbations to mitochondrial structure and protein composition, converged on stress to the IMM and activation of damage responses.

To further pinpoint the point of convergence among the PINK1–Parkin activators, we next assessed the impact of 18 of the 20 using a suite of quantitative flow cytometry-based reporters we enabled for CRISPRi (Fig. 3B, top scheme). These included reporters for early events in the PINK1–Parkin pathway that are upstream of Parkin activation: PINK1-YFP and the MMP-sensitive dye MitoLite NIR. They also included reporters of late events that are downstream of Parkin activation: MFN2-Halo and mt-Keima. Using these flow cytometry reporters, we found MMP loss, PINK1 stabilization, and Parkin degradation of MFN2 were highly correlated following each gene perturbation (Fig. 4A–C). We grouped these based on their strength of PINK1–Parkin pathway activation, ranging from targets (group 1) that showed little activation as a population to those that had very strong activation (group 4) as a population across the three measures. Top activators were also found to promote PINK1–Parkin mitophagy, measured with the mt-Keima reporter (Fig. 4D). Comparison of PINK1-YFP and MMP measured in the same experiments showed a non-linear inverse relationship between MMP and PINK1-YFP, with PINK1-YFP markedly increasing after MMP dye intensity was reduced to ~50% of control levels (Fig. 4C). Strikingly, this followed a similar pattern as observed with increasing doses of the mitochondrial uncoupler CCCP (Fig. 2I). To further validate these findings, we used immunoblotting to see if the top activators also stabilized and activated endogenous PINK1 (Fig. EV2A). These showed a similar pattern as observed by flow cytometry. Notably, the strongest activators (groups 3 and 4) are involved in pathways critical for the maturation of newly imported matrix and IMM targeted proteins, including import (TOMM22 and TIMM23), folding of newly

imported proteins (HSPA9 and DNAJA3), Fe-S cluster biosynthesis (ISCA2 and NDUFAB1), and cleavage of the preprotein sequence (PMPCA and PMPCB). This identifies maturation of newly imported proteins as a critical vulnerability leading to MMP loss and PINK1–Parkin activation.

Our findings so far suggest that PINK1–Parkin activation is sensitive to MMP disruption in response to diverse forms of mitochondrial disruption. MMP is also critical for mitochondrial ATP production, and so the pattern of PINK1–Parkin activation could be explained by decreased mitochondrial ATP production. This prompted us to compare our screen results for PINK1–Parkin activators to a previously published genome-wide screen of mitochondrial ATP production that used a similar CRISPRi library (Bennett et al, 2020). Interestingly, our top PINK1–Parkin activators correlated only weakly with mitochondrial perturbations that lowered mitochondrial ATP production (Fig. 4E; Dataset EV4). Similarly, they correlated poorly with gene perturbations that increase mitochondrial superoxide production (e.g., genes encoding CI-, III-, and CoQ10 biosynthesis-related proteins) (Bennett et al, 2024) (Fig. 4E; Dataset EV4). These findings support the notion that loss of MMP is closely associated with PINK1–Parkin activation, in contrast to mitochondrial ATP production and mitochondrial superoxide production.

Next, we analyzed the flow cytometry data at the single-cell level to more precisely correlate MMP loss with PINK1-YFP increase for each gene perturbation. These results were visualized in 2D kernel density plots (Figs. 4F,G and EV2B). Among the 18 activators tested, single cells were generally found in one of two states: high MMP/low PINK1-YFP or low MMP/high PINK1-YFP. That is, single cells generally retained their MMP, in which case PINK1 levels were low, or lost their MMP, in which case PINK1 levels were high. This pattern was also observed for PMPCB and PMPCA (the two subunits of the mitochondrial processing protease (MMP)). This was surprising, as their knockdown was previously reported to stabilize PINK1 in the absence of MMP lowering (Greene et al, 2012). This prompted us to examine the knockdown of PMPCB by live confocal microscopy as an orthogonal approach. Consistent

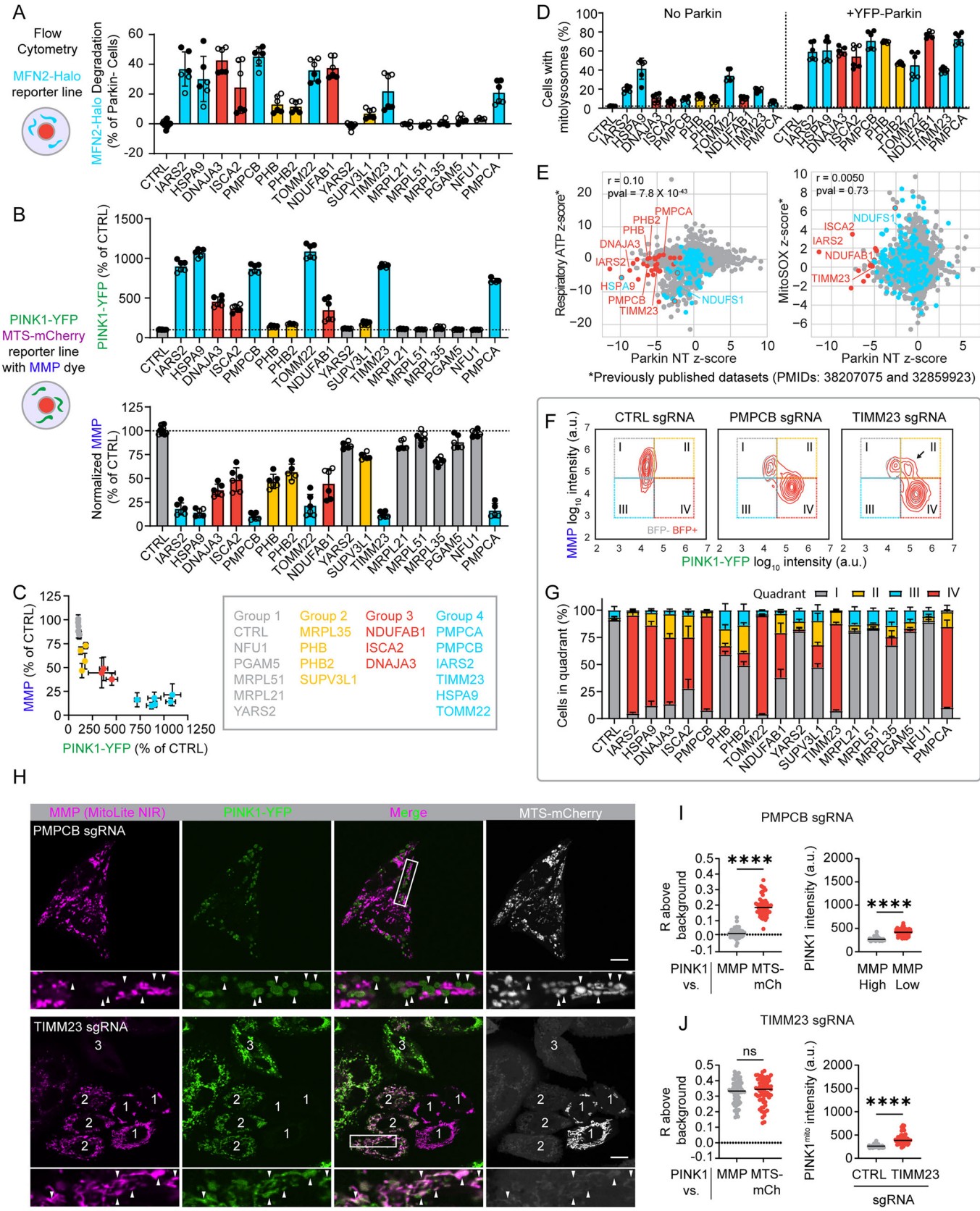

◄

**Figure 4. Diverse mitochondrial stresses disrupt the mitochondrial membrane potential to activate the PINK1–Parkin pathway.**

(A) Flow cytometry measurements of MFN2-Halo degradation (HeLa$^{MFN2-Halo+mCh-Parkin}$ cells). $N = 6$ biological replicates measured from two independent transductions (separate transductions denoted by open or closed circles). Error bars mean $+/−$ SD. (B) Flow cytometry measurements of PINK1-YFP (top) and MMP with dye MitoLite NIR (bottom) obtained from the same cells. (HeLa$^{PINK1-YFP}$ cells) $N = 6$ biological replicates measured from 2 independent transductions (separate transductions denoted by open or closed circles). Error bars mean $+/−$ SD. (C) Graph of experiment in (B) directly comparing PINK1-YFP and MMP for each gene perturbation. (D) Flow cytometry measurements of mitophagy using HeLa$^{mt-Keima}$ cells without Parkin (left) or with exogenous YFP-Parkin expression (right). $N = 6$ biological replicates measured from 2 independent transductions (separate transductions denoted by open or closed circles). Error bars mean $+/−$ SD. (E) Dot plots comparing z-scores from the MFN2-Halo CRISPRi screen in not treated HeLa cells expressing mCh-Parkin (Fig. 1H, middle) with previously published CRISPRi screens of respiratory ATP (left) and mitochondrial superoxide (right) (Bennett et al, 2024, 2020). (F) Representative 2D kernel density plots comparing single-cell PINK1-YFP intensity and intensity of the MMP-sensitive dye MitoLite NIR from the same experiments as shown in (B, C). The arrow indicates a distinct population of cells, high in both MMP and PINK1-YFP, that was observed following TIMM23 KD. (G) Quantification of the proportion of cells in each of the quadrant 2D kernel density plots, such as those seen in (Figs. 3F and EV2B). (H) Representative confocal image of live HeLa$^{PINK1-YFP+MTS-mCh}$ cells transduced with a guide targeting PMPCB (top) or TIMM23 (bottom). Rectangle images are zoomed in the area of the white rectangle outlined in the square images above. PMPCB KD zoomed in area—arrowheads point to cells that have high PINK1-YFP expression, lost MMP, and import is not blocked. TIMM23 KD—Cells labeled 1— population with preserved MMP, no PINK1-YFP expression, and import is not blocked. Cells labeled 2— population with import block, high PINK1-YFP expression, and preserved MMP (examples denoted by white arrowheads in TIMM23 KD zoomed in area). Cells labeled 3— population with high PINK1-YFP expression, and import is blocked. Scale bar = 10 μm. (I) The graph on the left is as described in (Fig. 3G). ****$P ≤ 0.0001$ (exact $P$ value $P < 1e-15$) by two-tailed Mann–Whitney test. $N = 54$ cells were analyzed in total from four wells and two separate transductions. The graph on the right is as described in (Fig. 3F). ****$P ≤ 0.0001$ (exact $P$ value $P < 1e-15$) by two-tailed Wilcoxon matched-pairs signed rank test. $N = 54$ cells were analyzed in total from four wells and two independent transductions. (J) The graph on the left is as described in (Fig. 3G). Exact $P$ value, ns$P = 0.6838$ by two-tailed Mann–Whitney test. $N = 65$ cells were analyzed in total from six wells and two separate transductions. Graph on the right compares PINK1-YFP intensity on mitochondria following CTRL or TIMM23 KD. Only cells with high MMP were analyzed from both groups, and only cells with mitochondrial block (based on MTS-mCh signal) were analyzed for the TIMM23 KD group. ****$P ≤ 0.0001$ (exact $P$ value $P < 1e-15$) by two-tailed Mann–Whitney test. $N = 65$ TIMM23 KD cells and 82 CTRL KD cells were analyzed in total from six wells and two independent transductions. Source data are available online for this figure.

with the flow cytometry findings, PINK1-YFP accumulated preferentially on the population of de-energized mitochondria within single cells following PMPCB knockdown (Fig. 4H, top and I; Appendix Fig. S3B–D), similar to the pattern observed for NDUFAB1 above. Thus, PINK1-YFP accumulation correlated closely with loss of the MMP for the top activators at the single-cell level.

This general pattern had a notable difference, however, for one activator: TIMM23, a core subunit of the TIM23 IMM translocase. Examining the correlation between MMP and PINK1-YFP following TIMM23 KD in single cells by flow cytometry, we observed a unique population of cells, in which PINK1-YFP was elevated, and yet MMP was retained (Fig. 4F, arrow). This population was not observed following the expression of a non-targeting guide or guides against the other activators. To further explore this population, we examined TIMM23 KD by live confocal microscopy in cells that co-expressed PINK1-YFP, the matrix-targeted protein MTS-mCherry, and the MMP dye MitoLite NIR. This combination also allowed us to correlate MMP and PINK1-YFP with chronic import block, as we could assess whether MTS-mCherry was localized to mitochondria or the cytosol. Consistent with the flow cytometry results, we observed three distinct populations of cells, including a population with import block, high PINK1-YFP expression, and preserved MMP (Fig. 4H, bottom, cells numbered "2", and J). In these cells, PINK1-YFP expression was uniformly elevated throughout the mitochondrial network, as evident in the strong Pearson correlation (R) between PINK1-YFP and MMP for this cell population (Fig. 4J, left).

We next tested whether these distinct cell populations following TIMM23 KD had unique ultrastructural features using CLEM. We first imaged a field of living cells by confocal microscopy and then fixed and processed the cells for TEM. In cells with retained MMP and high PINK1-YFP, the mitochondria had dilated cristae but did not exhibit the cupping characteristic of depolarized mitochondria (Fig. EV3, region 1). By contrast, mitochondria with PINK1-YFP accumulation and absent MMP had frequent mitochondrial cupping (Fig. EV3, regions 3–5), similar to NDUFAB1 (Fig. 3D).

Cells with cupped mitochondria also had frequent lipid droplets and lysosomes surrounding the mitochondria; these cellular changes may occur secondary to severe OXPHOS disruption following chronic import block. Together, these results suggest a distinct progression following TIMM23 KD. PINK1-YFP initially accumulates on import-blocked mitochondria that are still able to maintain MMP (Fig. 4H, bottom, cells numbered "2"). As the import block continues, however, OXPHOS fails, and the MMP is reduced (Fig. 4H, bottom, cell numbered "3"). This sequence differs from other activators, exemplified by NDUFAB1 and PMPCB, in which MMP loss precedes and plausibly causes import block.

These results show that blocking PINK1 import to the IMM, through TIMM23 KD, causes PINK1 to accumulate on the surface of mitochondria, where it can activate Parkin to degrade MFN2 and activate mitophagy. PINK1 stabilization on the OMM has also been reported following disruption of PARL, the protease that cleaves PINK1 following its import to the IMM (Jin et al, 2010; Deas et al, 2011). However, PARL was not identified as PINK1–Parkin activator in our screen. To explore this further, we directly compared the effect of TIMM23 KD to PARL KD on endogenous PINK1 stabilization and activation. Consistent with the flow cytometry and confocal results, TIMM23 KD stabilized and activated full-length PINK1. PARL KD, by contrast, stabilized an MTS-cleaved form of PINK1 that was not active against its substrate ubiquitin (Appendix Fig. S3F). Consistent with prior reports (Jin et al, 2010), this suggests that PARL KD stabilizes an inactive form of PINK1, in contrast to TIMM23 KD.

Considered together, these findings demonstrate that import block through the TIM23 complex (either from TIMM23 KD or loss of the driving force for import through the TIM23 translocase) is the primary trigger for PINK1 accumulation on mitochondria.

## PINK1 is stabilized in the TOM complex following loss of TIM23 translocase

PINK1 was recently suggested to form a supercomplex with the TOM and TIM23 translocases following loss of the MMP (Akabane

et al, 2023; Eldeeb et al, 2024). Formation of this supercomplex is proposed to stabilize PINK1 on the OMM. It was, thus, surprising to observe that disruption of TIM23 can cause PINK1 accumulation and activation on the OMM, even in the absence of a key component of this supercomplex. To explore this further, we first verified that PINK1-YFP binds both the TOM and TIM23 translocases following loss of MMP, using affinity purification mass spectrometry (AP-MS). Anti-GFP beads were used to affinity capture PINK1-YFP from HeLa cell extracts after stabilization of PINK1 with CCCP. Consistent with PINK1-YFP forming a supercomplex with the TOM and TIM23 translocases, PINK1-YFP pulled down core components of both the TOM complex (TOMM40 and TOMM22) and the TIM23 translocase (TIMM23 and TIMM17B) (Fig. 5A; Dataset EV5).

Next, we tested which of the core components of the TOM–TIM23 supercomplex is required to stabilize PINK1 on the OMM by comparing knockdown of essential TOM subunits (TOMM22 and TOMM40) to knockdown of TIMM23, an essential TIM23 subunit (Fig. 5B). We first measured PINK1-YFP and MTS-mCh in single cells by flow cytometry. As expected, all knockdowns caused a sustained block in the import of precursors directed to the mitochondrial matrix, as reflected in decreased MTS-mCh expression (Fig. 5B, bottom). Strikingly, all knockdowns also greatly increased whole-cell PINK1-YFP expression (Fig. 5B, top).

To determine where PINK1-YFP accumulates in the cell, we next examined PINK1-YFP by confocal microscopy. Notably, the pattern of PINK1-YFP accumulation differed dramatically following TOM vs. TIM23 perturbation. With TIMM23 KD, PINK1-YFP co-localized with the OMM marker TOMM70, consistent with our findings above (Fig. 5C, bottom). Following TOMM22 or TOMM40 KD, however, little PINK1 accumulated on mitochondria. Instead, PINK1-YFP was found largely on lipid droplets, adjacent to mitochondria (Fig. 5C, top and middle, arrows, and D), and in the cytosol in a mixed diffuse/punctate pattern. Some weak PINK1-YFP signal co-localized with TOMM70 following TOMM22 KD but not TOMM40 KD (Fig. 5C, top, arrowhead), suggesting a more severe phenotype following depletion of TOMM40. To verify that TOMM70 labels mitochondria in the absence of TOMM40, we performed CLEM, using a HeLa^PINK1-YFP cell line in which TOMM70 was additionally endogenously tagged with HaloTag (Appendix Fig. S4). As expected, TOMM70-Halo localized to residual double-membraned mitochondria in TOMM40 KD cells, in proximity to but separate from lipid droplets with PINK1-YFP accumulation.

To more precisely determine where PINK1-YFP accumulates in the absence of TOM, we examined PINK1-YFP localization by super-resolution confocal microscopy and immunogold EM. These methods showed that a large portion of PINK1-YFP accumulates directly on lipid droplets: PINK1-YFP formed a halo around lipid droplets stained with LipidTox by super-resolution microscopy, and immunogold decorated the surface of lipid droplets by TEM (Appendix Fig. S5A,B). To determine whether a particular region of PINK1 is required for lipid droplet localization, we tested a deletion series of PINK1-YFP for lipid droplet localization by confocal microscopy. This showed that lipid droplet localization depended on a helical region (residues 74 - 93) that was previously identified as required for PINK1 OMM localization (Okatsu et al, 2015) (Appendix Fig. S5C). These results suggest that while disruption of either TIM23 or TOM translocases leads to increased cellular PINK1-YFP, the localization is different in each case. PINK1-YFP

accumulates on the mitochondria following disruption of TIM23. By contrast, PINK1-YFP accumulates largely in the cytosol and on lipid droplets following disruption of TOM.

We sought to verify these results using an orthogonal method. Stable knockout (KO) lines for TIMM23, TOMM22, and TOMM40 cannot be generated, as the genes are essential. However, we found it is possible to generate short-lived (7–8 days) KO pools for these three genes. KO of the translocases could be determined by confocal microscopy by examining cells for chronic import block of MTS-mCh and by immunostaining against TIMM23 and TOMM40. Consistent with the findings above, TIMM23 KO caused the accumulation of PINK1-YFP on mitochondria that had lost TIMM23 immunostaining (Fig. 5E–G). KO pools targeting TOMM22 and TOMM40, by contrast, also showed PINK1-YFP accumulation in the cytosol and in rings presumably around lipid droplets (Fig. EV4A). Notably, the disruption of both TIMM23 (by KD) and TOMM40 (by KO) phenocopied the TOMM40 KO (Appendix Fig. S6). Together, these results confirm that disruption of the TIM23 complex (by either KD or KO) stabilizes PINK1-YFP on mitochondria, while TOMM40 disruption accumulates PINK1-YFP disassociated from mitochondria.

Our findings so far have focused on the examination of fluorescence from stable, exogenously expressed PINK1-YFP. To determine whether full-length PINK1-YFP accumulates and whether it is active, we next examined PINK1-YFP by immunoblotting following each KD. Consistent with the flow cytometry and confocal results, full-length PINK1-YFP accumulated following disruption of either TOM or TIM23 translocases. Importantly, however, PINK1 was only activated by TIMM23 KD, as evidenced by the accumulation of its product pS65-Ub (Fig. 5H, left blot). This suggests that the full-length PINK1-YFP, which accumulates in the absence of TOM, is largely inactive.

We next examined the effect of TIM23 vs. TOM disruption on endogenous PINK1. Immunoblotting for endogenous PINK1 showed a similar pattern as exogenous PINK-YFP, except that endogenous full-length PINK1 did not accumulate following TOM disruption (Fig. 5H, right blot). We hypothesize that the YFP tag may block degradation of full-length PINK1, allowing PINK1-YFP but not endogenous PINK1 to accumulate following loss of TOM. As above, PINK1 activity was increased by TIM23 but not by TOM disruption. Together, these results demonstrate that the loss of TIM23 but not TOM causes PINK1 activation.

Our results so far demonstrate that TIM23 disruption causes PINK1 stabilization and activation on the mitochondrial surface. To test whether PINK1 is stabilized in the TOM complex in this condition, we next evaluated PINK1-YFP's interaction with the TOM complex by both AP-MS and CN-PAGE. In the absence of the TIM23 translocase, immunoprecipitated PINK1-YFP maintained its association with the TOM subunits but lost its association with TIM23 subunits (Fig. 5I; Dataset EV5). Consistently, by CN-PAGE, PINK1-YFP accumulated in discrete complexes that corresponded to those observed following PINK1-YFP accumulation by CCCP. These complexes could be shifted to a higher molecular weight by antibodies against TOMM20, TOMM22, or TOMM40 (Figs. 5J and EV4B). This demonstrates that the PINK1-YFP complexes observed by CN-PAGE contain the TOM translocase. Together, this establishes that following chronic ablation of TIM23, PINK1 is stabilized in TOM in an active conformation, similar to its stabilization by loss of the MMP.

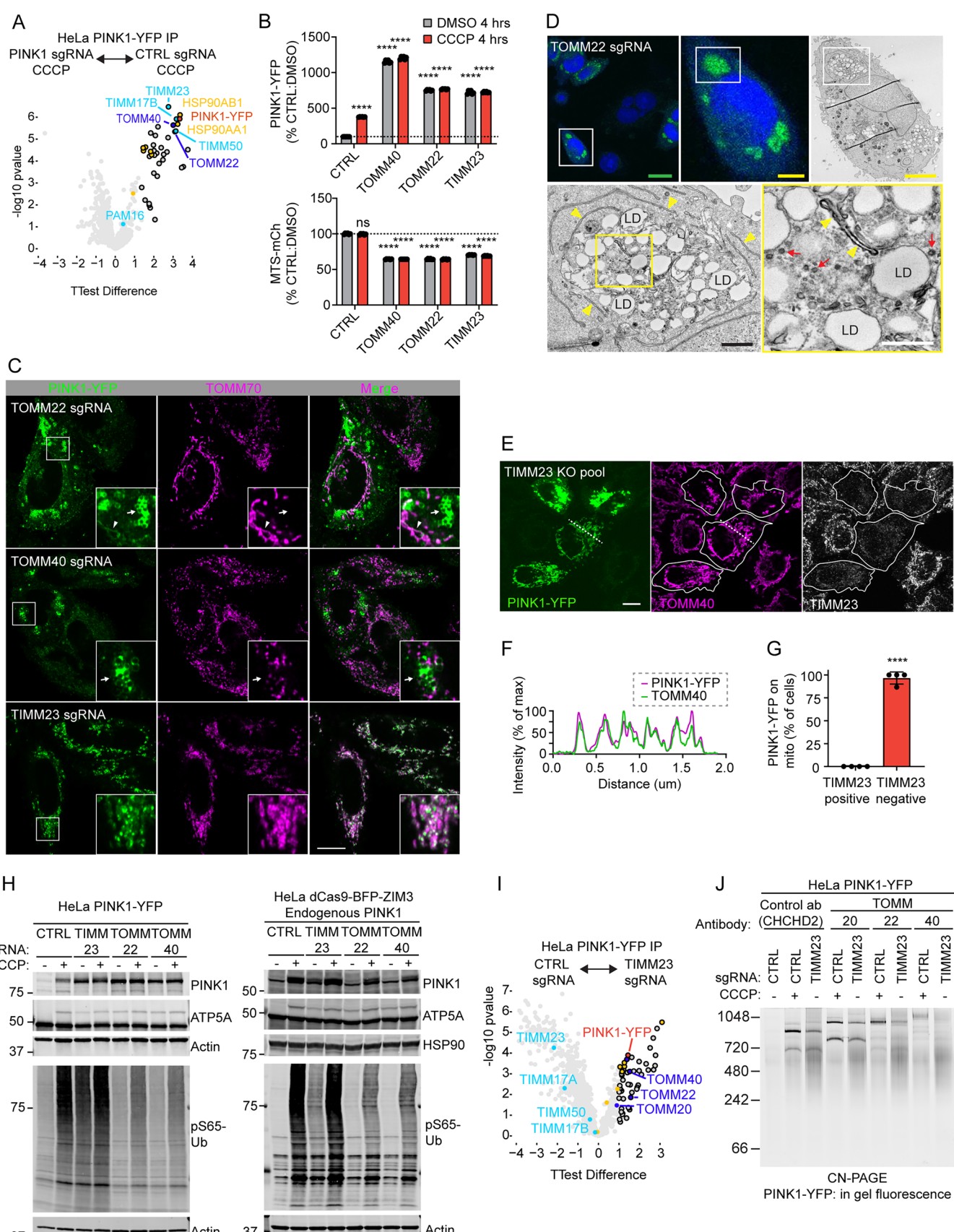

**Figure 5. PINK1 stabilization on the OMM requires the TOM but not the TIM23 translocase.**

(A) Volcano plot shows interactors of PINK1-YFP (red) identified by AP-MS following transduction with CTRL or PINK1 guides in HeLa[PINK1-YFP] cells. Cells in both groups were treated with 10 μM CCCP overnight. Two-sided Student's *t* tests were performed. Values were corrected for multiple comparisons by calculating a FDR with the permutation method. Proteins were annotated as significant interactors if they had an absolute log2 fold change of >1 (black outline). Proteins associated with TOM (dark blue) and TIM23 (cyan) translocases and cytosolic chaperones (yellow) are indicated. N = 4 replicates/sgRNA on one occasion. (B) Flow cytometry measurements of PINK1-YFP intensity (top) and intensity of MTS-mCh (bottom) from the same HeLa[PINK1-YFP+MTS-mCh] cells. N = 6 biological replicates measured from two independent transductions. All statistical comparisons are to the vehicle-treated CTRL guide group. Exact *P* values, ns*P* = 0.2179, ****P* ≤ 0.0001 (exact *P* values *P* = 1.2e-14 for all comparisons) by two-way ANOVA with Dunnett's multiple comparison test. Error bars mean +/− SD. (C) Representative confocal Airyscan images shown as z-projections in HeLa[PINK1-YFP] cells immunostained for TOMM70 to mark mitochondria. The arrow indicates PINK1-YFP accumulated around lipid droplets; the arrowhead indicates PINK1 co-localizing with TOMM70. Scale bar = 10 μm. (D) CLEM images demonstrating that the PINK1-YFP accumulates around lipid droplets (LD) following loss of the TOM translocase (HeLa[PINK1-YFP] cells). ER (yellow arrowheads) and small vesicles (red arrows) were adjacent to lipid droplet collections and accumulated PINK1-YFP. Scale bars: green = 20 μm; yellow = 5 μm; black = 1 μm; white = 500 nm. (E) Representative confocal images of TIMM23 KO pools in HeLa[PINK1-YFP] cells showing PINK1-YFP accumulates on mitochondria in cells lacking TIMM23 by immunostaining. Scale bar = 10 μm. (F) Line scan of dotted line in (E). (G) Quantification of (E). ****P* ≤ 0.0001 (exact *P* value *P* = 9.0037e-05) via two-tailed unpaired *t* test with Welch's correction. N = 4 wells analyzed, plated on two separate occasions from the same KO pool. Cells were analyzed 7 or 8 days after electroporation, and cells without TIMM23 were scored for PINK1-YFP accumulated on TOMM40-positive mitochondria. Error bars mean +/− SD. (H) Representative immunoblots from HeLa cells with exogenous PINK1-YFP (left blot) or endogenous PINK1 (right blot) +/− 10 μM CCCP treatment for 4 h, probing for PINK1 activation and stabilization. N ≥ 3 independent experiments. (I) Volcano plot shows PINK1-YFP (red) interactors, following CTRL vs. TIMM23 KD in HeLa[PINK1-YFP] cells. Two-sided Student's *t* tests were performed. Values were corrected for multiple comparisons by calculating a FDR with the permutation method. Proteins were annotated as significant interactors if they had an absolute log2 fold change of >1 (black outline). Proteins associated with TOM (dark blue) and TIM23 (cyan) translocases and cytosolic chaperones (yellow) are indicated. N = 4 replicates/sgRNA on one occasion. (J) In gel fluorescence of PINK1-YFP complexes separated by CN-PAGE after mixing lysates with the indicated antibodies. The specific interaction of the antibody with PINK1-YFP-containing complex increases the molecular weight of the complex, causing it to shift up in the gel. 10 μM CCCP treatment in the indicated lanes was for 3 h. N = 2 independent experiments with TOMM20 gel shift, one of which also tested TOMM22 and TOMM40 antibodies. (HeLa[PINK-YFP] cells). Source data are available online for this figure.

Recently, it has been demonstrated that precursor import along the TOM–TIM23 pathway can be blocked in mammalian cells using fusion proteins that clog the mitochondrial import pore in TOM (Krakowczyk et al, 2024; Hsu et al, 2025). These fusion proteins contain a N-terminus that is directed to the IMM or matrix and a tightly folded C-terminus in the cytosol that cannot be pulled through the TOM complex. This results in clogging of the TOM pore and limits the import of some precursors. We predicted that clogging a high proportion of TOM pores might prevent PINK1 activation on the OMM by blocking available TOM binding sites for PINK1 import. We first evaluated the recently described mitochondrial import clogger ATP5MG-mCherry-sfGFP (Krakowczyk et al, 2024). We transiently expressed ATP5MG-mCherry-sfGFP in a MFN2-Halo reporter line that co-expressed TagBFP2-Parkin, as this combination was compatible with the fluorophores in ATP5MG-mCherry-sfGFP. In cells with the highest ATP5MG-mCherry-sfGFP expression, MFN2-Halo degradation by TagBFP2-Parkin was substantially blocked following CCCP treatment (Fig. EV4C,D). Notably, this was not observed in the population of cells with more modest ATP5MG-mCherry-sfGFP expression (Fig. EV4C,D). A second mitochondrial clogger (IMMT-DHFR) was also tested (Hsu et al, 2025). IMMT-DHFR was found to have a minimal effect on PINK1 stabilization and activation (Fig. EV4E), as also reported previously (Hsu et al, 2025). Together, these results suggest that clogging of the TOM translocase by other stalled precursors can inhibit PINK1–Parkin activation, but likely at only high (super-physiologic) levels.

## PAM import motor facilitates but is not required for PINK1 import

Our results so far suggest that TIM23 block leads to PINK1 stabilization in the TOM complex. We next considered whether blocking the PAM import motor might similarly stabilize PINK1 (Makki and Rehling, 2023). The PAM motor associates with the TIM23 complex and is required for the import of some but not all substrates along the precursor import path. One component of the PAM motor, HSPA9, but not others (e.g., TIMM44 and PAM16), was identified as a top activator in the screen. However, as HSPA9 has other functions in the mitochondrial matrix, including folding of newly imported proteins and Fe-S cluster biosynthesis, its disruption may activate the PINK1–Parkin pathway independently of its function in import. Notably, DNAJA3, a co-chaperone of HSPA9, was also identified as an activator in the screen. DNAJA3 participates in the folding of newly imported proteins but does not function in the PAM motor (Wiedemann and Pfanner, 2017; Shin et al, 2021).

The dual role of HSPA9 in import and protein folding has led to a proposal that misfolded proteins can activate the PINK1–Parkin pathway independently of the MMP by competing with HSPA9 away from the PAM complex (Michaelis et al, 2022). To assess this proposal, we first used immunoblotting to directly compare disruption of the HSPA9–DNAJA3 complex to disruption of the HSPA9-containing PAM complex (Fig. 6A,B). KD of both PAM16 and DNAJA3 blocked import of the ATP5A precursor (Fig. 6A). However, only KD of DNAJA3 caused endogenous PINK1 stabilization and activation, similar to KD of TIMM23 (Fig. 6A). As an orthogonal strategy, we used confocal microscopy to compare KO pools of TIMM23 and TIMM44 for their ability to stabilize PINK1-YFP in cells with blocked import of MTS-mCh (Fig. 6C,D). A greater PINK1-YFP increase was observed, followed by an import block with TIMM23 KO compared to TIMM44 KO (Fig. 6C,D). Similarly, PAM16 and TIMM44 KD caused less PINK1-YFP accumulation than TIMM23 KD when measured by flow cytometry (Fig. EV4F,G). Thus, the PAM motor is not required for PINK1 import under endogenous conditions but may facilitate PINK1 import in the setting of higher levels of PINK1-YFP by exogenous expression.

We next examined the mechanism of PINK1 stabilization by DNAJA3 KD in greater detail. We first sought to confirm that our DNAJA3 KD strategy causes protein aggregation in the mitochondrial matrix (Shin et al, 2021). Using quantitative proteomics, we

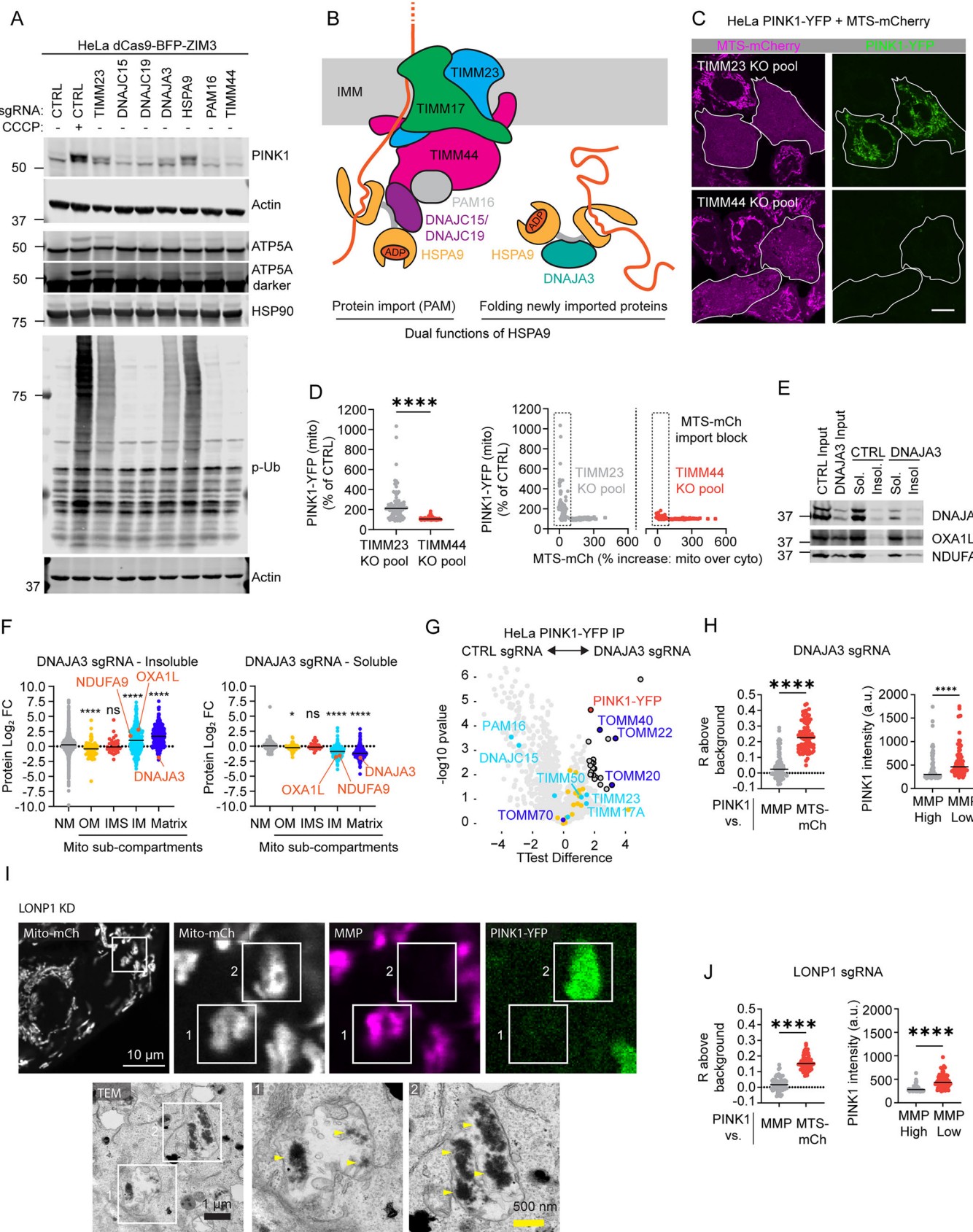

**Figure 6.   PINK1 is stabilized by protein misfolding but not disruption of the PAM import motor.**

(A) Representative immunoblot comparing endogenous PINK1 activation to import block of ATP5A as described for (Fig. 2J) in HeLa$^{dCas9-BFP-ZIM3}$ cells. $N = 3$ independent experiments. (B) Schematic showing the dual functions of HSPA9 in the PAM import motor and folding of newly imported proteins. HSPA9 performs the latter with its co-chaperone DNAJA3. (C) Representative confocal images of PINK1-YFP and MTS-mCh in TIMM23 and TIMM44 KO pools 7 days after electroporation. Chronic import block through the TIM23 translocase is identified by accumulation of MTS-mCh in the cytosol (cells with white outlines). Scale bar $= 10$ μm. (HeLa$^{PINK1-YFP+MTS-mCh}$ cells). (D) Quantification of the experiment in (C). Mitochondrial PINK1-YFP intensity was measured in the subset of cells with import block (left) and correlation of PINK1-YFP with import block of MTS-mCh for all cells (right). ****$P \leq 0.0001$ (exact $P$ value $P < 1e{-}15$) by Mann–Whitney test. 89 or more cells per condition were analyzed from $N = 4$ wells, plated on two separate days, from the same KO pool. Cells were analyzed 7 days after electroporation. (E) Immunoblot of 1% Triton X100 soluble and insoluble fractions from whole-cell lysates, probing for OXA1L and NDUFA9. $N = 3$ replicates on at least two occasions in HeLa$^{dCas9-BFP-ZIM3}$ cells. (F) LFQ proteomics depicting protein abundance from the 1% Triton X100 insoluble and soluble heavy membrane fractions, normalized to the median value of non-mitochondrial proteins in the sample. Left ns$P = 0.9827$, right ns$P = 0.9930$, *$P = 0.0489$, ****$P \leq 0.0001$ (Insoluble exact $P$ values from left to right, $P = 7.05e{-}05$, $P = 1.42e{-}09$, $P < 1e{-}15$; Soluble exact $P$ values from left to right, $P < 1e{-}15$, p $< 1e{-}15$) by Brown–Forsythe and Welch ANOVA tests with Dunnett's T3 multiple comparisons test. $N = 4$ replicates/sgRNA on one occasion. (HeLa$^{dCas9-BFP-ZIM3}$ cells). (G) Volcano plot showing PINK1-YFP (red) interactors, following CTRL vs. DNAJA3 KD in HeLa$^{PINK1-YFP}$ cells. Two-sided Student's $t$ tests were performed. Values were corrected for multiple comparisons by calculating a FDR with the permutation method. Proteins were annotated as significant interactors if they had an absolute log2 fold change of >1 (black outline). Proteins associated with TOM (dark blue) and TIM23 (cyan) translocases and cytosolic chaperones (yellow) are indicated. $N = 4$ replicates/sgRNA on one occasion. (H) The left graph quantification of cells in (Fig. EV5A) was performed as in (Fig. 3G). ****$P \leq 0.0001$ (exact $P$ value $P < 1e{-}15$) by two-tailed Mann–Whitney test. $N = 86$ cells from six wells and two separate transductions. The right graph quantification of cells in (Fig. EV5A) was performed as in (Fig. 3F). ****$P \leq 0.0001$ (exact $P$ value $P < 1e{-}15$) by two-tailed Wilcoxon matched-pairs signed rank test. $N = 91$ cells from six wells and two separate transductions. (I) CLEM images from a LONP1 KD HeLa$^{PINK1-YFP+MTS-mCh}$ cell demonstrating that mitochondrial aggregates (yellow arrowheads) were observed both in mitochondria that retained MMP (white box 1 in image) and mitochondria that had lost MMP (white box 1 in image) via TEM, but PINK1-YFP preferentially accumulated on aggregate-containing mitochondria that had lost MMP (white box 2 in image). Scale bars white $= 10$ μm, black $= 1$ μm, yellow $= 500$ nm. (J) Left graph quantification of cells in (Fig. EV5E) was performed as in (Fig. 3G). ****$P \leq 0.0001$ (exact $P$ value $P < 1e{-}15$) by two-tailed Mann–Whiteney test. $N = 80$ cells from 4 wells and 2 separate transductions. Right graph quantification of cells in (Fig. EV5E) was performed as in (Fig. 3F). ****$P \leq 0.0001$ (exact $P$ value $P < 1e{-}15$) by Wilcoxon matched-pairs signed rank test. $N = 81$ cells from four wells and two separate transductions. Source data are available online for this figure.

measured the abundance of proteins following the separation of mitochondria into soluble and insoluble fractions by differential detergent fractionation. DNAJA3 KD globally reduced the solubility of matrix and IMM proteins, as expected (Fig. 6E,F; Dataset EV6). These insoluble proteins included the established substrates of DNAJA3: OXA1L and NDUFA9. These findings are also consistent with our observations by EM above, which showed visible matrix aggregates following DNAJA3 KD (Figs. 3C and EV1). We next asked whether PINK1-YFP stabilized by DNAJA3 KD accumulates in a similar complex with TOM and TIM23, as is observed with pharmacological disruption of the MMP with CCCP. By AP-MS, PINK1-YFP pulled down the same subunits of the TOM and TIM23 translocase regardless of whether it was stabilized by DNAJA3 KD or CCCP treatment (Fig. 6G; Dataset EV5). This demonstrates that PINK1 accumulates in a TOM-TIM23 super-complex with a similar composition in response to matrix protein misfolding or pharmacological uncoupling.

Examining single cells by flow cytometry, PINK1-YFP accumulation was tightly correlated with MMP loss following DNAJA3 KD as it was for most activators (Fig. EV2B). This is consistent with DNAJA3 causing PINK1 accumulation secondary to MMP loss. To evaluate this further with an orthogonal method, we assessed the correlation between PINK1-YFP accumulation and MMP loss by live confocal microscopy. Consistent with single-cell analysis by flow cytometry, PINK1-YFP was elevated selectively on mitochondria with low MMP (Figs. 6H and EV5A). This followed the same pattern as NDUFAB1 and PMPCB and was distinct from the pattern resulting from direct import block with TIMM23 KD (Fig. 4H–J). Together, these findings suggest that misfolded proteins from DNAJA3 KD activate the PINK1–Parkin pathway by disrupting the MMP.

If matrix protein misfolding activates PINK1-YFP through MMP loss, this correlation should be seen following the disruption of other matrix chaperones and proteases needed to maintain matrix proteostasis. To test this prediction, we disrupted select matrix chaperones and proteases, including the quality control protease LONP1, subunits of the mtHSP60 chaperonin (HSPD1 and HSPE1), and the mtHSP90 chaperone TRAP1. These factors are important for the folding and degradation of endogenous mitochondrial matrix proteins after they complete import. We first assessed the KD of each on PINK1-YFP and MMP intensity by flow cytometry. Similar to DNAJA3 KD and the other activator assays above (apart from TIMM23), cells fell into two populations with either high MMP/low PINK1 or low MMP/high PINK1 (Fig. EV5B,C), indicating a strong correlation between MMP loss and PINK1-YFP accumulation. To test whether disruption of mitochondrial proteostasis had similar effects on endogenous PINK1 activation, we next assessed the same gene perturbations by immunoblotting (Fig. EV5D). The perturbations that most strongly increased PINK1-YFP levels by flow cytometry also most strongly stabilized and activated endogenous PINK1. The extent of import block, measured by the accumulation of the ATP5A precursor, also correlated with extent of PINK1 activation. Together, this suggests that disruption of mitochondrial proteostasis in the matrix activates PINK1 by the same mechanism as observed with the other mitochondrial activators. MMP loss leads to import block of PINK1, causing its stabilization and activation on the mitochondrial surface.

We next examined select knockdowns by live confocal microscopy. Perturbation of either LONP1 or HSPD1 resulted in single cells containing a heterogeneous population of mitochondria with respect to MMP, similar to the disruption of NDUFAB1, PMPCB, and DNAJA3 above. As with the other perturbations, PINK1-YFP accumulated preferentially on mitochondria with low MMP in single cells following either LONP1 or HSPD1 KD (Figs. 6I,J and EV5E–G). For LONP1 KD, some cells were fixed immediately after measuring MMP live and processed for CLEM. By TEM, mitochondrial aggregates could be directly observed both in mitochondria that retained MMP and mitochondria that had lost MMP (Fig. 6I). PINK1-YFP accumulated preferentially on

aggregate-containing mitochondria that had lost MMP. This suggests that aggregate formation is not the direct stimulus for PINK1 accumulation; instead, PINK1 accumulates only after MMP is lost. Together, these results demonstrate that disruption of mitochondrial matrix proteostasis activates PINK1 by disruption of the MMP.

Finally, we tested a mutant form of ornithine carbamoyltransferase (ΔOTC), previously shown to activate the PINK1–Parkin pathway in an MMP-independent manner (Burman et al, 2017; Jin and Youle, 2013). Again, two populations were seen, but the population with increased PINK1-YFP had an MMP that was only mildly reduced (Fig. EV5H). ΔOTC stress may differ from matrix protein misfolding following loss of proteostasis, as ΔOTC must pass through TIM23 as an aggregation-prone protein. This may disrupt the TIM23 import path (analogous to TIMM23 KD) in addition to lowering MMP, although this will require further investigation to confirm.

## TOMM20 and TOMM5 are required for PINK1 stabilization on the OMM, while TOMM70 is dispensable

Our findings so far suggest that TOM but not the TIM23 translocase is required for PINK1 stabilization on the OMM, and that the MMP is the primary driving force for PINK1 import through the TIM23 translocase. We next explored what specific subunits of the TOM translocase are required to hold PINK1 on the OMM, in addition to the core subunits TOMM40 and TOMM22 (Fig. 5C). The TOM translocase has two receptors for mitochondrial precursors in addition to TOMM22: TOMM20 and TOMM70 (Araiso et al, 2022). We first examined whether these are required for endogenous PINK1 stabilization and activation by immunoblotting. Consistent with recent reports (Eldeeb et al, 2024; Raimi et al, 2024), we found TOMM20 KD blocks PINK1 stabilization following OXPHOS inhibition (Fig. 7A, left, and B, left blot, lane 4 vs. 2). By contrast, we found that TOMM70, which is required for PINK1 import in in vitro import assays and in yeast (Kato et al, 2013; Raimi et al, 2024), was not required for PINK1 import in intact mammalian cells (Fig. 7A, left, and B, left blot, lane 6 vs. 2).

Consistent with the relative importance of TOMM20 vs. TOMM70 for PINK1 import, PINK1-YFP pulled down TOMM20 but not TOMM70 by AP-MS, following treatment with CCCP (Fig. 5A). Interestingly, we found the situation was different, however, for PINK1-YFP accumulated by TOMM40 KD (Fig. 7C; Dataset EV5). Under these conditions, PINK1-YFP additionally pulled down cytosolic HSP70 chaperones and co-chaperones, which are established binding partners of TOMM70 (Araiso et al, 2022) (Fig. 7C; Dataset EV5). We additionally examined PINK1-YFP complex formation following TOMM40 KD by CN-PAGE. Most of the PINK1-YFP in TOMM40 KD cells was found in a smear between 480 and 720 kDa that did not shift following incubation with an antibody against TOMM22, in contrast to PINK1-YFP stabilized by CCCP treatment or DNAJA3 KD (Figs. 7D and EV5I). Together, these data suggest that TOMM70 may bind some excess unfolded PINK1 that cannot be incorporated into the TOM translocase or degraded in the cytosol. It is likely that most PINK1-YFP is not directly bound to TOMM70, however, as PINK1-YFP does not strongly co-localize with TOMM70-positive mitochondria (Fig. 5C). At the same time, these results demonstrate

that TOMM70 is not required for endogenous PINK1 import or stabilization in the TOM translocase.

Finally, we considered the remaining three small subunits of the TOM complex, TOMM5, TOMM6, and TOMM7 (Araiso et al, 2022) (Fig. 7A,B,E,F). TOMM7 has previously been shown to be critical for PINK1 stabilization in the TOMM40 complex (Hasson et al, 2013), however, the mechanism has not been clear. We confirmed by immunoblotting that TOMM7 is required for PINK1 stabilization on the OMM. Surprisingly, we found that TOMM5 is also needed for endogenous PINK1 stabilization and activation (Fig. 7A, left, and B, right blot, lane 4 vs. 2 and lane 8 vs. 2). TOMM5 was previously identified as an accelerator of PINK1–Parkin mitophagy, as part of a screen, but how TOMM5 modulates the PINK1–Parkin pathway was not further explored (Hoshino et al, 2019). Using flow cytometry, we found that stabilization of PINK1-YFP could be restored by exogenous expression of full-length TOMM5 but not C-terminally truncated TOMM5 (Fig. 7E). These findings demonstrate that TOMM5 is critical at the initial step of the PINK1–Parkin pathway: stabilization of PINK1 in the TOM complex.

TOMM5 has been previously suggested to be involved in the biogenesis of the TOM translocase (Araiso et al, 2019). Thus, loss of TOMM5 may block PINK1 stabilization by destabilizing the TOM complex. To examine this, we compared the effect of small TOM subunit knockdown on the stability of the TOM translocase by quantitative proteomics. Notably, TOMM5 and TOMM7 were not required for the stability of the core subunits TOMM40 or TOMM22 or the receptor TOMM20 (Fig. 7G, blue data points; Dataset EV7). Consistently, neither TOMM5 nor TOMM7 was required for import of presequence-containing proteins through the TOM translocase (Fig. 7G, red data points; Dataset EV7). This was in sharp contrast to TOMM40 KD or import block with CCCP overnight. Both strongly depleted presequence-containing mitochondrial proteins (Fig. 7G, red data points; Dataset EV7). In the previously published TOM structures, TOMM5 and TOMM7 are not in direct contact, but are predicted to stabilize the interior of the translocase pore, with TOMM5 helping to shape the import path for hydrophobic precursors lacking an import presequence, in part by stabilizing the N-terminal extension of TOMM40 (Fig. 7F) (Araiso et al, 2019). Consistently, the abundance of presequence-less mitochondrial proteins was reduced following KD of TOMM5 (but not TOMM6 or TOMM7) (Fig. 7G), demonstrating functional conservation between mammalian TOMM5 and its yeast ortholog (Araiso et al, 2019). Together, these findings suggest that TOMM5 and TOMM7 may stabilize a binding site for PINK1 in the translocase overlapping with the import path for proteins lacking a precursor sequence. This binding site is likely separate from the previously identified binding site between PINK1 and TOMM20 (Eldeeb et al, 2024; Raimi et al, 2024), as TOMM20 was not destabilized by loss of the small TOM subunits.

## Discussion

Here, we used a novel single-cell reporter for the PINK1–Parkin pathway, MFN2-Halo, to identify facilitators and activators of the PINK1–Parkin pathway in genome-wide screens. Notably, functionally diverse mitochondrial perturbations, including those causing severe mitochondrial protein misfolding, converged on

   

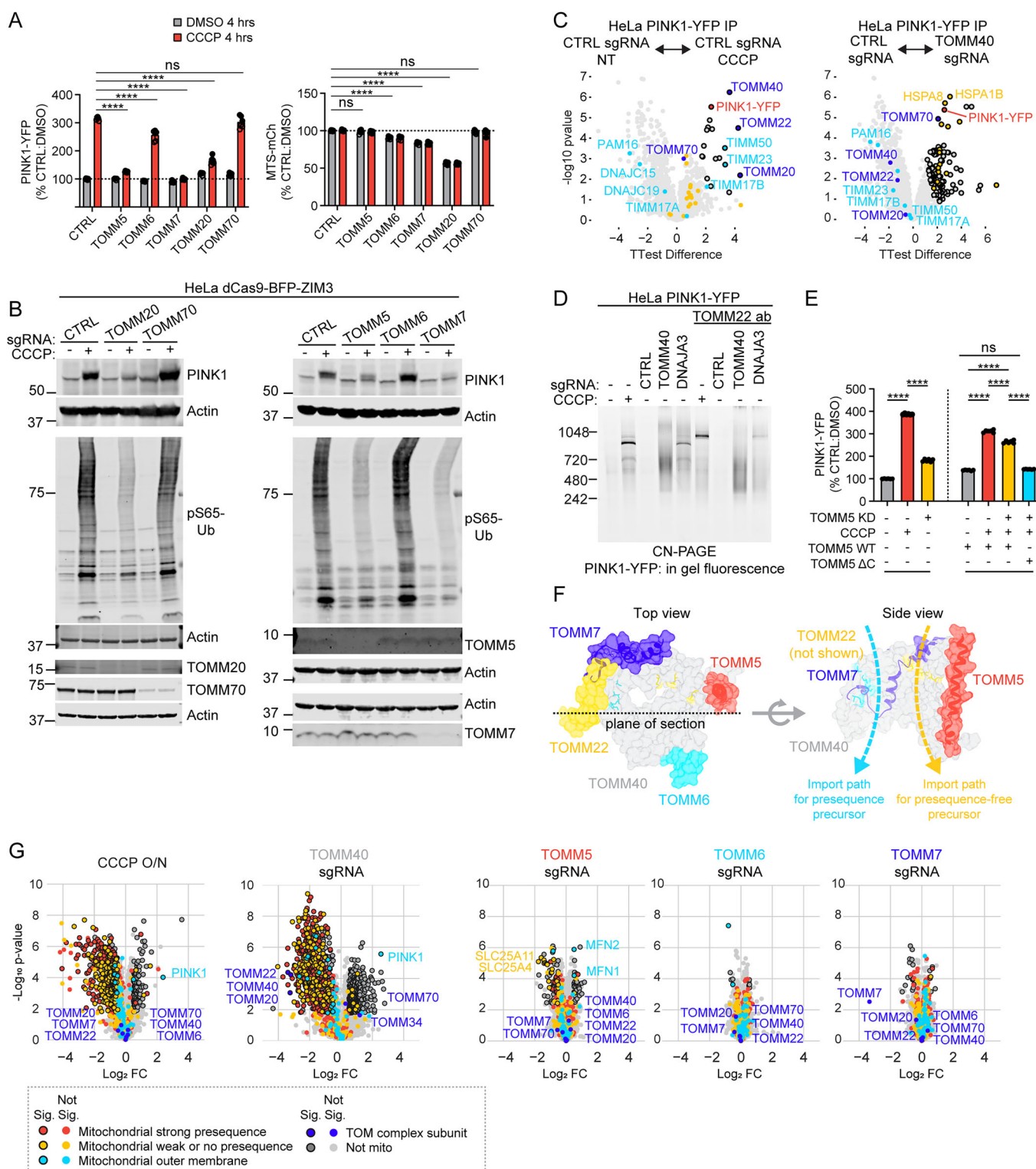

loss of MMP as the primary mechanism of PINK1–Parkin activation. Consistently, we established that the MMP is the main driving force for PINK1 import through the TIM23 translocase, with mitochondrial ATP and the PAM motor playing supporting roles (Appendix Fig. S7).

Surprisingly, TIMM23 KD was among the novel PINK1–Parkin activators identified. In contrast to the other activators, TIMM23 KD stabilized PINK1 on energized mitochondria in complex with the TOM translocase, suggesting that block of transport through the TIM23 translocase is a key step for PINK1 stabilization and

◄ **Figure 7. TOMM20 and TOMM5 are required for PINK1 stabilization on the OMM; TOMM70 is not.**

(A) Flow cytometry measurements performed as in (Fig. 5B). Left graph ns$P$ = 0.0988, Right graph from left to right ns$P$ = 0.5132, 0.2695, ****$P \leq$ 0.0001 (left graph exact $P$ values from left to right $P$ = 7e-15, 1.7e-14, 7e-15, 7e-15; right graph exact $P$ values from left to right 4.62e-11, 7e-15, 7e-15) by two-way ANOVA with Dunnett's multiple comparisons test. Error bars mean +/− SD. $N$ = 6 independent experiments from two separate transductions (separate transductions denoted by open or closed circles). Error bars mean +/− SD. (B) Representative immunoblots of PINK1 stabilization and activity in HeLa$^{dCas9-BFP-ZIM3}$ cells with indicated sgRNA +/− 10 μM CCCP for 4 h. $N \geq 3$ independent experiments. (C) Volcano plots of PINK1-YFP (red) interactors measured by AP-MS as in (5I). CCCP treatment, where indicated, was overnight. Other samples were untreated. Two-sided Student's $t$ tests were performed. Values were corrected for multiple comparisons by calculating a FDR with the permutation method. Proteins were annotated as significant interactors if they had an absolute log2 fold change of >1 (black outline). Proteins associated with TOM (dark blue) and TIM23 (cyan) translocases and cytosolic chaperones (yellow) are indicated. $N$ = 4 replicates/sgRNA on one occasion. The untreated control guide group was the same as used in (Fig. 6G). (D) CN-PAGE separated PINK1-YFP complexes visualized by in-gel fluorescence as in (Fig. 5J). $N$ = 2 independent experiments with TOMM40 KD, one of which was with antibody gel shift. (E) Flow cytometry in HeLa$^{PINK-YFP}$ cells with doxycycline inducible expression of wild-type or C-terminal truncated TOMM5. Demonstrating rescue with TOMM5 WT but not TOMM5 ΔC. $N$ = 6 replicates on at least two occasions. ns$P$ = 0.4357, ****$P \leq$ 0.0001 (exact $P$ values $P$ < 1e-15 for all comparisons) by ordinary one-way ANOVA with Šídák's multiple comparisons test. (F) Crystal structure yeast TOM complex (PDB: 6JNF (Araiso et al, 2019)) demonstrating the location of TOMM5 and TOMM7 subunits. (G) LFQ proteomics of HeLa$^{PINK-YFP}$ whole lysates following the indicated knockdown/treatment. Two-sided Student's $t$ tests were performed. Values were corrected for multiple comparisons by calculating an FDR with the Benjamini–Hochberg procedure. Proteins were annotated as significant if they had an FDR < 0.05 and an absolute log$_2$ fold change of >0.5 (black outline). $N$ = 4 replicates/sgRNA on one occasion. Source data are available online for this figure.

activation. We further identified components of the TOM translocase required for PINK1 stabilization on the OMM, including a role for TOMM5, which may stabilize a PINK1 binding site in the TOM translocase distinct from the established TOMM20 binding site. Finally, we identified that the PINK1–Parkin response is additionally shaped by the overall energetic state of the cell due to the high-energy requirement for new PINK1 synthesis. Together, these findings support the following model for damage-sensing in the PINK1–Parkin mitophagy pathway: the PINK1–Parkin pathway is triggered when MMP across the TIM23 translocase is insufficient to complete PINK1 import during the transfer from TOM to TIM23.

This model helps clarify several open questions in the field. Mitochondrial uncoupling with CCCP was shown to be sufficient to activate the PINK1–Parkin pathway over a decade ago (Narendra et al, 2010, 2008), leading us and others to propose that activation of the PINK1–Parkin pathway is due to import block of PINK1 along the precursor pathway (Youle and Narendra, 2011). This model, however, was seemingly at odds with recent studies, which found that TIMM23 knockdown by transient siRNA transfection (for 2–3 days) or small molecule inhibitors does not phenocopy PINK1 stabilization by OXPHOS inhibition (Akabane et al, 2023; Sekine et al, 2019; Filipuzzi et al, 2017). In fact, acute KD of TIMM23 was found to inhibit PINK1–Parkin activation by OXPHOS inhibition. In this condition, OMA1 was shown to degrade PINK1 that was no longer shielded by the TIM23 complex, thereby suppressing the PINK1–Parkin pathway. Using methods that allowed for more sustained TIMM23 knockdown or knockout in single cells (assayed at 7–8 days after transduction or electroporation), we found that TIMM23 depletion is sufficient to stabilize and activate PINK1. As OMA1 levels are likely depleted by the sustained import block under these conditions, OMA1 may no longer be limiting for PINK1 accumulation in the TOM complex. This clarifies that the import block at TIM23 translocase is a key step in PINK1 stabilization on the OMM and reinforces the import block model of PINK1–Parkin activation.

Our results additionally help clarify which components of the recently identified PINK1-TOM-TIM23 supercomplex are required for PINK1 stabilization on the OMM (Akabane et al, 2023; Eldeeb et al, 2024). While we confirm that PINK1 binds both TOM and TIM23 translocases, we found that only the TOM translocase is required for PINK1 stabilization on the OMM. Stable contacts with the TIM23 translocase, while dispensable for PINK1 stabilization

on the OMM, may enable rapid import of PINK1 to the IMM for cleavage, if the MMP is restored (Jin et al, 2010). They may also protect import-arrested PINK1 from intermembrane space proteases such as OMA1, as recently proposed (Akabane et al, 2023). Among the TOM subunits required for PINK1 stabilization, we confirmed that previously identified subunits TOMM40, TOMM22, TOMM20, and TOMM7 are essential (Eldeeb et al, 2024; Hasson et al, 2013; Raimi et al, 2024), and additionally identified an essential role for TOMM5. TOMM5 was also previously identified as an accelerator of PINK1–Parkin mitophagy in a screen but was not characterized in detail (Hoshino et al, 2019). Here, we demonstrated that TOMM5 is specifically required for PINK1 stabilization in the TOM complex, and that the mechanism does not involve global destabilization of TOM40. Rather, our findings support a model in which TOMM5 helps maintain specific contacts between PINK1 and TOMM5 as also suggested by recent structural work discussed further below. By contrast, we found that TOMM70 was not required for endogenous PINK1 import but bound excess PINK1-YFP when the TOM translocase was depleted. This likely explains why prior studies found that TOMM70 is required for exogenous PINK1 import in in vitro assays and following exogenous PINK1 expression in yeast (Raimi et al, 2024; Kato et al, 2013). In these assays, exogenous PINK1 may exceed the capacity of the precursor pathway and reversibly bind cytosolic HSP70 chaperones and TOMM70.

While our manuscript was under review, a structure of human PINK1 in complex with the TOM translocase was reported (Callegari et al, 2025). The structural details agree with our functional data in several key respects. First, they confirm extensive contacts between PINK1 and TOMM20 that participate in the stabilization of PINK1 on the OMM. Second, they establish that both direct and indirect contacts form between TOMM5 and PINK1. These indirect contracts include TOMM5 positioning the N-terminal segment of TOMM40 and VDAC2 relative to PINK1. These contacts provide a structural basis for our observation that TOMM5 is required along with TOMM7 for PINK1 stabilization and activation. Third, the structure lacks TOMM70, consistent with our finding that TOMM70 is dispensable for PINK1 activation in cells. Together, these structural findings are highly complementary with the functional data reported here.

Another open question concerns the driving force for PINK1 import along the precursor path: is only the MMP required or is the

 

PAM import motor also needed? This question has been made more relevant by a recent proposal that the PAM motor may "sense" protein misfolding in the mitochondrial matrix to activate the PINK1–Parkin pathway independently of the MMP (Michaelis et al, 2022). This mechanism assumes that the PAM motor is essential for PINK1 import, but this assumption was not directly evaluated (Michaelis et al, 2022). Here, we identified that the MMP is the primary driving force for PINK1 import across the TIM23 translocase, while the PAM motor is not required for endogenous PINK1 import. The greater dependence of PINK1 on MMP than PAM for import is supported by prior work in yeast: import of single-pass proteins to the IMM, which involves translocation of only a short N-terminal stretch of the positively charged amino acids across the IMM, is less dependent on the PAM motor than precursor import to the mitochondrial matrix, involving a substantially longer and more varied stretch of amino acids (Makki and Rehling, 2023). Consistently, we found that import of the matrix-targeted proteins ATP5A1 and MTS-mCh were more sensitive to disruption of the PAM import motor.

While our work resolves several questions regarding how PINK1 import to the IMM is connected to PINK1–Parkin activation, it is worth noting that our work does not directly address whether PINK1 may also be imported to the mitochondrial matrix, as has been reported previously (Morais et al, 2014). However, there are some implications of our results for PINK1 import to the mitochondrial matrix. First, our results confirm that the majority of PINK1 imported into mitochondria under basal conditions is likely released into the IMM (Jin et al, 2010; Deas et al, 2011). This is supported by the observation that PINK1 is strongly stabilized by blocking its PARL-dependent cleavage in the IMM or by inhibitors of its degradation by the cytosolic proteasome. Matrix-targeted PINK1 should be resistant to these blocks in PINK1 processing. Import of PINK1 to the IMM is thought to follow the route of other IMM-resident transmembrane domain-containing proteins (Makki and Rehling, 2023). Import proceeds through a groove in the TIM23 translocase (located in the TIMM17A/B subunit) until it reaches the hydrophobic transmembrane region. This sequence stops the transfer of the precursor, and the precursor is laterally released by TIM23 into the IMM. For PINK1 to continue import into the mitochondrial matrix, however, it would need to bypass this stop transfer event in the TIM23 translocase. Matrix import of PINK1 would also be predicted to require the action of the PAM motor, as it does for other matrix-directed precursors (Makki and Rehling, 2023), as the whole length of PINK1 (581 residues) and not just the positively charged N-terminus would need to pass through the TIM23 channel. Notably, we did not see accumulation of endogenous PINK1 after PAM disruption. Thus, our results do not provide evidence for a pool of PINK1 in the mitochondrial matrix, but they also don't rule it out.

Together, our data suggest MMP is key to PINK1 stress sensing, as MMP is the primary driving force for import, as PINK1 is transferred from the TOM to the TIM23 translocase along the precursor import path. Importantly, this proposed model also points to how the PINK1–Parkin pathway may be pharmacologically tuned to promote clearance of damaged mitochondria in disorders caused by mitochondrial damage. Small molecules that upregulate glycolysis, such as the PGK1 activator terazosin (Riley et al, 2024; Chen et al, 2015; Kokotos et al, 2024), may help support PINK1–Parkin surveillance of mitochondrial damage

by providing ATP needed for new PINK1 synthesis and PINK1–Parkin activation. This may be particularly critical in sporadic Parkinson's disease, where energy deficiency is likely present in the affected dopamine neurons (Kokotos et al, 2024). Additionally, the MMP threshold of PINK1–Parkin activation may be lowered by decreasing mitochondrial ATP levels. This may have an effect that is similar to MTK458, a PINK1 activator under development by Mitokinin/AbbVie, which was shown to lower the CCCP dose required for PINK1–Parkin activation (Chin et al, 2023), similar to the effect of oligomycin observed here. The driving force required for PINK1 import through TIM23 may also be tuned through its binding to other proteins in the IMM, such as TIMM44 and ANT1/2 (Hoshino et al, 2019). Conversely, molecular glues that stabilize the binding site for PINK1 in the TOM complex, buttressed by TOMM5 and TOMM7, may allow PINK1 stabilization at a higher MMP. Either strategy—strengthening PINK1 binding to the TOM translocase or lowering the driving force through the TIM23 translocase – would have a similar effect: increasing the sensitivity of the PINK1–Parkin mitophagy pathway for damaged mitochondria. Finally, we anticipate the MFN2-Halo reporter will help aid drug discovery, as a quantitative, single-cell reporter that is orthogonal to the widely used mitophagy-based reporters.

## Methods

### Reagents and tools table

| Reagent/resource | Reference or source | Identifier or catalog number |
| --- | --- | --- |
| **Experimental models** | | |
| HeLa dCas9 BFP KRAB | Hsu et al, 2025 | N/A |
| HeLa dCas9 BFP KRAB MFN2 Halo | This study | N/A |
| HeLa dCas9 BFP KRAB MFN2 Halo mCherry Parkin | This study | N/A |
| HeLa dCas9 BFP KRAB MFN2 Halo BFP Parkin | This study | N/A |
| HeLa dCas9 BFP ZIM3 | This study | N/A |
| HeLa PINK1 KO PINK1 eYFP dCas9 BFP ZIM3 | This study | N/A |
| HeLa PINK1 KO PINK1 eYFP dCas9 BFP ZIM3 TOMM70-Halo | This study | N/A |
| HeLa PINK1 KO PINK1 eYFP dCas9 BFP ZIM3 MTS-mCherry | This study | N/A |
| HeLa dCas9 BFP KRAB mt-Keima | This study | N/A |
| HeLa dCas9 BFP KRAB mt-Keima YFP Parkin | This study | N/A |
| HEK293 dCas9 BFP ZIM3 MFN2 Halo | This study | N/A |
| i11W-mNC iPSC | Fernandopulle et al, 2018 | N/A |
| HeLa PINK1 KO PINK1 YFP | Gift from Dr. Shiori Sekine | Clone 21 |

| Reagent/resource | Reference or source | Identifier or catalog number |
|---|---|---|
| HeLa PINK1 KO PINK1 eYFP dCas9 BFP ZIM3 TOMM70-Halo IDF146-BFP | This study | N/A |
| WT HeLa | ATCC | RRID:CVCL_0030 |
| WT HEK293 | ATCC | RRID:CVCL_0045 |
| Lenti-X 293 T Cell Line | Takara/Clontech | 632180 |
| **Recombinant DNA** | | |
| pHAGE-mt-mKeima | Addgene | 131626 |
| pBMN-mCherry-Parkin | Addgene | 59419 |
| pHAGE-YFP-Parkin | Gift from Dr. Richard Youle | N/A |
| MTS-mCherry | This study | N/A |
| Dual sgRNA CRISPRi Library 1–2 | Addgene | 187246 |
| pJR103 | Addgene | 187242 |
| pSBtet-RN | Addgene | 60503 |
| pCMV(CAT)T7-SB100 | Addgene | 34879 |
| MFN2 Halo donor plasmid | This study, Addgene | 239866 |
| TOMM70 FKBP12 Halo donor plasmid | This study, Addgene | 239867 |
| ATP5MG-tFT | Krakowczyk et al, 2024 | N/A |
| iC30 PB-Zim3-d Cas9-P2A-BFP_hygro | Addgene | 236743 |
| IMMT-DHFR (IDF146-BFP) | This study, gift from Dr. Richard Youle | N/A |
| pEYFP-N1-PINK1 WT | Addgene | 101874 |
| pEYFP-N1-PINK1 ΔOMS (Δ74–93) | Sekine et al, 2019 | N/A |
| pEYFP-N1-PINK1 Δ94-110 | Sekine et al, 2019 | N/A |
| pEYFP-N1-PINK1 Δ111-117 | Sekine et al, 2019 | N/A |
| pEYFP-N1-PINK1 3EA (E112A, E113A, E117A) | Sekine et al, 2019 | N/A |
| pEYFP-N1-PINK1 (F104M) | This study, gift from Dr. Richard Youle | N/A |
| pLenti_ DualsgRNAEmptyVector_ Puro_T2A_BFP2 | This study, Addgene | 236729 |
| **Antibodies** | | |
| PINK1 | CST | 6946S |
| β-Actin | Sigma-Aldrich | A2228 |
| Parkin | CST | 2132S |
| HSP90 | Proteintech | 13171-1-AP |
| MFN2 | Abcam | ab56889 |
| β-Tubulin | Sigma-Aldrich | T8328 |
| pUb S65 | CST | 62802S |
| Puromycin | Sigma-Aldrich | MABE343 |
| ATP5A | Abcam | ab14748 |
| ENO1 | Proteintech | 11204-1-AP |
| PGAM1 | Proteintech | 16126-1-AP |

| Reagent/resource | Reference or source | Identifier or catalog number |
|---|---|---|
| HK2 | CST | 2867S |
| ALDOA | Proteintech | 11217-1-AP |
| TOMM22 | Proteintech | 66562-1-Ig |
| TOMM40 | Proteintech | 18409-1-AP |
| TOMM5 | Proteintech | 25607-1-AP |
| TOMM7 | ABclonal | A17711 |
| TOMM20 | Santa Cruz | sc-17764 |
| TOMM70 | Proteintech | 14528-1-AP |
| TID-1L/S (DNAJA3) | Santa Cruz | sc-18819 |
| OXA1L | Proteintech | 21055-1-AP |
| NDUFA9 | Abcam | ab14713 |
| TIMM23 | Santa Cruz | sc-514463 |
| DNAJC19 | Proteintech | 12096-1-AP |
| mtHSP70 (HSPA9) | Invitrogen | MA3-028 |
| TIMM44 | Proteintech | 13859-1-AP |
| PAM16 | Proteintech | 15321-1-AP |
| SYNJ2 | Proteintech | 13893-1-AP |
| CHCHD2 | Proteintech | 66302-1-Ig |
| MAP2 | Proteintech | 17490-1-AP |
| TUBB3 | BioLegend | 801201 |
| GFP | Abcam | ab6556 |
| nanogold-conjugated Fab secondary antibody | Nanoprobes | 2004 |
| Goat anti-Rabbit IgG (H + L) Highly Cross-Adsorbed Secondary Antibody, Alexa Fluor™ Plus 555 | Invitrogen | A32732 |
| Goat anti-Mouse IgG (H + L) Highly Cross-Adsorbed Secondary Antibody, Alexa Fluor™ Plus 488 | Invitrogen | A32723 |
| Goat anti-Mouse IgG (H + L) Highly Cross-Adsorbed Secondary Antibody, Alexa Fluor™ Plus 555 | Invitrogen | A32727 |
| IRDye® 800CW Goat anti-Rabbit IgG Secondary Antibody | LI-COR | 926-32211 |
| IRDye® 680RD Goat anti-Mouse IgG Secondary Antibody | LI-COR | 926-68070 |
| **Oligonucleotides and other sequence-based reagents** | | |
| sgRNA gBlock Sequences | This study | Dataset EV9 |
| Guide sequences for endogenous tagging via nucleofection – MFN2 sgRNA #1 | This study | ACCTGCAGCCC AGCAGATAG |
| Guide sequences for endogenous tagging via nucleofection – MFN2 sgRNA #1 | This study | atctgctggg ctgcaggtac |

| Reagent/resource | Reference or source | Identifier or catalog number |
| --- | --- | --- |
| Guide sequences for endogenous tagging via nucleofection – TOMM70 | This study | AACCACCAAC ATTATAAAAC |
| Screen PCR/Sequencing primers | This Study | Dataset EV8 |
| Gene Knockout Kit - human - TOMM40 | EditCo | human - TOMM40 |
| Gene Knockout Kit - human – TOMM22 | EditCo | human – TOMM22 |
| Gene Knockout Kit - human – TIMM44 | EditCo | human – TIMM44 |
| Gene Knockout Kit - human – TIMM23 | EditCo | human – TIMM23 |
| TOMM5 WT_gBlock | This study | Dataset EV9 |
| TOMM5 Truncated_gBlock | This study | Dataset EV9 |
| ERdAPEX_TagBFP2_NheI EcoRI_frag_gBlock | This study | Dataset EV9 |
| **Chemicals, enzymes, and other reagents** | | |
| Oligomycin | Sigma-Aldrich | 75351-5MG |
| Antimycin A | Sigma-Aldrich | A8674 |
| 2-deoxy-D-glucose (2-DG) | Sigma-Aldrich | D8375-5G |
| Carbonyl cyanide 3-chlorophenylhydrazone (CCCP) | Sigma-Aldrich | C2759-100MG |
| MG132 | Sigma-Aldrich | SML1135 |
| Cycloheximide | Sigma-Aldrich | C7698-5G |
| Heptelidic acid | Cayman Chem. | 14079 |
| Rotenone | Calbiochem | 557368-1GM |
| KL11743 | Millipore Sigma | SML3458-25MG |
| Aclar® 33 C Film | Electron Microscopy Sciences | 50425 |
| Paraformaldehyde 16% aqueous, EM grade | Electron Microscopy Sciences | 15710 |
| Glutaraldehyde 50% aqueous, EM grade | Electron Microscopy Sciences | 16320 |
| Glutaraldehyde, 10% aqueous grade | Electron Microscopy Sciences | 16120 |
| Sodium Cacodylate Buffer, 0.2 M, pH 7.4 | Electron Microscopy Sciences | 11653 |
| Calcium Chloride, 2 M | Quality Biological, Inc. | 351-130-721 |
| Glycine | Sigma-Aldrich | G7126 |
| 3,3'-Diaminobenzidine tetrahydrochloride hydrate | Sigma-Aldrich | D5637 |
| Hydrogen Peroxide | Sigma-Aldrich | H1009 |
| Osmium Tetroxide 4% aqueous solution | Electron Microscopy Sciences | 19150 |
| Potassium hexacyanoferrate (II) trihydrate | Sigma-Aldrich | P3289-100G |
| Sodium Acetate, Trihydrate Reagent | Electron Microscopy Sciences | 21120 |

| Reagent/resource | Reference or source | Identifier or catalog number |
| --- | --- | --- |
| Uranyl Acetate | Electron Microscopy Sciences | 22400 |
| Ethyl Alcohol, 200 proof | The Warner-Graham Co. | 6505001050000 |
| Embed 812 Embedding Kit with Dmp-30 | Electron Microscopy Sciences | 14120 |
| Gelatine capsule, size "0" | Electron Microscopy Sciences | 70110 |
| Slot grids with thick Formvar coating | Electron Microscopy Sciences | FF-2010-CU-TH |
| Reynold's Lead Citrate, 3% | Electron Microscopy Sciences | 22410 |
| Microscope Cover Glass, 18 mm | Fisherbrand, Fisher Scientific | 12-545-100 18CIR.-1 |
| SEM finder grid | Electron Microscopy Sciences | SEMF2-Cu |
| ProLong Gold Antifade Mountant | ThermoFisher Scientific | P36934 |
| Puromycin | Invivogen | ant-pr-1 |
| Janelia Fluor 646 HaloTag Ligand | Promega Corporation | GA1121 |
| GFP-Trap Magnetic Particles M-270 | chromotek | gtd |
| Lipofectamine® RNAiMax Reagent | Life Technologies | 13778 |
| ON-TARGETplus SMARTPool Human PINK1 | Dharmacon | L-004030-00-0005 |
| siGENOME Non-Targeting siRNA #1 | Dharmacon | D-001210-01-05 |
| LentiX Concentrator | Takara Bio | 631231 |
| BD Cytofix™ Fixation Buffer | BD Biosciences | 554655 |
| HiFi Cas9 Nuclease V3 | IDT | 1081061 |
| SE Cell Line 4D-Nucleofector X Kit L | Lonza | V4XC-1012 |
| SF Cell Line 4D-Nucleofector X Kit L | Lonza | V4XC-2024 |
| HDR enhancer | IDT | 1081073 or 10007921 |
| Lipofectamine RNAiMax Reagent | Life Technologies | 13778 |
| ON-TARGETplus SMARTPool Human PINK1 | Dharmacon | L-004030-00-0005 |
| siGENOME Non-Targeting siRNA #1 | Dharmacon | D-001210-01-05 |
| FuGENE 6 Transfection Reagent | Promega | E2691 |
| Lipofectamine 3000 Transfection Reagent | Invitrogen | L3000001 |
| hygromycin | MP Bio | 194170 |
| FuGENE HD Transfection Reagent | Promega | E2311 |
| ONE SHOT STBL3 COMP E COLI | Life Technologies|AB/ Invitrogen | C737303 |

| Reagent/resource | Reference or source | Identifier or catalog number |
|---|---|---|
| SPRI beads | Beckman Coulter | B23318 |
| NEBNext Ultra II Q5 Master Mix | New England Biolabs | M0544 |
| NEBuilder® HiFi DNA Assembly Master Mix | New England Biolabs | E2621 |
| high glucose DMEM with pyruvate | Gibco | 11995-065 |
| fetal bovine serum | R&D Systems | S11550 |
| Penn/Strep | Gibco | 15140-122 |
| DMEM without glucose and pyruvate | Gibco | 11966025 |
| Matrigel | CORNING | 354277 |
| Essential 8 Medium + Supplement | Gibco | A1517001 |
| Y-27632 ROCK inhibitor | Tocris Bioscience | 1254 |
| EDTA | Invitrogen | 15575-038 |
| accutase | Gibco | A1110501 |
| KnockOut DMEM/F12 | Gibco | 12660012 |
| N2 Supplement | Gibco | 17502048 |
| Non-essential amino acids | Gibco | 11140050 |
| L-glutamine | Gibco | 25030081 |
| Doxycycline | Millipore Sigma | D9891 |
| Poly-L-ornithine | Sigma-Aldrich | P3655 |
| BrainPhys neuronal medium | STEMCELL Technologies | 05790 |
| B27 supplement | Life Technologies|AB/ Invitrogen | 17504044 |
| BDNF | PeproTech, Inc. | 450-02-50UG |
| NT-3 | PeproTech, Inc. | 450-03-50UG), |
| Laminin | Life Technologies|AB/ Invitrogen | 23017015 |
| RIPA buffer | CST | 9806 |
| PI/PS | CST | 5872 |
| BCA assay | Thermo Scientific | 23225 |
| 4× buffer | BIO-RAD | 1610747 |
| 2-Mercaptoethanol (BME) | BIO-RAD | 1610710 |
| Criterion TGX Precast Gels 7.5% | BIO-RAD | 5671024 |
| Criterion TGX Precast Gels 4–15% | BIO-RAD | 5671084 |
| nitrocellulose membranes | BIO-RAD | 1704272 |
| NativePAGE Sample Buffer | Invitrogen | BN2003 |
| NativePAGE Bis-Tris Mini Protein Gel 3-12% | Invitrogen | BN-1001BOX |
| NativePAGE Running Buffer | Invitrogen | BN2001 |
| Sodium Cholate | Sigma-Aldrich | C6445-10g |
| n-Heptyl-β-D-thioglucoside | Sigma-Aldrich | H3264 |
| NheI-HF | New England Biolabs | R3131 |
| XhoI | New England Biolabs | R0146 |

| Reagent/resource | Reference or source | Identifier or catalog number |
|---|---|---|
| BsmBI-v2 | New England Biolabs | R0739 |
| XbaI | New England Biolabs | R0145 |
| SfiI | New England Biolabs | R0123 |
| EcoRI-HF | New England Biolabs | R3101 |
| Cell Meter™ NIR Mitochondrion Membrane Potential Assay Kit (Mitolite) | AAT Bioquest | 22802 |
| Methotrexate | Sigma-Aldrich | M8407 |
| HCS LIPIDTOX DEEP RED NEUTRAL | Life Technologies AB/ Invitrogen | H34477 |
| Tetramethylrhodamine, Ethyl Ester, Perchlorate (TMRE) | ThermoFisher Scientific | T669 |
| Bovine serum albumin | Sigma-Aldrich | A7906 |
| Saponin | Sigma-Aldrich | S7900 |
| HQ Silver enhancement kit | Nanoprobes | 2012 |
| uranyl acetate | EMS | 22400 |
| HT Supplement | Gibco | 11067030 |
| **Software** | | |
| Adobe Illustrator | https://www.adobe.com/products/illustrator.html | version 2023 |
| Adobe Photoshop | https://www.adobe.com/products/photoshop.html | version 2023 |
| ImageJ/FIJI | NIH | https://imagej.nih.gov/ij/ |
| TrakEM2 Fiji Plugin | NIH | https://imagej.net/plugins/trakem2/ |
| Microsoft Excel | Microsoft Corporation | N/A |
| Graphpad Prism 10 | Graphpad | Version 10 |
| CellStreamAnalysis | Cytek | Version 1.5.17 |
| Image Studio Lite | LiCor | Version 5.2 |
| **Other** | | |
| Lonza 4D-Nucleofector Core Unit | Lonza | AAF-1003B |
| 4D-Nucleofector X Unit | Lonza | AAF-1003X |
| Cytek Amnis CellStream Flow Cytometer | Cytek | CS-100496 |
| Ultramicrotome | Leica Microsystems, Wetzlar, Germany | EM UC7 |
| Diamond knife | DiATOME, Hatfield, PA, USA | Ultra 45 |
| Transmission Electron Microscope, 120KV | JEOL USA, Inc., Peabody, MA, USA | JEOL1400 Flash |
| CMOS detector, 29 Mpix (for electron microscope) | Advanced Microscopy Techniques, Danvers, MA, USA | Biosprint29 |
| Laser Scanning Confocal Microscope | Olympus | FluoView4000 |

  

| Reagent/resource | Reference or source | Identifier or catalog number |
|---|---|---|
| Microscope Digital Camera, Model DP23 | Evident Scientific, Inc. | DP23M-CU-1-2 |
| ACE900 Freeze Fracture System | Leica Microsystems, Wetzlar, Germany | EM ACE900 |
| Cell chamber | ThermoFisher Scientific | A7816 |
| Dissecting microscope | Zeiss | Stemi508 |
| Tissue culture microscope | Olympus | CKX53 |
| QIAprep Spin Miniprep Kit | Qiagen | 27106 |
| NucleoSpin® Blood Genomic DNA from blood kit | Macherey-Nagel™, | 740954.100 |
| LI-COR Imager | LI-COR | Odyssey CLx |
| Confocal laser scanning microscope | Olympus | FLUOVIEW FV3000 |

## Cell culture

All HeLa and HEK293 cells were grown in high-glucose DMEM with pyruvate (Gibco, 11995-065) supplemented with 10% fetal bovine serum (R&D Systems, S11550) and 1% Penn/Strep (Gibco, 15140-122) and maintained in an incubator at 37 °C and 5% $CO_2$, except where indicated. For experiments to test alternative carbon sources (galactose, pyruvate, low glucose, and no glucose) on PINK1 stabilization, DMEM without glucose and pyruvate but with L-glutamine (Gibco, 11966025) was supplemented with the indicated carbon source. The following lines were used throughout the manuscript to generate new cell lines described below. HeLa dCas9-BFP-KRAB (Hsu et al, 2025) (gift of the Dr. Richard Youle Lab), HeLa PINK1 KO PINK1-eYFP Clone #21 (Sekine et al, 2019) (from the Dr. Shiori Sekine Lab), WT HeLa (ATCC), WT HEK293 (ATCC). Cell lines were tested for mycoplasma, and the original lines were authenticated from ATCC.

## Generation of cell lines

dCas9-BFP-ZIM3 was introduced to cell lines via the piggybac system—iC30 PB-zim3-mycNLS-BFP-hygro-ucoe (addgene, 236743) and K13-EF1a-transposase. Cells were transfected with the transposase and the dCas9-BFP-ZIM3 plasmid with FuGENE Transfection Reagent (HEK293 cells—FuGENE 6 Promega, E2691 and HeLa cells— FuGENE HD Promega, E2311) following the manufacturer's instructions. Briefly, for HeLa cells a ratio of 3:1 FuGENE HD was used (6 µL:2 µg) and for DNA 1:2 ratio transposase:dCas9. For HEK293 cells, a ratio of 4:1 FuGENE 6 was used (8 µL:2 µg) and for DNA 1:2 ratio transposase:dCas9. Once cells were expressing BFP the cells were selected with hygromycin 200 µg/mL (MP Bio, 194170) for 72 h. Finally, cells were sorted via FACS for a clonal population, which was expanded and used for experiments. HeLa dCas9 BFP ZIM3 was added to WT HeLa cells, HeLa PINK1 KO PINK1 EYFP (Clone #21 from Dr. Shiori Sekine Lab), and WT HEK293 cells.

To generate stable lines, cells were transduced with the indicated virus and 8 µg/mL polybrene. Following transduction, cells were sorted via FACS to obtain a homogeneous population. Retroviral or

lentiviral vectors were used encoding mCherry-Parkin, YFP-Parkin, MTS(Cox8a)-mCherry, and/or mt-Keima.

Endogenously tagged cell lines were made by nucleofection according to the manufacturer's instructions. MFN2 was endogenously tagged in HeLa dCas9-BFP-KRAB and HEK293 dCas9-BFP-ZIM3 cell lines (MFN2 Halo donor plasmid). TOMM70 was endogenously tagged in the HeLa PINK1 KO PINK1-eYFP dCas9-BFP-ZIM3 cells (TOMM70 FKBP12 halo donor plasmid). Briefly, guides targeting the protein of interest and donor plasmids (that included the Halo tag sequence as well as 1 kb homology arms) were added together with HiFi Cas9 Nuclease V3 (IDT, 1081061), along with the nucleofection reagent. For HeLa cells, the SE Cell Line 4D-Nucleofector X Kit L (Lonza, V4XC-1012) and for HEK293 cells, the SF Cell Line 4D-Nucleofector X Kit L (Lonza, V4XC-2024). The mix was added to cells and nucleofected via the Lonza 4D-Nucleofector Core Unit (Lonza, AAF-1003B) and 4D-Nucleofector X Unit (Lonza, AAF-1003X) nucleofector. Program for HeLa cells CN-114 and for HEK293 cells CM-130. The cells were then plated in media containing HDR enhancer (IDT, 1081073 or IDT, 10007921). The next day, media was replaced with media not containing the HDR enhancer. Cells were sorted via FACS and expanded to use for further experiments (see Reagents and Tools Table for guide sequences).

Nucleofection was also used to generate cells for the KO pool experiments. Briefly, the same protocol was completed as detailed above—Cas9 protein and sgRNA were combined to form a ribonucleoprotein complex in vitro which was electroporated into the cells—but there was no donor plasmid added or HDR enhancer. Cells were analyzed at least 7 days after electroporation. (See material list for guide sequences). For the TOMM40 KO/TIMM23 KD combination experiments cells were nucleofected with TOMM40 KO pools as explained above, the following day the cells were transduced with CTRL or TIMM23 sgRNA following the same sgRNA transduction protocol explained above. The cells were fixed in 4% PFA at 8 days post KO/7 days post KD.

## Drug treatments and reagents

HeLa or HEK293 cells were treated with the following drugs/concentrations unless otherwise indicated oligomycin 10 µg/mL (Sigma-Aldrich, Inc 75351-5MG), antimycin A 4 µg/mL or 8 µg/mL (Sigma-Aldrich, Inc, A8674), 2-deoxy-D-glucose (2-DG) 10 mM (Sigma-Aldrich, D8375-5G), carbonyl cyanide 3-chlorophenylhydrazone (CCCP) 10 µM (Sigma, C2759-100MG), MG132 50 µM (Sigma, SML1135), cycloheximide 50 µg/mL (Sigma-Aldrich, Inc, C7698-5G), heptelidic acid 10 µM (Cayman chem, 14079), rotenone 1 µM (Calbiochem, 557368-1GM), KL11743 10 µM (Millipore Sigma, SML3458-25MG).

Day 10, i³Neurons were treated with CCCP 20 µM where indicated.

When appropriate, Janelia Fluor 646 HaloTag Ligand (Promega Corporation, GA1121) was added to the cells at 75 nM (based on titration experiments) for 25 min before analyzing.

## Lentivirus production

Lentivirus was produced in Lenti-X cells. FuGENE 6 Transfection Reagent (Promega, E2691) following the manufacturer's instructions was used to generate individual viruses. Briefly, Lenti-X cells were

transfected using a ratio of 3:1 FuGENE 6:DNA (6 µL:2 µg), the day after cell seeding. For a six-well plate, the cells were transfected with packaging plasmids psPAX2 (750 ng) and pMD2.G (250 ng) as well as the appropriate lentiviral plasmid (1 µg). Media was removed the following day and replaced with 1 mL. In all, 72–96 h post transfection media was collected and centrifuged at $3400 \times g$, 10 min, 4 °C. Being careful to not disturb the pellet, the supernatant was moved to a new tube, aliquoted, and frozen at −80 °C.

To make virus of the dual-sgRNA library from the Weissman Lab (addgene, 187246) Lipofectamine 3000 (Invitrogen, L3000001) was used following the manufacturer's instructions (Replogle et al, 2022). Briefly, cells were plated on poly-L-ornithine (PLO) (Sigma, P3655) coated 15 cm plates. The next day, cells were transfected with packaging plasmids (psPAX2 (13.3 µg), pMD2.G (4.5 µg), and pAdvantage (1.8 µg)) as well as with the lentiviral dual-sgRNA plasmid (19.5 µg), 60 µL of Lipofectamine 3000 reagent and 80 µL P3000 enhancer reagent. Full media change was completed the next day, and the media was collected 96 h post transfection. Virus was concentrated with Lenti-X Concentrator (Takara Bio, 631231) following the manufacturer's instructions. Briefly, the media was collected and centrifuged at $3400 \times g$, 10 min, 4 °C. Being careful to not disturb the pellet, the supernatant was moved to a new tube and Lenti-X Concentrator was added at a 1:3 ratio. The media/concentrator mix was placed at 4 °C for 24 - 48 h. Next, the solution was pelleted at $1500 \times g$, 45 min, 4 °C. The supernatant was discarded, and the pellet was resuspended in PBS at 1/10 of the original volume of media collected from the plate. Virus was aliquoted and frozen at −80 °C.

## siRNA transfections

The indicated cell lines were transiently transfected utilizing Lipofectamine RNAiMax Reagent (Life Technologies, 13778) per the manufacturer's instructions with 20 nM final concentration of siRNA. siRNA used –ON-TARGETplus SMARTPool Human PINK1 (Dharmacon, L-004030-00-0005) and siGENOME Non-Targeting siRNA #1 (Dharmacon, D-001210-01-05).

## Flow cytometry

All flow cytometry data were acquired on a Cytek Amnis CellStream Flow Cytometer Four-Laser System with 405 nm, 488 nm, 561 nm, and 642 nm and AutoSampler (Cytek, CS-100496). Cells were assessed live, except for one experiment to demonstrate retention of MFN2-Halo signal in fixed samples. For that experiment, HeLa$^{MFN2\text{-}Halo + mCh\text{-}Parkin}$ cells were incubated in fixation buffer (BD Biosciences, 554655) for 10 min followed by PBS washes. The cells were immediately analyzed by flow cytometry or stored at 4 °C in PBS for 1 day and then analyzed.

For all flow cytometry experiments using dual guides, a population gated for single cells was further gated for cells expressing the guide (TagBFP2 + ) and cells not expressing the guide (TagBFP2−).

For analysis of MFN2-Halo degradation among the PINK1–Parkin activators and facilitators, MFN2-Halo in the mCh-Parkin-positive (Parkin + ) population was compared to the mCh-Parkin-negative (Parkin-) population within each sample to determine Parkin-specific changes to MFN2-Halo intensity

following each knockdown. Specifically, the raw intensity of each population, Parkin+ and Parkin−, was first normalized by dividing by the average raw intensity of CTRL guide samples, giving MFN2-Halo normalized intensity values (MFN2normint) (except in Fig. 1E where cells were not gated on mCh-Parkin). The percent MFN2-Halo degradation was then calculated as follows:

$$MFN2 \text{ degradation } (\%) = 100 * ((MFN2normint^{Parkin+} - MFN2normint^{Parkin-})/MFN2normint^{Parkin+}).$$

For analysis of PINK1-YFP, the raw YFP intensity for guide-positive cells was normalized by dividing by the average of the CTRL replicates in the sample. Similarly, for analysis of MTS-mCh, the raw mCherry intensity for guide-positive cells was normalized by dividing by the average of the CTRL replicates in the sample. For experiments using TMRE to measure MMP, cells were incubated with TMRE 20 nM for 15–20 min prior to trypsinization for cell sorting. The raw TMRE intensity was normalized to the untreated condition.

Mitophagy was measured with the mt-Keima reporter by plotting the ex. 488–em. 611/31 channel (representing neutral mKeima) vs. ex. 561–em. 611/31 (representing acidic mKeima). To compensate for bleed-through from the YFP-Parkin channel, 10% compensation was applied for the ex. 488–em. 528/46 channel into the ex. 488–em. 611/31 channel. A triangular gate was drawn to capture events that shifted with mitophagy into a highly acidic/neutral population. The percent of cells in the "mitophagy" gate was reported.

For experiments measuring both PINK1-YFP and MMP, the cells were incubated for 15–20 min with MitoLite NIR following the manufacturer's instructions prior to sorting. We found the raw intensity of the MitoLite NIR staining from well-to-well was sensitive to cell number. To control for well-to-well variability, we normalized the MitoLite NIR signal as follows: (1) the raw values for the BFP+ population was divided by the raw value for the BFP − population to obtain within-sample normalized values, and then (2) the normalized values for each guide were divided by the average of the normalized values for the CTRL guide samples in the same experiment.

For 2D kernel density plots comparing PINK1-YFP and MMP (measured with MitoLite NIR), analysis was performed using a custom Python script and the Pandas and NumPy libraries. To control for variability in MitoLite NIR staining from sample to sample, the average MitoLite NIR intensity for events in the PINK1-YFP low/BFP low population for each sample was calculated. These represented non-transduced cells within each sample ("MMP guide negative"). The ratio of the "MMP guide negative" MitoLite NIR intensity of each sample was divided by that of the CTRL guide sample to give a "normalization factor" for each sample. The MitoLite NIR intensity for events in each sample was divided by this normalized factor. PINK1-YFP intensity values were not modified. The Seaborn library was used to plot a 2D kernel density plot for each sample. Events in each of the four quadrants were calculated using a Python script using the same cutoff values for each sample. Stacked bar graphs of these data were generated in Graphpad Prism 10.

## SUNset assay

Cells were treated with 10 µM Puromycin (Invivogen, ant-pr-1) for 10 min before immediately lysing the cells. Lysates were run via western blot and probed with puromycin antibody (see "Immunoblotting section").

## iPSC culture and neuronal differentiation

i11W-mNC iPSC were used for all neuronal differentiations (these cells are the WTC11 iPSCs expressing doxycycline-inducible mouse neurogenin-2 (NGN2) and CAG-dCas9-BFP-KRAB). Cells were maintained and differentiated to i³Neurons according to published protocols (Fernandopulle et al, 2018). Briefly, iPSCs were plated on Matrigel (CORNING, 354277) coated plates and cultured in Essential 8 Medium + Supplement (Gibco, A1517001). Following a thaw or a passage 10 µM Y-27632 ROCK inhibitor (Tocris Bioscience, 1254) was added to the media to promote survival. Passaging was done with 0.5 mM EDTA (Invitrogen, 15575-038) or when single cells were needed with Accutase (Gibco, A1110501). For differentiation, cells were dissociated via Accutase and plated in induction medium consisting of KnockOut DMEM/F12 (Gibco, 12660012), N2 Supplement 100X (Gibco, 17502048), non-essential amino acids 100X (Gibco, 11140050), L-glutamine 100X (Gibco, 25030081), Y-27632 ROCK inhibitor (Tocris Bioscience, 1254), and doxycycline (Sigma, B9285). Complete media changes were done for the next 2 days, on day 3 cells were plated to poly-L-ornithine (PLO) (Sigma, P3655) coated plates. Cells were plated in cortical neuron culture media consisting of BrainPhys neuronal medium (STEMCELL Technologies, 05790), B27 supplement (Life Technologies|AB / Invitrogen, 17504044), BDNF (PeproTech, Inc., 450-02-50UG), NT-3 (PeproTech, Inc., 450-03-50UG), Laminin (Life Technologies|AB/Invitrogen, 23017015), and doxycycline. Half media changes were performed on the cells every other day until the collection day.

## Genome-wide CRISPRi screen

The genome-wide CRISPRi screens were all performed in duplicate. Cells were transduced with the dual-sgRNA library from the Weissman Lab (Addgene, 187246) at an MOI of 0.3 based on the virus titer (Replogle et al, 2022). Puromycin selection was added to the cells (2.5 µg/mL) 48 h post transduction for 2 days followed by 3 days of no puromycin media. For HeLa cells, AO was added for 4 h 40 min, and for HEK293 cells CCCP treatment was added overnight. Each condition was performed in duplicate. The day of the screen sort, Halo Ligand 646 (Promega Corporation, GA1121) was added to the cells for 25 min pre-collection at 75 nM. Cells were collected and resuspended in cell sorting media (145 mM NaCl, 5 mM KCl, 1.8 mM CaCl$_2$, 0.8 mM MgCl$_2$, 10 mM Hepes, 10 mM glucose, 0.1% BSA). Finally, the cells were sorted on a MoFlo Astrios flow cytometer (Beckman Coulter, B52102) using Summit software. Two populations were collected, representing the top 30% and bottom 30% MFN2-Halo expressing cells (6 million cells were collected per bin). After sorting, cells were spun down at 400 g for 5 min and pellets were frozen at -80 °C. Genomic DNA was extracted using the NucleoSpin® Blood Genomic DNA from blood kit (Macherey-Nagel™, 740954.100) according to the manufacturer's instructions. Briefly, cell pellets were thawed at room temperature, washed, and resuspended in 2 mL PBS with 150 µL Proteinase K. In all, 2 mL of BQ1 was added to each sample and vortexed vigorously, followed by a 15 min incubation at 56 °C. Samples were cooled at room temperature for 1 h. In total, 2 mL of 100% ethanol was added to each sample and immediately inverted to mix. Samples were loaded onto the columns provided in the kit in 2 stages (3 mL for the first spin and then the rest of the sample for the second spin). Samples were spun at $4500 \times g$ for 3 min and 5 min, respectively. Membranes were washed twice with 2 mL BQ2 at $4500 \times g$ for 2 min and 10 min, respectively. Finally, DNA was eluted with pre-warmed Buffer BE. In total, 100 µL of Buffer BE was added to the membrane, incubated for 2 min, and spun at $4500 \times g$ for 2 min. This was repeated with another 100 µL of Buffer BE. PCR was done to amplify the region of interest/add barcodes/Illumina adapters, as described (Replogle et al, 2022). PCR reactions were made up of 10 µg of genomic DNA, 100 µM of each primer (Dataset EV8), and NEBNext Ultra II Q5 Master Mix (New England Biolabs, M0544). The PCR conditions were as follows, 98 °C for 30 s, 19 cycles of 98 °C for 10 s and 66 °C for 75 s, then 72 °C for 5 min, and 4 °C hold. Finally, the samples were purified with 0.5×–0.65× SPRI beads (Beckman Coulter, B23318) cleanup. 150 µL of beads was added to 300 µL of PCR reaction, mixed and incubated at room temperature for 10 min. Beads were separated from samples on a magnetic stand for 5 min, and the supernatant was moved to a new tube. Next, 45-µL beads were added to the supernatant, mixed, and incubated for 10 min at room temperature. Tubes were placed on a magnetic stand for 5 min, and the supernatant was disposed of. Beads were washed for 2 min with 1 mL fresh 80% ethanol twice. Finally, cells were air-dried for 3 min and eluted in 30 µL of elution buffer (Replogle et al, 2022). PCR size and quality were verified by TapeStation. Libraries were pooled and sequenced on a NovaSeq 6000 V1.5 with a 15% PhiX spike, SP 100 cycle kit, targeted seq, PE-50-10-10-50, with Illumina and custom primers. Primers: Read 1- gtgtgtttttgagactataagtatcccttggagaac-caccttgttgG Read 2-tgctatgctgtttccagcttagctcttaaac Index i5 Primer-acagttagggtgagtttccttttgtgct. Samples were demultiplexed using i5 index. FASTQ files were analyzed using MAGeCK-Vispr robust ranked algorithm pipeline to compare high 30% vs. low 30% MFN2-Halo fluorescence (Li et al, 2015). The log twofold change values and uncorrected two-sided *P* values for individual guides in the sgRNA summary output file were used for further analysis, using a custom Python script. Guides with 0 read counts in one of the replicates were filtered out. Bonferroni-corrected *P* values were corrected by dividing the uncorrected *P* values by the total number of guides. Gene scores were calculated as the product of the −log 10 corrected *P* values and the log2 fold change. Genes were annotated as mitochondrial based on their inclusion in MitoCarta3.0 (Rath et al, 2021). The following gene names were modified prior to annotation with MitoCarta3.0: "TIMM23B" was changed to "TIMM23"; "HSPE1-MOB4" was changed to "HSPE1"; "PARK2" was changed to "PRKN". "Parkin activators" were significant and had a log2 fold change of < −1 in not treated HeLa cells with mCherry-Parkin. In addition, the difference in log2 fold change between the HeLa cells with mCherry-Parkin and without Parkin was < −1.25. "Parkin facilitators" were significant and had a log2 fold change >1 in HeLa cells with mCh-Parkin that were treated with oligomycin and antimycin. In addition, the log2 fold change was at least greater than 1 compared to log2 fold change from the Parkin mCh-Parkin untreated samples and the log2 fold change from HeLa no Parkin cells treated with oligomycin and antimycin.

"MFN2 downregulators" were significant and had a log2 fold change less than −1 in both the HEK293 untreated and HeLa no Parkin untreated comparisons. Additionally, they were not "Parkin activators". "MFN2 upregulators" were significant and had a log2 fold change greater than 1 in both the HEK293 untreated and HeLa no Parkin untreated comparisons. In addition, they were not "Parkin facilitators".

## Guide transductions

pJR103 plasmid (Addgene plasmid # 187242) was digested with XbaI and XhoI and ligated with "empty vector fragment" via NEBulider (according to the manufacturer's instructions, New England Biolabs, E2621). Briefly, 50 ng vector was ligated with "empty vector fragment" at a 2:1 insert:vector ratio. This new plasmid, lenti_DualsgRNAEmptyVector_Puro_T2A_BFP2, was then digested with BsmBI, and NEBuilder reactions (same amount/ratio as above) were done with the indicated sgRNA gene block fragments (Dataset EV9). Ligations were transformed into ONE SHOT STBL3 COMP E COLI (Life Technologies, C737303) according to the manufacturer's instructions. Individual colonies were picked and grown up in LB broth, mini-prepped with the QIAprep Spin Miniprep Kit (Qiagen, 27106), and sequenced (sequencing primer-ggcttaatgtgcgataaaagacaga). Lentivirus was generated for each sgRNA (see above Lentivirus production section). On the day of transduction, sgRNA was added to cells with 8 µg/mL polybrene (Millipore Sigma, TR-1003-G). Forty-eight hours post transduction, the media was changed to include puromycin 2.5 µg/mL for 2 days, cells were then allowed to recover in non-puromycin media and were treated and collected on day 7 post transduction (unless otherwise indicated).

## Immunoblotting

Sample prep—Cells were collected in PBS (Gibco, 10010-031) and lysed in RIPA buffer (CST, 9806) containing PI/PS (CST, 5872), sonicated (QSonica Q800R3, Q800R3-110), and a BCA assay (Thermo Scientific, 23225) was performed. Lysates were diluted in 4X buffer (BIO-RAD, 1610747) with 2-Mercaptoethanol (BME) (BIO-RAD, 1610710) and run on Criterion TGX Precast Gels 7.5% (Bio-Rad, 5671024) or 4–15% (Bio-Rad, 5671084). Gels were transferred via the Bio-Rad TransBlot Turbo Transfer System (BIO-RAD, 1704150) onto nitrocellulose membranes (BIO-RAD, 1704272). Blocked in 5% nonfat milk (Dot Scientific Inc., DSM17200) and incubated with primary antibody overnight at 4 °C. The following day, appropriate secondary antibodies were added before imaging the blots on a LI-COR Odyssey CLx Imager. Any western blot quantification was done using Image Studio Lite Version 5.2.

In samples where we probed for endogenous PINK1, which is present in cells at a low protein copy number, we followed a different procedure. Cells were directly lysed in 1× sample buffer containing BME and PI/PS. Samples were run on a 7.5% gel, transferred using the Biorad system, and blocked in Licor PBS blocking buffer (LI-COR, 927-70001). Membranes were incubated in PINK1 primary antibody for 2 days at 4 °C before the secondary antibody was added, and the membrane was imaged.

See the Reagents and Tools Table for primary and secondary antibody information.

## Confocal microscopy

Most confocal microscopy was performed using an Olympus FLUOVIEW FV3000 microscope in Galvano mode with a PlanApo N 60×1.42 oil objective. The exception was for images appearing in panels Fig. 5C,D. For these, a Zeiss LSM 880 AiryScan Confocal Microscope was used. For live cell experiments, an Okolab stage heater and $CO_2$/air mixer kept samples at 37 °C in a 5% $CO_2$ environment. Live imaging was typically performed over the course of 1–2 h. All confocal imaging of HeLa cells was performed in uncoated ibidi µ-Slide 8 Well chambered coverslips.

To measure PINK1-YFP intensity on mitochondrial with high and low MMP within the same cell, live HeLa$^{PINK1-YFP+MTS-mCh}$ cells were loaded with MitoLite NIR following the manufacturer's instructions for 15–20 min prior to imaging. Cells were imaged directly without additional washing. A midplane confocal image was obtained of each field and further processed in FIJI (Version 1.54 m), using a custom FIJI macro script. Individual cells in the field were manually enclosed in an ROI using the polygon tool. The image was then duplicated and cropped around the cell of interest. The image was again duplicated, the background color was set to (0 0, 0), the images were downgraded to 8 bits, and the channels were split. MTS-mCh and MitoLite NIR masks were generated from duplicated images of the corresponding channels using the setThreshold (80, 255, "raw") command followed by the run("Convert to Mask") command. A mitochondrial mask was generated from the union of the MTS-mCh and MitoLite NIR masks. A mask of high MMP mitochondria was generated by subtracting the MitoLite NIR mask from the mitochondrial mask. A mask of low MMP mitochondria was then generated by subtracting the high MMP mitochondria mask from the mitochondrial mask. Two groups of ROIs were then made from the high and low MMP mitochondria masks using the run ("Create Selection") command to measure the intensity of each channel for each ROI. Measurements were made using the roiManager ("multi-measure measure_all") command. The results table was exported as a .csv file. A custom Python script was used to collate data from the results .csv files for each cell. A similar process was used to measure mitochondrial PINK1-YFP from MitoLite NIR high cells following CTRL or TIMM23 KD, except that the MitoLite NIR mask was used to generate mito measurements, and a non-mito mask was created by subtracting the MitoLite NIR mask from a cell mask defined by manual cell selection. For measurements of co-localization, cells were individually selected using the polygon tool, duplicated, and cropped as above. The "co-localization test" plug-in was then run, which measures the Pearson correlation coefficient for both the image and 20 control images produced by random transposition of the two channels. The average Pearson coefficient from the random images was subtracted from the observed Pearson coefficient for each cell to give "R over background". The imager was not blinded to the conditions, but automated image analysis was performed whenever possible.

Immunocytochemistry was performed following fixation in 4% PFA in 1× PBS (diluted from paraformaldehyde 16% Aqueous Solution EM Grade, cat# 15700) for 10 min at room temperature. Cells were permeabilized by incubation with 0.25% Triton X100 for 10 min at room temperature. Cells were blocked for at least 15 min in 5% BSA in 1× PBS. The primary antibody was diluted in 5% BSA in 1× PBS at a 1:500 ratio and incubated at room temperature for

    

1–2 h or overnight at 4 °C. Cells were washed three times in 1× PBS. The secondary antibody (1:500) in 5% BSA 1× PBS was added, and cells were incubated for 45 min at room temperature. The cells were washed three times in 1× PBS and imaged in 1× PBS.

## Neuronal staining

Neurons were cultured on eight-well glass-bottom chamber slides (ibidi USA, Inc., 80827-90) and fixed in 4% PFA at d9, followed by permeabilization and block in 0.1% saponin (Sigma-Aldrich, S7900-100G) and 3% donkey serum (GeneTex, Inc., GTX73205) in PBS at room temperature for 30 min. Cells were incubated in primary antibodies prepped in permeabilization and block buffer overnight, rocking gently at 4 °C. Primary antibody was washed off with PBS, and secondary antibodies were diluted (1:500) in permeabilization and block buffer and added to cells for 1 h rocking at RT. Secondary antibodies were washed off with PBS and DAPI (Invitrogen, 62248) was added 1:5000 in PBS for 5 min. See the Reagents and Tools Table for primary and secondary antibody information. Cells were imaged on an Olympus FLUOVIEW FV3000 confocal laser scanning microscope.

## Clear native page and NAMOS assay

HeLa$^{PINK1-YFP}$ cells were cultured with the indicated sgRNA for at least 7 days before Clear Native Page sample collection. Samples were lysed and run according to a previously published protocol (Okatsu et al, 2013). Briefly, cells were lysed in cold NativePAGE Sample Buffer (Invitrogen, BN2003) with 1% digitonin and 1X protease/phosphatase Inhibitor (CST, 5872S) and were placed on a rocker for 15 min at 4 °C. Samples were mixed by pipetting ten times and then spun at $20,000 \times g$, 4 °C, 30 min. Following BCA assay, 35 µg of cell lysate was incubated with 1 µg of the indicated antibodies at room temperature for 1 h (shaking). The whole volume of sample was loaded onto a NativePAGE Bis-Tris Mini Protein Gel 3-12% (Invitrogen, BN-1001BOX) and was run in NativePAGE Running Buffer (Invitrogen, BN2001) supplemented with 0.05% Sodium Cholate (Sigma, C6445-10g) and 0.01% $n$-Heptyl-β-D-thioglucoside (Sigma, H3264) in the cold room (4 °C). Gel was immediately imaged after running on an Amersham Typhoon 5 using 488 nm cy2 laser-filter set combination. Total protein was measured via SimplyBlue SafeStain according to the manufacturer's protocol (ThermoFisher, LC6065) and imaged on a Cytiva Amersham ImageQuant 800.

## Antibodies

TOMM22 (Proteintech, 66562-1-Ig), TOMM40 (Proteintech, 18409-1-AP), TOMM20 (Santa Cruz, sc-17764), and CHCHD2 (Proteintech, 66302-1-Ig).

## Preparations that were submitted for proteomics

### Mitochondria isolation and insoluble/soluble for proteomics
Mitochondria were isolated from cells with CTRL or DNAJA3 sgRNA (four replicates/per condition) according to a previously published protocol (Lazarou et al, 2012; Huang et al, 2018). Cells were homogenized with 40 strokes. Separation of insoluble and soluble fractions was completed according to a

published protocol with some modifications (Shin et al, 2021). Briefly, extracted mitochondria were resuspended Triton X-100 buffer (20 mM Tris-HCl, pH 7.4, 150 mM NaCl, 2 mM EDTA, 1% Triton X-100), PI/PS (CST, 5872), and incubated for 30 min on ice (pipetting every 10 min). Cell lysates were centrifuged at $20,817 \times g$ for 10 min at 4 °C. The supernatant (soluble fraction) was moved to a new tube, an aliquot was taken and added to 10% SDS to get a final solution of 5% SDS. The insoluble pellet was resuspended in Triton buffer, PI/PS, 5% SDS, and agitated into solution. Samples were submitted for proteomics.

### Whole-cell lysate preparation
Cells were pelleted in PBS at 4 °C, $400 \times g$, 5 min, and resuspended in RIPA buffer (CST, 9806) and PI/PS (CST, 5872). The lysates were sonicated (QSonica Q800R3, Q800R3-110) for 10 min at 4 °C followed by centrifugation at top speed ($21,130 \times g$) 4 °C, 5 min. BCA assay (Thermo Scientific, 23225) was performed on the supernatant. In all, 40 µg of lysate with 5% SDS was submitted for proteomics.

### Immunoprecipitation of PINK1 YFP
Immunoprecipitation of HeLa$^{PINK1-YFP}$ cells was performed using GFP-Trap Magnetic Particles M-270 (Chromotek, Gtd) beads according to the manufacturer's instructions. Samples were eluted from beads in 30 µL RIPA buffer (CST, 9806) + 5% SDS (Invitrogen, 15553-035) + PI/PS (CST, 5872), and the bound fraction was submitted for proteomics.

## Mass spectrometry-based proteomics

Liquid chromatography-tandem mass spectrometry (LC-MS/MS) data acquisition was performed on an Orbitrap Ascend mass spectrometer (ThermoFisher Scientific) coupled with a 3000 Ultimate high-pressure liquid chromatography instrument (ThermoFisher Scientific) unless noted otherwise. Peptides were separated on an ES902 column (ThermoFisher Scientific) with the mobile phase B (0.1% formic acid in acetonitrile) increasing from 3 to 18% over 61 min, then from 18% to 30% over 14 min. The LC MS/MS data were acquired in data-independent mode (DIA). Both MS1 and MS2 scans were performed in Orbitrap. For MS1 scan, the following parameters were used: mass range ($m/z$) = 380–985; resolution = 60 K; the automatic gain control (AGC) = 4e5; RF lens = 60%. The parameters for MS2 scan were: mass range ($m/z$) = 145–1450; resolution = 15 K; isolation window ($m/z$) = 10; window overlap ($m/z$) = 1; AGC = 1e5; maximum ion injection = 40 ms. The cycle time was set at 3 s.

Spectronaut 19.5 software was used to analyze DIA data with the directDIA analysis method. Data were searched against Sprot Mouse database. Trypsin/P was set as a specific enzyme with 2 max missed cleavages. Carbamidomethyl on cysteine was a fixed modification. Acetyl (Protein N-terminal and Oxidation on methionine were set as variable modifications. FDRs for PSM, peptide, and protein levels were 1%. MS2 abundances were used for protein quantitation. Cross-run normalization was performed. Unpaired $t$ test was used for the hypothesis test.

Note: There were datasets acquired with a data-dependent method (DDA) as indicated in Dataset EV10. The general method used for those samples was described previously (Shammas et al, 2022) with some main differences listed below.

(1) Proteome Discoverer 3.1 (ThermoFisher Scientific) was used for database search and quantitation analysis of DDA datasets. (2) "WCL exog vs endog PINK1" and "mitochondria insoluble (I)/soluble (S) fractions" datasets were acquired on an LC-MS system where an Orbitrap Ascend mass spectrometer (ThermoFisher Scientific) was connected to a Vanquish NEO nano-HPLC (ThermoFisher Scientific).

After quantification, the proteomics data were further analyzed and visualized in Python, using custom scripts. Proteins were annotated for their mitochondrial localization and mitochondrial pathways using human MitoCarta3.0. Mitochondrial proteins with predicted strong presequences were identified using data from MTSviewer (https://mtsviewer.neurohub.ca/). Proteins were annotated as having a strong presequence if (1) they localized to mitochondria but not primarily the outer mitochondrial membrane, (2) their MTS1 score was >= 2, and (3) their MTS1 start position was <20 amino acids from the N-terminus. For all proteomics experiments except AP-MS experiments, two-sided Student's $t$ tests were performed using SciPy 1.14.0 Python library, and values were corrected for multiple comparisons across all gene groups/features by calculating a FDR with the Benjamini–Hochberg procedure, using the statsmodels 0.14.1 Python library. Proteins were annotated as significant if they had an FDR < 0.05 and an absolute $\log_2$ fold change of >1 for most experiments; the exception was experiments in (Fig. 7G) where an absolute $\log_2$ fold change of >0.5 was used instead. For the AP-MS experiment, statistics were performed in Perseus: missing data were first imputed using the "Replace missing values from normal distribution" function; a two-sided Student's $t$ test was then performed within Perseus using default settings; finally, an FDR was calculated using the permutation method. For all proteomics data, volcano plots were generated using the Seaborn Python library. All other graphs were generated in Graphpad Prism 10.

## Peroxidase staining and TEM of APEX-BFP expressing HeLa cells

HeLa[dCas9-BFP-ZIM3] cells were transduced with dual RNA guides for the top eight proteins identified as PINK1–Parkin activators and expressing ER-BFP2-APEX targeted to the ER lumen. Previously cloned sgRNA plasmids were cut with restriction enzymes EcoRI and NheI, followed by NEBuilder cloning with an APEX BFP gBlock (Dataset EV9). Cells transduced with sgRNAs were cultured on Aclar film coverslips (Electron Microscopy Sciences (EMS)) in 12-well plates. Coverslips were prepared by cutting Aclar into ~15 mm squares that fit into the wells of 12-well plates. Coverslips were rinsed in distilled water, then sterilized with 70% ethanol and transferred to a 12-well plate, where they were rinsed ten times with autoclaved ultrapure water and rinsed once with culture medium before plating of cells. Cells were fixed 7 days after transduction for 30 min at room temperature in fixative containing 2% paraformaldehyde, 2% glutaraldehyde, and 50 mM calcium chloride in 0.1 M sodium cacodylate buffer (EMS). Peroxidase staining was performed as previously described (Zhang et al, 2019), with some modifications. After fixation, coverslips were rinsed twice in 0.1 M sodium cacodylate buffer (cacodylate buffer), followed by a 10-min incubation in cacodylate buffer containing 50 mM glycine (Sigma-Aldrich) and a final rinse in cacodylate buffer. Coverslips were then incubated in 1–2 mL per well of cacodylate buffer containing 0.3 mg/mL of DAB (Sigma-Aldrich) at room temperature in the dark for 30 min. During DAB incubation, a 0.3% solution of hydrogen peroxide in cacodylate buffer was prepared by diluting 30% hydrogen peroxide (Sigma-Aldrich) by 1:100 in cacodylate buffer. The resulting 0.3% hydrogen peroxide in cacodylate buffer was then added at a 1:100 dilution per well to the DAB-containing cacodylate buffer in the 12-well plate, resulting in a final concentration per well of 0.003% hydrogen peroxide in 0.3 mg/mL DAB in cacodylate buffer, and the plate was returned to the dark. The intensity of the brown reaction product from the peroxidase reaction with DAB and peroxide was monitored by light microscopy and 10–12 min was determined to be the optimal development time (Fig. EV1). Coverslips were rinsed twice with cacodylate buffer and then post-fixed with 3% glutaraldehyde in cacodylate buffer. For EM processing, coverslips were rinsed twice for 10 min in cacodylate buffer, and then stained with reduced osmium containing 1% osmium tetroxide (EMS) and 1% potassium ferrocyanide (Sigma-Aldrich) on ice for 60 min. Next, coverslips were rinsed three times with cacodylate buffer and dehydrated through a graded series of 5-min ethanol rinses prior to infiltration in EmBED812 epoxy resin using the manufacturer's hard resin formulation (EMS). To embed cells, Aclar coverslips were laid cell-side-up on a backing piece of Aclar, and then 2–3 gelatine capsules (EMS) filled with resin were inverted on top of the coverslips and placed in a 60 °C oven for 2 days.

For sectioning, Aclar coverslips were peeled away from the resin, and en face ultrathin sections of embedded cells were cut to a thickness of 70 nm using an ultramicrotome (EM UC7, Leica Microsystems) with a diamond knife (DiATOME). Sections were picked up on 1 mm-slot copper grids with a thick formvar support film (EMS), and post-stained with 3% Reynold's lead citrate (EMS) for 1 min. Sections were viewed using a JEOL1400 Flash transmission electron microscope (JEOL USA, Inc.) operated at 120 KV. Images were recorded with a Biosprint29 CMOS detector (Advanced Microscopy Techniques). Images were viewed and adjusted for display using Adobe Photoshop Software version 2023 (Adobe). Linear adjustments to black and white points were made using the Levels tool to optimize contrast. Figures were prepared using Adobe Illustrator software version 2023 (Adobe).

## Assessment of TEM ultrastructure of mitochondria transduced with sgRNAs

Thin sections of cells expressing sgRNAs were searched, and cells with ER that was darkly stained with the APEX-DAB reaction product, indicating unequivocally high transduction levels, were recorded for analysis. At least ten cells from each transduction condition were analyzed. First, the ultrastructure of mitochondria in control cells was surveyed to establish the baseline appearance of mitochondria in control HeLa cells. Then, cells that were transduced with sgRNA to knock down mitochondrial proteins were reviewed and scored as exhibiting normal versus abnormal mitochondria. Then, the unusual features in mitochondrial cristae structure, matrix, and mitochondrial shape in cells deemed abnormal were recorded and are presented in Dataset EV2.

## Correlative light and TEM

Transduced HeLa cells were plated on either 15 mm Alcar film coverslips (EMS) or glass coverslips (18 mm or 25 mm diameter) (Fisher Scientific). For CLEM, prior to sterilization and cell plating, coverslips were shadowed with a 4-nm-thick layer of carbon deposited

using a vacuum evaporator system (Leica Microsystems). The coverslips were cleaned with water and ethanol and air-dried. Then, a 10-mm-wide finder grid (EMS) was placed at the center of a coverslip before loading it into the carbon evaporator system. The layer of carbon was deposited on the surface of the coverslip, and the presence of the finder grid created a numerical grid pattern that was visible by light microscopy. After 7 days of transduction, cells were fixed with 4% paraformaldehyde in 0.1 M sodium cacodylate buffer (cacodylate buffer) (EMS) and then either loaded into a cell chamber in cacodylate buffer (ThermoFisher Scientific) or rinsed and mounted on a glass slide (Fisher Scientific) in ProLong Gold Antifade mountant (ThermoFisher Scientific) for confocal imaging.

Confocal imaging was performed with either an Olympus FLUOVIEW FV3000 Microscope (Figs. 3D, 6I, and EV3; Appendix Fig S5A) or Zeiss LSM 880 AiryScan Confocal Microscope l (Fig. 5D) and images are shown as z-projections. Transmitted light images were taken to record the location of the cells of interest relative to the carbon shadowed grid. Following confocal imaging, coverslips in mounting medium were released from glass slides by soaking the slide in cacodylate buffer until the coverslip were free. Coverslips were rinsed in cacodylate buffer and then fixed in 2% glutaraldehyde (EMS).

EM processing was carried out as for APEX-BFP expressing HeLa cells, with the added step of staining with 1% uranyl acetate (EMS) in acetate buffer, pH 5.2 (EMS) overnight at 4 °C. Specifically, after reduced osmium staining, cells were rinsed with cacodylate buffer and then 50 mM acetate buffer pH 5.2 (EMS) before overnight incubation in uranyl acetate. The next day, cells were rinsed in acetate buffer before dehydration and embedding. To embed correlative samples, after infiltration with resin, a resin-filled gelatin capsule was inverted and placed directly over the location on the correlative grid pattern where the cell of interest was located. Glass coverslips were laid cell-side-up on a glass slide prior to placement of the inverted gelatine capsule, and Aclar coverslips were laid on a piece of Aclar film before placement of the gelatine capsule and polymerized in a 60 °C oven for 2 days. Glass coverslips were separated from the resin by heating the back of the glass slide with a flame until the resin block separated from the glass; Aclar coverslips were peeled away from the resin. The carbon grid transferred to the surface of the resin block was used to locate the cell of interest for thin sectioning. Serial sections were cut at 70–140 nm thickness and picked up on 1-mm slot grids (EMS), post stained with 3% Reynold's lead citrate (EMS) before viewing on the electron microscope as described above. Alignment of the confocal and EM images was achieved using the TrakEM2 plugin for Fiji (Schindelin et al, 2012).

## Pre-embedding silver-intensified immunogold labeling TEM

HeLa cells expressing PINK1-YFP and transduced with TOMM22 sgRNA for 7 days were plated on Aclar coverslips in a 12-well plate (as described above). Cells were fixed with 4% paraformaldehyde (EMS) in 0.1 M Sorensen's phosphate buffer (PB) (EMS) for 30 min at room temperature, then rinsed with PBS and incubated in PBS containing 50 mM glycine for 30 min. Cells were then blocked and permeabilized with PBS containing 2% bovine serum albumin (Sigma Cat. No. A7906) and 0.1% saponin (Sigma Cat. No. S7900) for 30 min and then incubated with primary antibody to GFP

(rabbit anti-GFP; Abcam Cat. No. ab6556) for one hour. Cells were rinsed twice with blocking solution for 30 min and then incubated with nanogold-conjugated Fab secondary antibody (Nanoprobes, Cat. No. 2004) for an hour, after which cells were rinsed twice with PBS and twice with PB and then fixed in 2% glutaraldehyde (EMS) in PB for at least an hour. To silver intensify the nanogold, cells were rinsed thoroughly with distilled water and then silver intensified using the HQ Silver enhancement kit (Nanoprobes, Cat. No. #2012) according to the manufacturer's instructions. After enhancement, cells were rinsed with distilled water and PB and postfixed with 0.2% osmium in PB for 30 min on ice, then rinsed with PB. Next, cells were rinsed with 50 mM acetate buffer (pH 5.2) and incubated for 30 min in 0.25% uranyl acetate (EMS) in acetate buffer. After rinsing in acetate buffer, cells were then dehydrated through a series of ethanol rinses and embedded in EmBED812 embedding medium. Ultrathin sections were prepared, and cells were imaged as described above.

## TOMM5 rescue

TOMM5 WT or TOMM5 1–39 (truncation after residue 39) were cloned into the doxycycline inducible pSBtet-RN, a gift from Eric Kowarz (Addgene plasmid # 60503) (Kowarz et al, 2015). Briefly, pSBtet-RN was digested with SfiI and ligated via NEBuilder with either HA-TOMM5 WT or HA-TOMM5 1–39 gene blocks (Dataset EV9). Ligations were transformed in One Shot TOP10 Chemically Competent E. coli (Invitrogen, C404003) according to the manufacturer's instructions. Single colonies were picked and grown up in LB broth, mini-prepped with the QIAprep Spin Miniprep Kit, and sequenced (sequencing primer-AGCTCGTTTAGTGAACCGTCA-GATC). These plasmids were transfected via FuGENE HD according to the manufacturer's instructions into HeLa PINK1 KO PINK-YFP dCas9-BFP-ZIM3 cells with the transposase (pCMV(CAT)T7-SB100), which was a gift from Zsuzsanna Izsvak (Addgene plasmid # 34879) (Mátés et al, 2009). Briefly, a ratio of 3:1 FuGENE HD was used (3 μL:1 μg) and for DNA a 1:20 ratio transposase:plasmid. Following FACS sorting to obtain a polyclonal population, the cells were transduced with CTRL or TOMM5 sgRNA following the above sgRNA transduction protocol. The cells were maintained in +/− 1 μg/mL doxycycline for the duration of the experiment. Cells were treated with vehicle or 10 μM CCCP for 4 h and run on the CellStream after at least 7 days post sgRNA transduction.

## Clogger experiments

### ATP5MG-mCherry-sfGFP

HeLa^MFN2-Halo cells +/− BFP-Parkin were transiently transfected with ATP5MG-mCherry-sfGFP (Krakowczyk et al, 2024). Approximately 22 h post transfection, cells were treated with 10 μM CCCP for 4 h (Halo646 ligand was added for the final 25 min of treatment). Cells were collected and run on CellStream gating on no GFP, Low GFP, and High GFP expressing cells to look at MFN2-Halo levels under different amounts of the clogger.

### IMMT-DHFR (IDF146-BFP)

HeLa^PINK1-YFP cells endogenously tagged with TOMM70-Halo were transduced with IMMT-DHFR (IDF146-BFP) (gift from Dr. Richard Youle) and sorted for IDF146-positive cells (Hsu et al, 2025). Cells were cultured in HT (hypoxanthine and thymidine) supplemented DMEM as described previously (Hsu et al, 2025).

Briefly, cells were treated with DMSO or Dox (1 µg/mL) and methotrexate (MTX) (200 µM) for 28 h (with +/− CCCP 20 µM being added for the final 4 h). Cells were collected and run on the CellStream to look at stabilization of PINK1-YFP in the presence or absence of the clogger.

## Statistical analysis

All statistics were calculated in Graphpad Prism 10, except as noted above for the proteomics experiments. When the variance was similar among the groups and the data were normally distributed, standard one-way ANOVA was performed when comparing differences between more than two groups, and an unpaired, two-tailed $t$ test was performed when comparing two groups. Where these assumptions were not true, multiple groups were compared using a Brown–Forsythe and Welch ANOVA was performed followed by Dunnett's T3 multiple comparisons test, and for comparison to two groups a Mann–Whitney test was performed. In general, sample sizes were at least three per condition where statistics were performed.

## Data availability

Next Generation Sequencing data: Gene Expression Omnibus GSE298056. Any additional information required to reanalyze the data reported in this paper is available from the lead contact upon request.

The source data of this paper are collected in the following database record: biostudies:S-SCDT-10_1038-S44318-025-00604-z.

## Materials availability

All constructs or cell lines generated in this study are available from the lead contact upon request and completion of a Material Transfer Agreement.

## Peer review information

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

## Acknowledgements

We thank Dr. Dragan Maric and the NINDS Flow Cytometry Facility for technical assistance with FACS, Dr. Carolyn Smith and the NINDS Light Microscopy Facility for technical assistance with confocal microscopy, and Dr. Yuesheng Li and the NHLBI DNA Sequencing and Genomics Core for technical assistance with next-generation sequencing. This work utilized the computational resources of the NIH HPC Biowulf cluster (https://hpc.nih.gov). We thank Dr. Kei Okatsu (Kyoto University) for technical advice on in-gel imaging of PINK1-YFP by CN-PAGE. We thank Dr. Richard Youle for his critical reading of the manuscript and insightful comments. This research was supported in part by the Intramural Research Program (1ZIANS003169) of the National Institutes of Health (NIH), National Institute of Neurological Disorders and Stroke (NINDS). The contributions of the NIH author(s) were made as part of their official duties as NIH federal employees, are in compliance with agency policy requirements, and are considered Works of the United States Government. However, the findings and conclusions presented in this paper are those of the author(s) and do not necessarily reflect the views of the NIH or the U.S. Department of Health and Human Services.

## Author contributions

**Julia A Thayer**: Conceptualization; Investigation; Visualization; Methodology; Writing—original draft; Writing—review and editing. **Jennifer D Petersen**: Investigation; Visualization; Methodology; Writing—review and editing. **Xiaoping Huang**: Investigation; Visualization; Methodology; Writing—review and editing. **Luiza M Gruel Budet**: Investigation. **James Hawrot**: Resources; Software. **Daniel M Ramos**: Resources; Software. **Shiori Sekine**: Resources. **Yan Li**: Formal analysis; Methodology. **Michael E Ward**: Resources. **Derek P Narendra**: Conceptualization; Resources; Supervision; Funding acquisition; Investigation; Visualization; Methodology; Writing—original draft; Writing—review and editing.

Source data underlying figure panels in this paper may have individual authorship assigned. Where available, figure panel/source data authorship is listed in the following database record: biostudies:S-SCDT-10_1038-S44318-025-00604-z.

## Funding

## Disclosure and competing interests statement

The authors declare no competing interests.

# Expanded View Figures

**Figure EV1. APEX-ER staining reliably identifies HeLa cells with knockdown of mitochondrial proteins for assessment of mitochondrial ultrastructure by TEM.** ▶

(A) Transmitted light image of HeLa[dCas9-BFP-ZIM3] cells shows the brown DAB reaction product after ~12 min of development time in cells transduced with APEX-ER (solid arrowheads) compared to cells expressing low or no APEX-ER (open arrowheads). Scale bar = 100 μm. (B) TEM image of a cell with low or no APEX-ER staining in the ER lumen (open arrowheads) next to a cell with darkly stained ER lumen (closed arrowheads) indicating a high level of APEX-ER expression. Scale bar = 2 μm. (C) Examples of abnormal ultrastructural features of mitochondria expressing sgRNAs for the protein indicated in the lower left corner of each image. Yellow arrows indicate the abnormal cristae feature listed at the top of each image; yellow asterisks indicate areas with sparse cristae. Red arrows indicate the fluffy aggregate observed in the matrix of cells transduced with sgRNA DNAJA3. Blue asterisks indicate cytosol enclosed cup-shaped mitochondria. In some cases, examples shown were cropped from the same cell. Scale bar = 500 nm.

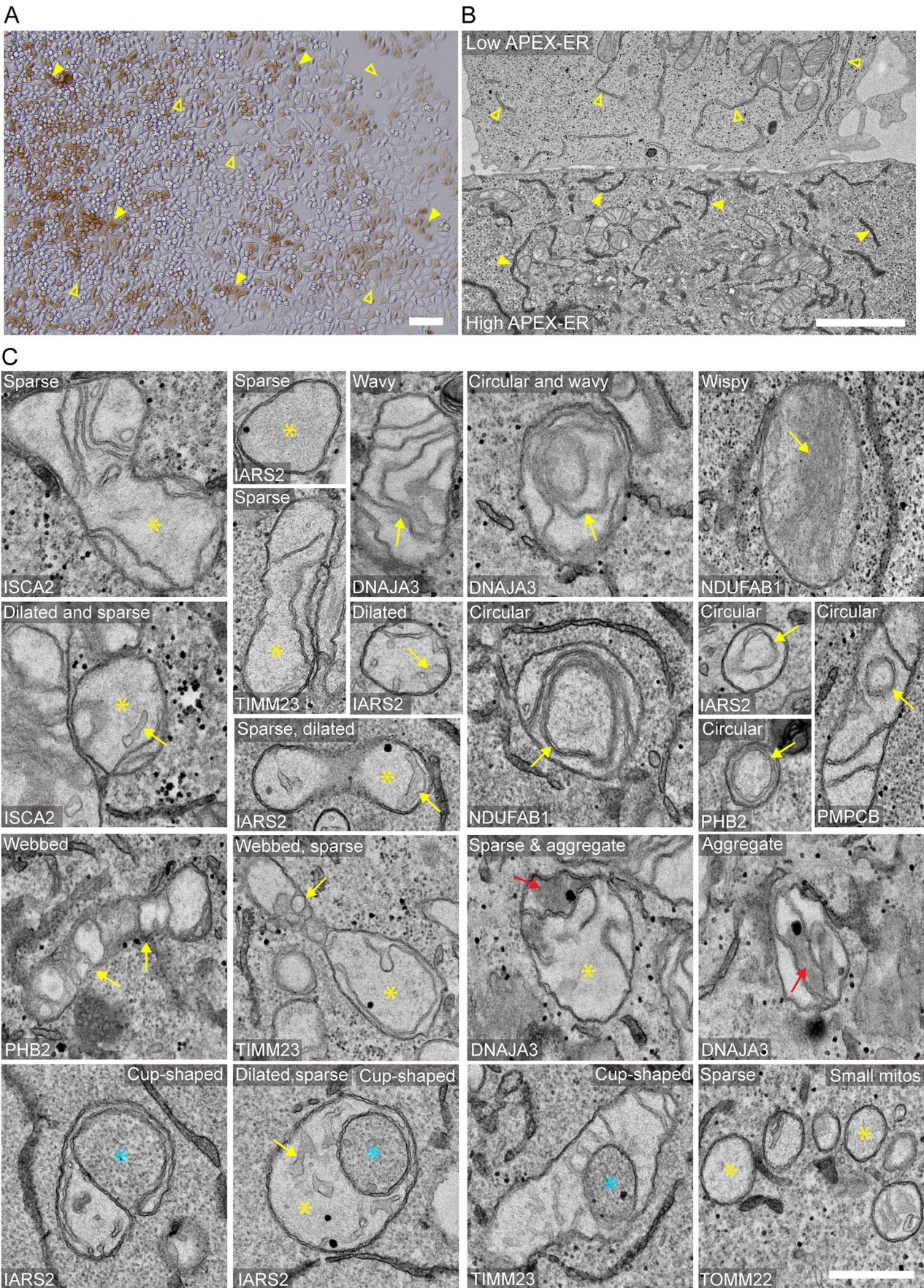

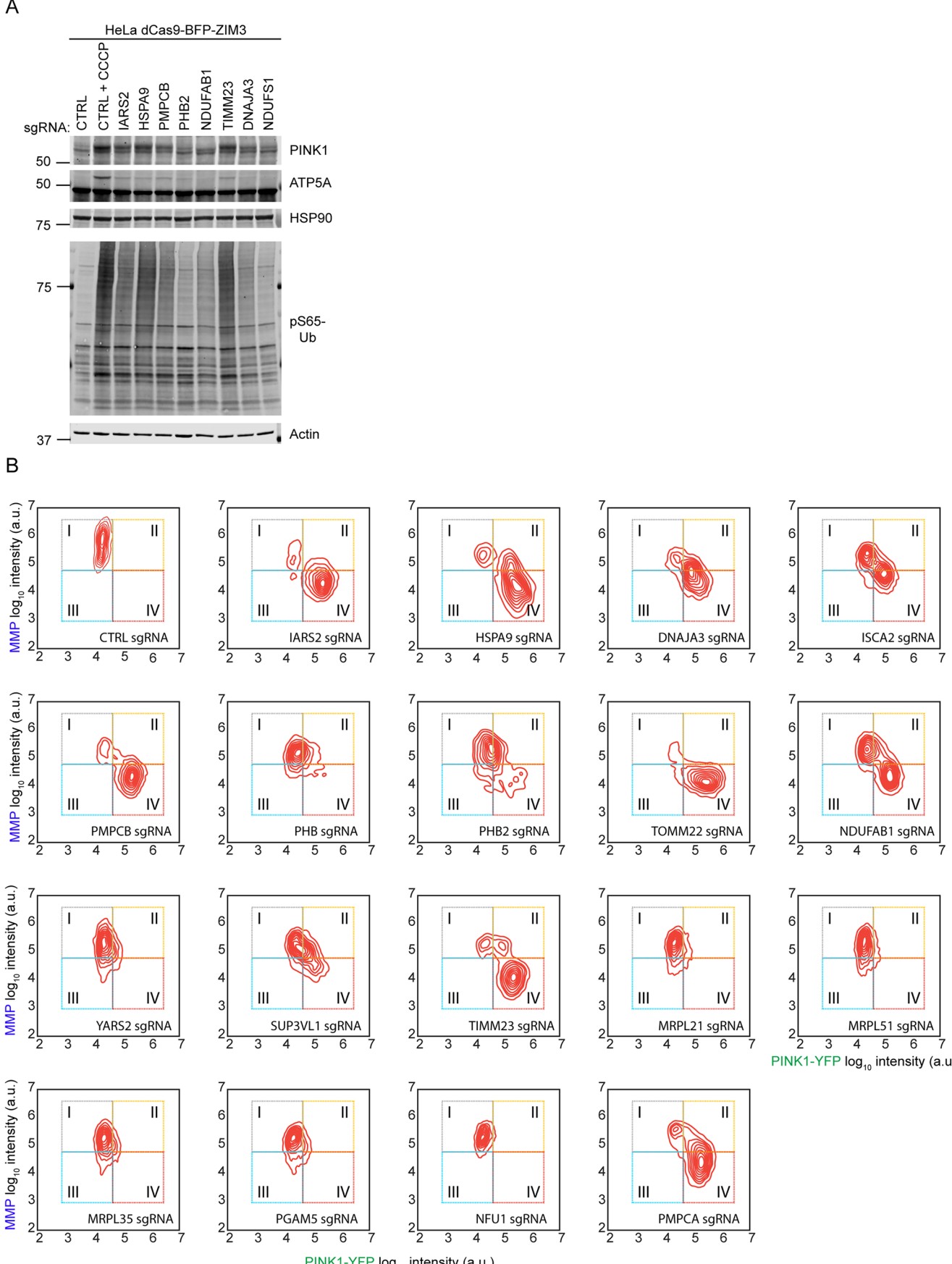

◀ **Figure EV2.  Top PINK1–Parkin activators stabilize endogenous PINK1 and lower MMP.**

(A) Representative immunoblots of HeLa[dCas9-BFP-ZIM3] cells treated with 10 μM CCCP for 4 h or transduced with indicated sgRNAs, illustrating PINK1 stabilization and activation. $N = 3$ independent experiments. (B) Representative 2D kernel density plots comparing single-cell PINK1-YFP intensity and intensity of the MMP sensitive dye MitoLite NIR as in (Fig. 4F).

  

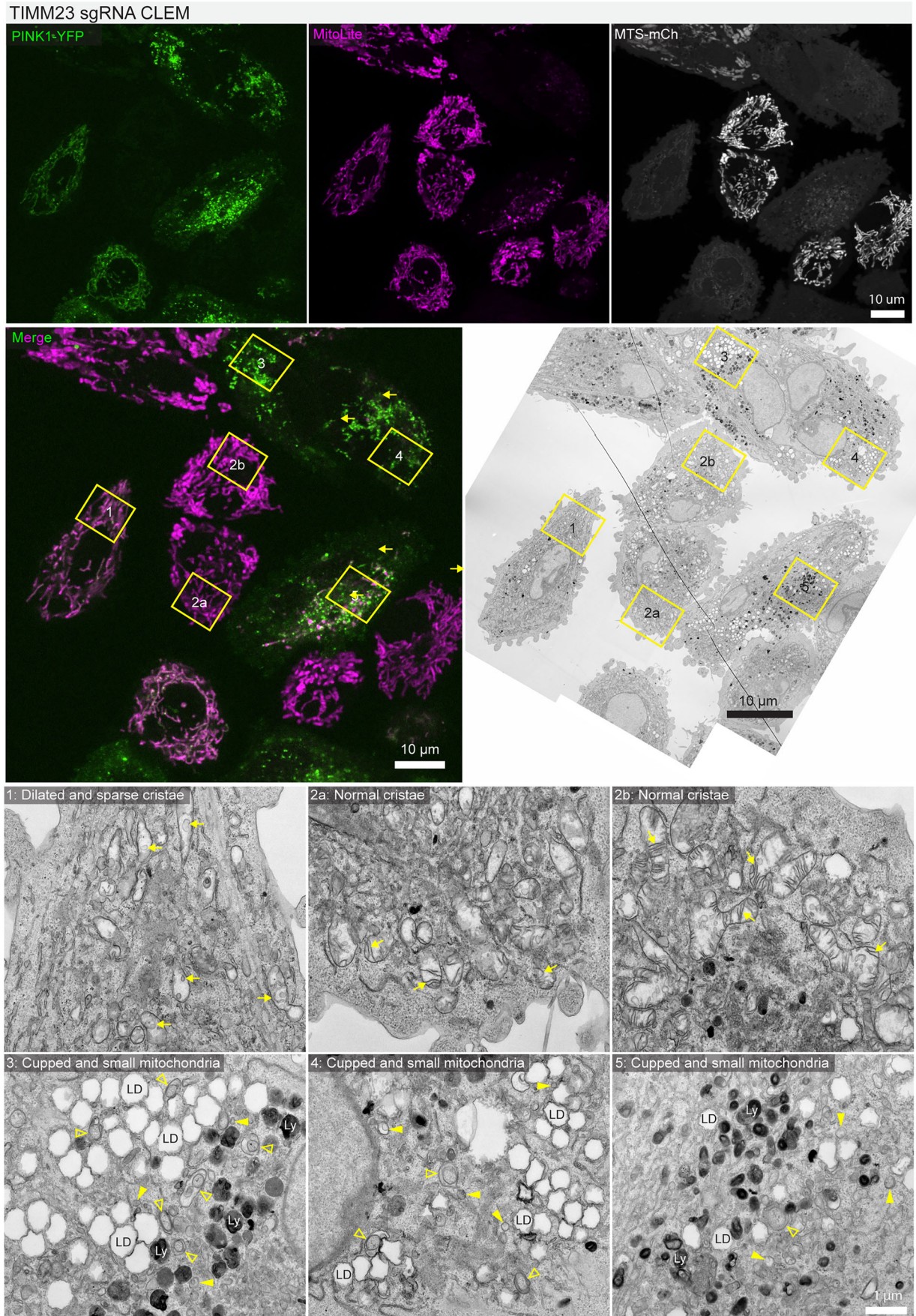

◀  **Figure EV3.   CLEM of TIMM23 knockdown cells demonstrates different ultrastructural features of PINK1-targeted mitochondria with and without MMP.**

CLEM images from a TIMM23 KD HeLa[PINK1-YFP+MTS-mCh] cells, illustrating that cells with maintained MMP and PINK1-YFP accumulation did not result in mitochondrial cupping (region 1) while cells where PINK1-YFP accumulated and MMP was lost mitochondrial cupping was observed (regions 3–5), these cells also had lipid droplets and lysosomes around the mitochondria. Live cells were imaged in the presence of MMP dye MitoLite, followed by fixation and EM processing/imaging. Yellow boxes show the corelating cells between the light microscopy and EM. Arrows – cristae feature, empty arrowhead – cupped mito, closed arrowhead – small mito, LD – lipid droplet, and Ly – lysosome. Scale bar $=10\,\mu m$ on all images but Scale bar $=1\,\mu m$ on zoomed in regions of TEM (bottom of figure).

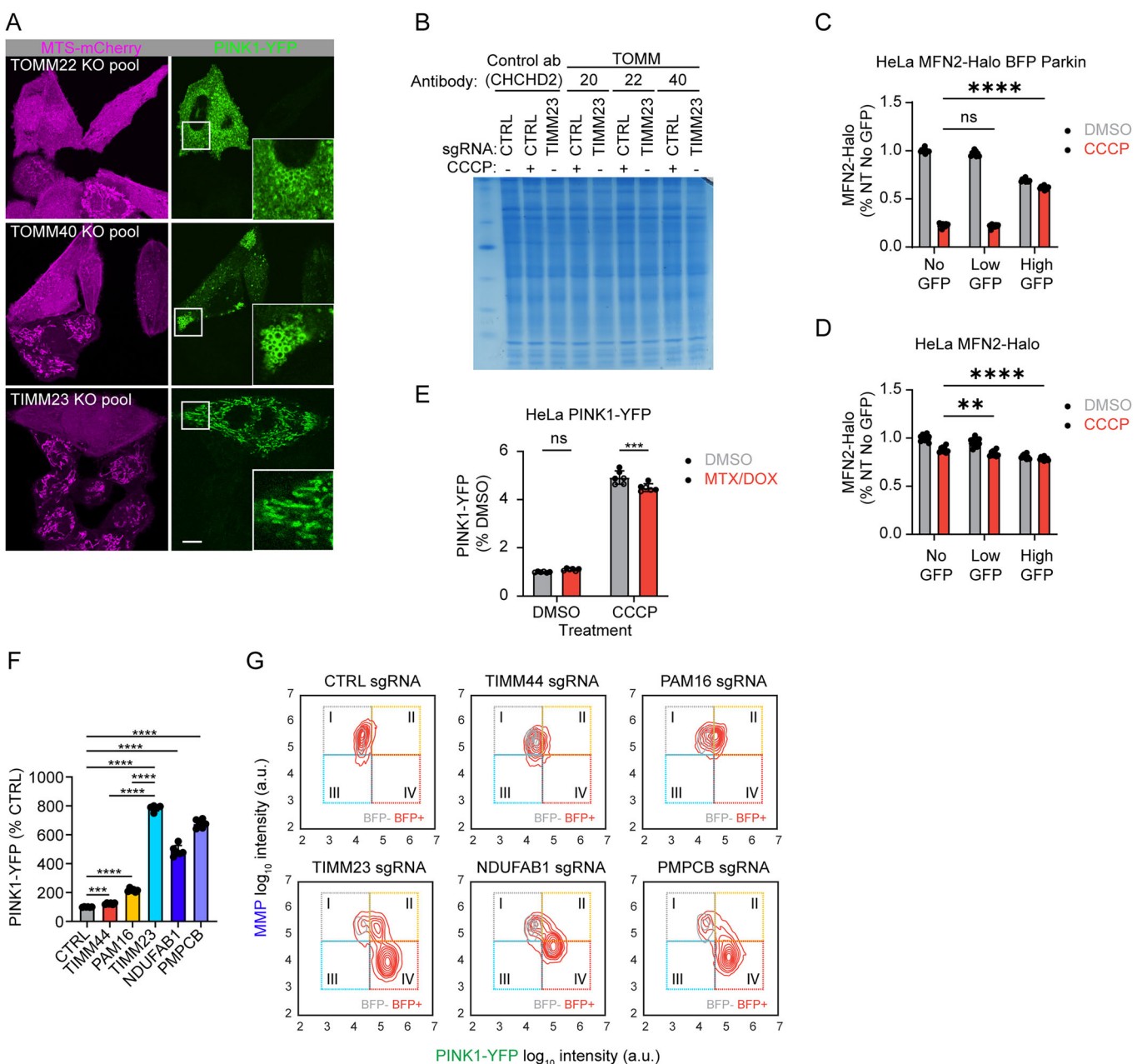

**Figure EV4. PINK1 is differentially affected by loss of the TOM complex, TIM23, and the PAM import motor.**

(A) Representative confocal images of TOMM22, TOMM40, TIMM23 KO pools in HeLa^PINK1-YFP+MTS-mCh^ cells showing PINK1-YFP accumulates in the same pattern as observed by CRISPRi. Images were obtained 7 or 8 days after electroporation with Cas9 ribonucleoprotein complexes. Scale bar 10 = μm. (B) Total protein measured via SimplyBlue SafeStain of same gel as in (Fig. 5J), demonstrating equal loading. (C) Flow cytometry data of HeLa^MFN2-Halo^ cells + BFP-Parkin transfected with ATP5MG-mCherry-sfGFP and treated +/− 10 μM CCCP for 4 h, illustrating differences in MFN2-Halo levels in the presence of the clogger. ns$P = 0.5976$, ****$P \leq 0.0001$ (exact $P$ value $P < 1e-15$) by two-way ANOVA with Tukey's multiple comparisons test. Error bars mean +/− SD. $N = 3$ independent experiments, 9 replicates. (D) Flow cytometry data of HeLa^MFN2-Halo^ cells transfected with ATP5MG-mCherry-sfGFP and treated +/− 10 μM CCCP for 4 h, illustrating differences in MFN2-Halo levels in the presence of the clogger. **$P = 0.0086$, ****$P \leq 0.0001$ (exact $P$ value $P = 7.4e-09$) by two-way ANOVA with Tukey's multiple comparisons test. Error bars mean +/− SD. $N = 3$ independent experiments, 10 replicates. (E) Flow cytometry data of HeLa^PINK1-YFP^ cells expressing TOMM70 endogenously tagged with HaloTag and IMMT-DHFR clogger. Cells were treated +/− 20 μM CCCP for 4 h, demonstrating differences in PINK1-YFP stabilization. ns$P = 0.4962$, ***$P = 0.0004$ by two-way ANOVA with Šídák's multiple comparisons test. Error bars mean +/− SD. $N = 6$ replicates run on two different occasions (separate occasions denoted by open or closed circles). (F) Flow cytometry of HeLa^PINK1-YFP^ cells. ***$P = 0.0003$, ****$P \leq 0.0001$ (exact $P$ values - CTRL vs PAM16, $P = 1.1e-05$; CTRL vs TIMM23, $P = 9.3e-07$; CTRL vs NDUFAB1, $P = 9e-06$; CTRL vs PMPCB, $P = 3.6e-07$; TIMM44 vs TIMM23, $P = 1.2e-06$; PAM16 vs TIMM23, 1.3e-08). Error bars mean +/− SD. $N = 6$ replicates (5 for TIMM23) from 2 independent transductions. (G) Representative 2D kernel density plots comparing single-cell PINK1-YFP intensity and intensity of the MMP sensitive dye MitoLite NIR as in (Fig. 4F).

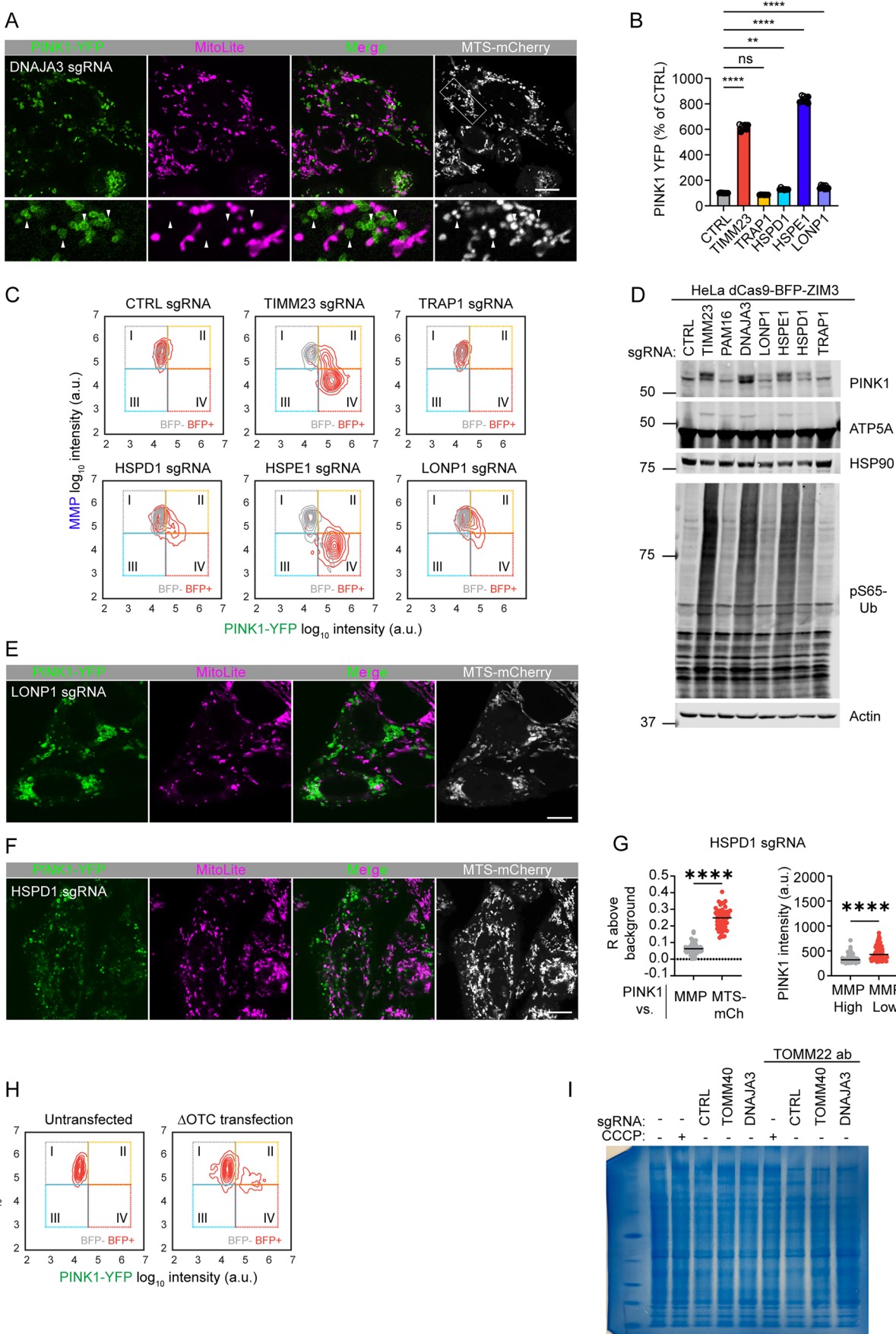

◀ **Figure EV5. Protein misfolding in the mitochondrial matrix activates the PINK1–Parkin pathway through its effect on MMP.**

(A) Representative confocal image of live HeLa^PINK1-YFP+MTS-mCh cells transduced with a guide targeting DNAJA3. Rectangle images are zoomed in area of white rectangle outlined in square image above. Arrowheads point to cells that have high PINK1-YFP expression on mitochondria that have lost MMP and import is not blocked. Scale bar = 10 μm. (B) Flow cytometry of HeLa^PINK1-YFP cells transduce with the indicated sgRNA, illustrating PINK1-YFP levels. ns$P = 0.3001$, **$P = 0.0013$, ****$P \leq 0.0001$ (exact $P$ values - CTRL vs TIMM23, $P < 1e-15$; CTRL vs HSPE1, p = 1e-15; CTRL vs LONP1, $P = 8.1e-07$), by ordinary one-way ANOVA with Šídák's multiple comparisons test. Error bars mean $+/-$ SD. $N =$ at least 7 replicates from 2 independent transductions (separate transductions denoted by open or closed circles). (C) Representative 2D kernel density plots comparing single-cell PINK1-YFP intensity and intensity of the MMP sensitive dye MitoLite NIR from the same experiment as shown in Fig. EV5B. (D) Representative immunoblots of HeLa^dCas9-BFP-ZIM3 cells transduced with indicated sgRNAs, illustrating PINK1 stabilization and activation. $N = 3$ independent experiments. (E) Representative confocal image of live HeLa^PINK1-YFP+MTS-mCh cells transduced with a guide targeting LONP1. PINK1-YFP accumulated preferentially on mitochondria with low MMP. Scale bar = 10 μm. (F) Representative confocal image of live HeLa^PINK1-YFP+MTS-mCh cells transduced with a guide targeting HSPD1. PINK1-YFP accumulated preferentially on mitochondria with low MMP. Scale bar = 10 μm. (G) Left graph quantification of cells in (Fig. EV5F) was performed as in (Fig. 3G). ****$P \leq 0.0001$ (exact $P$ value $P < 1e-15$) by two-tailed Mann–Whitney test. $N = 56$ cells from 4 wells and 2 separate transductions. Right graph quantification of cells in (Fig. EV5F) was performed as in (Fig. 3F). ****$P \leq 0.0001$ (exact $P$ value $P < 1e-15$) by Wilcoxon matched-pairs signed rank test. $N = 56$ cells from 4 wells and 2 separate transductions. (H) Representative 2D kernel density plots comparing single-cell PINK1-YFP intensity and intensity of the MMP sensitive dye MitoLite NIR in HeLa^PINK1-YFP + TOMM70-Halo cells $+/-$ transient transfection of ΔOTC. (I) Total protein measured via SimplyBlue SafeStain of same gel in (Fig. 7D), demonstrating equal loading.

