## [Peer Review File · The EMBO Journal]

A Unified Mechanism for Mitochondrial Damage Sensing in PINK1-Parkin–Mediated Mitophagy

Julia Thayer, Jennifer Petersen, Xioping Huang, Luiza Gruel Budet, James Hawrot, Daniel Ramos, Shiori Sekine, Yan Li, Michael E Ward, and Derek Narendra

Corresponding author(s): Derek Narendra (derek.narendra@nih.gov)

Review Timeline:

Submission Date:	26th Feb 25
Editorial Decision:	16th Apr 25
Revision Received:	6th Aug 25
Editorial Decision:	8th Sep 25
Revision Received:	10th Sep 25
Accepted:	22nd Sep 25

Editor: William Teale

Transaction Report:

Dear Derek,

Thank you again for the submission of your manuscript entitled "Novel reporter of the PINK1-Parkin mitophagy pathway identifies its damage sensor in the import gate" and for your patience during the review process. We have now received the reports from the referees, which I copy below.

As you can see from their comments, all referees were enthusiastically supportive of publication of your work. They do, however, raise some minor points that will require your attention before your manuscript can be published in The EMBO Journal. I am also of the opinion that the manuscript would benefit from key questions and discussion points about the nature of the mitochondrial damage sensor being more clearly and prominently elucidated from the start.

I would therefore like to invite you to address the comments of all referees in a revised version of the manuscript. I should add that it is The EMBO Journal policy to allow only a single major round of revision and that it is therefore important to resolve the main concerns at this stage. Please contact me if you have any questions, need further input on the referee comments or if you anticipate any problems in addressing any of their points. Please, follow the instructions below when preparing your manuscript for resubmission.

I would also like to point out that as a matter of policy, competing manuscripts published during this period will not be taken into consideration in our assessment of the novelty presented by your study ("scooping" protection). We have extended this 'scooping protection policy' beyond the usual 3 month revision timeline to cover the period required for a full revision to address the essential experimental issues. Please contact me if you see a paper with related content published elsewhere to discuss the appropriate course of action.

Again, please contact me at any time during revision if you need any help or have further questions.

Thank you very much again for the opportunity to consider your work for publication. I look forward to your revision.

Best regards,

William

William Teale, Ph.D.
Editor
The EMBO Journal

When submitting your revised manuscript, please carefully review the instructions below and include the following items:

- 1) a .docx formatted version of the manuscript text (including legends for main figures, EV figures and tables). Please make sure that the changes are highlighted to be clearly visible.
- 2) individual production quality figure files as .eps, .tif, .jpg (one file per figure).
- 3) a .docx formatted letter INCLUDING the reviewers' reports and your detailed point-by-point response to their comments. As part of the EMBO Press transparent editorial process, the point-by-point response is part of the Review Process File (RPF), which will be published alongside your paper.
- 4) a complete author checklist, which you can download from our author guidelines ([https://wol-prod-cdn.literatumonline.com/pb-assets/embo-site/Author Checklist%20-%20EMBO%20J-1561436015657.xlsx](https://wol-prod-cdn.literatumonline.com/pb-assets/embo-site/Author%20Checklist%20-%20EMBO%20J-1561436015657.xlsx)). Please insert information in the checklist that is also reflected in the manuscript. The completed author checklist will also be part of the RPF.
- 5) Please note that all corresponding authors are required to supply an ORCID ID for their name upon submission of a revised manuscript.
- 6) We require a 'Data Availability' section after the Materials and Methods. Before submitting your revision, primary datasets produced in this study need to be deposited in an appropriate public database, and the accession numbers and database listed under 'Data Availability'. Please remember to provide a reviewer password if the datasets are not yet public (see

<https://www.embopress.org/page/journal/14602075/authorguide#datadeposition>). If no data deposition in external databases is needed for this paper, please then state in this section: This study includes no data deposited in external repositories. Note that the Data Availability Section is restricted to new primary data that are part of this study.

Note - All links should resolve to a page where the data can be accessed.

8) For data quantification: please specify the name of the statistical test used to generate error bars and P values, the number (n) of independent experiments (specify technical or biological replicates) underlying each data point and the test used to calculate p-values in each figure legend. The figure legends should contain a basic description of n, P and the test applied. Graphs must include a description of the bars and the error bars (s.d., s.e.m.).

9) We would also encourage you to include the source data for figure panels that show essential data. Numerical data can be provided as individual .xls or .csv files (including a tab describing the data). For 'blots' or microscopy, uncropped images should be submitted (using a zip archive or a single pdf per main figure if multiple images need to be supplied for one panel). Additional information on source data and instruction on how to label the files are available at .

10) We replaced Supplementary Information with Expanded View (EV) Figures and Tables that are collapsible/expandable online (see examples in <https://www.embopress.org/doi/10.15252/embj.201695874>). A maximum of 5 EV Figures can be typeset. EV Figures should be cited as 'Figure EV1, Figure EV2" etc. in the text and their respective legends should be included in the main text after the legends of regular figures.

12) Our journal encourages inclusion of *data citations in the reference list* to directly cite datasets that were re-used and obtained from public databases. Data citations in the article text are distinct from normal bibliographical citations and should directly link to the database records from which the data can be accessed. In the main text, data citations are formatted as follows: "Data ref: Smith et al, 2001" or "Data ref: NCBI Sequence Read Archive PRJNA342805, 2017". In the Reference list, data citations must be labeled with "[DATASET]". A data reference must provide the database name, accession number/identifiers and a resolvable link to the landing page from which the data can be accessed at the end of the reference. Further instructions are available at .

13) In order to increase the reproducibility and reach of your work, The EMBO Journal includes a table of reagents that were used in the study. Please provide this along with your revisions.

Further instructions for preparing your revised manuscript:

We realize that it is difficult to revise to a specific deadline. In the interest of protecting the conceptual advance provided by the work, we recommend a revision within 3 months (15th Jul 2025). Please discuss the revision progress ahead of this time with the editor if you require more time to complete the revisions. Use the link below to submit your revision:

Referee #1:

This manuscript from the Narendra group describes a comprehensive study that aims to answer the question, how PINK1 is actually activated on damaged mitochondria to trigger mitophagy - a mechanisms that is particularly important for neuronal survival. PINK1 is a kinase that marks dysfunctional mitochondria and recruits the E3 ligase parkin that ubiquitinates mitochondrial proteins which are then recognized by mitophagy adaptors. Initial studies showed that in healthy mitochondria the vast majority of PINK1 is imported into mitochondria via the TOM-TIM23 pathway to the inner mitochondrial membrane, where it is gets cleaved by the rhomboid protease PARL. Proteolytically processed PINK1 is then diffusing back into the cytosol, where it is degraded by the proteasome. On damaged mitochondria, PINK1 import arrests prior to PARL cleavage and the translocation intermediate becomes stabilized at import sites through an unknown mechanism. Stabilized, TOM-associated PINK1 appeared to be the active form of the kinase, as a first step in the downstream cascade phosphorylating ubiquitin on the mitochondrial surface. Substantial evidence has been provided that the collapse of the mitochondrial membrane potential (MMP) is a major trigger for PINK1 stabilization and activation on the mitochondrial surface. However, several papers in recent years reported apparently contradicting results, e.g. on the role of TIM23 in PINK1 stabilization, and also questioned the loss of MMP as a central mechanism for PINK1 activation.

In order to clarify the situation to some extent, Narendra and co-workers established sophisticated genome-wide CRISPRi screens to identify in an unbiased manner 1) factors that are required for activation of PINK1 on damaged mitochondria and 2) gene knock-outs that induce PINK1 activation in the absence of external stressors. Although many papers have addressed similar questions in the past, this study is unprecedentedly comprehensive, perfectly controlled and in part independent from overexpression of parkin. This together makes the work original, unique and very important.

Amongst the most exciting novel findings is that block of glycolysis in the cytosol impairs de novo synthesis of PINK1 and reduces the PINK1 molecules en route to mitochondria and, thus, available as sensors. Moreover, for virtually all mutations inducing PINK1 stabilization the smallest common denominator is the loss of MMP. Sophisticated single cell analysis demonstrates convincingly that although differential mitochondrial defects are induced by the analysed mutations the best correlation to PINK1 stabilization is with the local collapse of MPP within the mitochondrial network. Interestingly, although this study confirms that PINK1 preferentially accumulates in TOM-TIM23 imports sites upon loss of MMP, TIM23 per se is not necessary to induce stabilization of PINK1 at the TOM complex.

This work represents a very important step forward in the crowded research field of PINK1/parkin biology. It solves several debates due to apparently conflicting results in the literature by applying a strict regime of controls and complementing experiments from a different angle. The astonishing amount of data and also of experimental setups and cell lines provides a valuable resource for the field. The manuscript impresses with its careful and stringent logical composition with a number of excellent controls and double-checks. This tremendous piece of work should definitely deserves publication in the EMBO

Journal and will be exciting to read for a broad audience. Prior to publication I have a few requests to the author that they should address in a convincing manner.

1. The manuscript is extremely dense and quite difficult to access for non-expert readers jumping forth and back between different cell lines and reporter combinations. I strongly encourage to authors to streamline the text, especially in the Results section to improve readability for a broader audience (different in terms of scientific background and mother tongue).
2. Recent studies on mitochondrial import stress used clogger constructs to block the TOM-TIM23 pathway in cells. Quantitative proteomics analyses show that the number of TOM complexes is several-fold higher than the number of TIM23 complexes. Thus, accumulation of a clogger targeted into the presequence pathway likely leads to a strong decrease of available TIM23 complexes, whereas TOM complexes are still available and the membrane potential is still intact. Can the authors either easily test or thoughtfully speculate what will happen to the PINK1-parkin pathway under these conditions that may reflect real pathological situation as well?
3. The only moment where the authors leave solid ground in the interpretation of their data is when they come up with the hypothesis that the energy provided by the MMP must be sufficient to separate PINK1 from its stable interaction with TOM for handover to TIM23. First, I do not fully understand how this would happen to the separated PINK1 in TIM23 knock-down cells. Second, and more important; I consider such semi-quantitative statement always dangerous. Can the authors provide any calculations or modelling approaches that reveal, if such a scenario is realistic from the bioenergetic perspective? Since the electrophoretic force depends on the electrical field and the number of charges in the moved object, will modifying the charge in the N-terminal region of PINK1 alter the MMP threshold for PINK1 activation?
4. I am wondering which role PARL cleavage play in the model favoured by the authors. Because PARL1-KO was not amongst the hits that show PINK1 stabilization without treatment, can one assume the PARL cleavage is actually not necessary to release PINK1 into the cytosol from intact mitochondria?
5. Did the authors consider the possibility that a fraction of PINK1 may also be completely imported into mitochondrial matrix, as some studies claim?

Referee #2:

Thayer et al present a comprehensive set of analyses aimed at furthering our understanding of the PINK1-Parkin pathway and in particular the mechanism(s) by which PINK1 senses mitochondrial dysfunction using a combination of complementary approaches. The authors use a novel putative reporter for PINK1-Parkin activity, MFN2-Halo, to perform a number of complementary genome-wide CRISPRi screens to identify regulators of PINK1-Parkin pathway activity at single-cell level. This revealed multiple hits that they describe as facilitators or activators. Some of these are known or anticipated and some are new. Focussing on hits relating to glycolysis, they presented evidence that glycolytic ATP provides the energy for PINK1 synthesis following mitochondrial perturbation. Interestingly, reduced mitochondrial ATP synthesis (a physiologically relevant mitochondrial disruption) sensitised PINK1 stabilisation to reduced MMP. Then focussing on PINK1-Parkin activators by comparing multiple reporters of PINK1-Parkin activity, proteomics, ultrastructure and MMP, they surmised that multiple perturbations that activate the PINK1-Parkin pathway converge on disrupting the MMP. This led the authors to investigate in detail the relative roles of TOM and TIM import channels. Here, it was noted that while loss of TIM23 blocked import on OMM, loss of TOMM22 and TOMM40 led to PINK1-YFP accumulating on an unidentified compartment near lipid droplets. Ultimately, the results showed that TOM but not TIM23 was required for OMM stabilisation. Further targeted experiments supported that TOMM5 and TOMM20 promote stabilisation, and disruption of the PAM complex likely stabilises PINK1 by affecting MMP.

Overall, the authors present a coherent and thoroughly performed study, originating from unbiased screens using a novel reporter approach, combined with multiple orthogonal assays to analysed in detail the mechanisms on which various screen hits converge. Consistently, these seem to point to loss of membrane potential and the stabilisation of PINK1 in the TOM channel, which is entirely in line with the predominant view in the field, albeit based on studies using potent mitochondrial poisons and depolarising agents. In this regard, there is little of surprise or controversy here and the observations could be at risk of confirmation bias since impacts on MMP are the most obvious mechanism to investigate. Nevertheless, the experiments are well executed and nicely presented in a cogent narrative. As such, this study is a valuable addition to the field and provides additional insight into the molecular mechanisms of an important quality control process. As ever it still remains to be seen what is happening in vivo and in particularly in ageing neurons relevant to Parkinson's disease, but for the present study I have only relatively minor comments for consideration.

Specific points:

1. Just for clarity, if the genome-wide screens were performed in duplicate, what is represented in the Supplementary Tables exactly? It looks like single values are presented for each screening condition. Is this the mean of the duplicates?

2. Fig. 3F-G. What is shown exactly? what does a single data point represent - measurement of a single mitochondrion or from a single cell or what? How is mean and variance determined?

Please make it clear how the statistics are applied to this and other similar data sets - are they applied to individual cells or independent transductions? Really, it should be the latter.

3. Fig. 4A-C. I guess that the open- vs closed-circle data points represent the two independent transductions as mentioned in the legends. This is a welcome touch, and it would be nice to make this clear in the legend also.

4. Fig. 5D. The CLEM here is very nice but the confocal image in D is not very clear and doesn't look like that shown in C. Can this be improved? It is the combination of observations in C and D that allow the conclusion that PINK1 accumulates near lipid droplets. Considering this, where do the authors think the PINK1 is exactly? On the vesicles? cytosolic? How does this compare with S5A where there are very few vesicular structures. Since diffraction in the Z axis can confound accurate colocalization, a series of TEM sections would be interesting to see what is above/below the LDs. It would be fantastic to see the localization of PINK1-YFP by immuno-gold EM but this may be beyond the desires of the authors at this stage.

Does this signal co-localise with TOMM70 or cytosolic HSP70s, given the results in Fig. 7. Does this localisation occur with endogenous PINK1, or is this an artefact of exogenous PINK1-YFP?

5. Also, in 5C, the TOMM22 sgRNA images; the PINK1-YFP zoom images in the separate vs merged channels looks subtly different. Different plane? Please check.

6. With the differing effects of TOMM22/40 vs TIM23 KD, I am curious to know what happens with a double KD, but this may perhaps have limited interpretation.

7. MitoLite NIR is a new reagent to me but looks like a useful tool for the field and an alternative to traditional dyes such as TMRE. Since there are very few technical details on the company website, it would be helpful if the authors (recognised experts in the field) could perform a few simple tests to demonstrate the functionality of this reagent compared to other well-known ones, for instance, showing that the dye does not localise to chemically depolarised mitochondria and, importantly, whether its localisation is reversible, i.e., pre-loading energised mitochondria with the dye, it redistributes upon depolarisation.

8. In general, colour-coding of different conditions is nice and generally helpful. However, the authors may want to change their colour schemes when the coding changes, e.g., in Fig. 4. A-D vs G.

9. If space allows, the authors could expand a bit on the abstract to make it more accessible to a broader audience.

10. Finally, the authors may want to comment on their findings in relation to the recent paper from Komander's group on the structure of PINK1 with TOM/VDAC complexes, particularly with respect to the interaction with TOMM5/7.

Corrections:

- Fig 4C. "PGRAM5" is a misspelling

- I assume "This identifies maturation of newly important proteins..." in the sentence 4 lines up from the bottom should read 'newly imported proteins'.

Referee #3:

Thayer et al report a novel reporter of Parkin-PINK1 pathway, based on MFN2 tagged with Halo. They use that reporter to perform a genome-wide CRISPR screen. While several such screen were reported previously, this is the first that looks at a direct substrate of Parkin, and the study is completed by very elegant biochemistry and proteomics. While most of the results are largely confirmatory, the paper brings about a substantial resource for the field and helps clear up several aspects of the mechanism of PINK1 activation. One of the most significant aspects is the consolidation of the idea that loss of MMP is the main driver of PINK1 accumulation, and that knockdown of several genes can cause heterogenous loss of MMP, observed by microscopy or single-cell level, which is why they cause bulk accumulation of PINK1. It is rare that I write this, but I would almost accept the paper as is, except for the minor revisions described below.

1) The requirement for glycolytic ATP has been reported previously by Lee et al (2015, JBC) and McLelland et al (2018, Elife), which should be cited. Lee et al indeed studied in depth the role of ATP generation and inhibition of glycolysis on PINK1 accumulation. Likewise, the impact of galactose on Parkin recruitment was also previously observed by McLelland et al. Please correct accordingly.

2) I don't understand this sentence on page 11: "Similarly, PINK1-YFP correlated with MTS-mSc but poorly with MMP (Fig. 3G)." Shouldn't it be the opposite, based on the previous sentence and Fig. 3F? In fact PINK1-YFP appears like it is anti-correlated with MMP. Something is weird in the way correlation is reported in Fig. 3G (as well as 4I, 4J, left panels Robs-Rrand)... in the case of depolarization-induced PINK1 accumulation, MMP and PINK1 levels should be anti-correlated, as shown by the higher levels of PINK1 in MMP low conditions (Fig. 3F and 4I, right panels). The authors should perhaps use a signed correlation

coefficient to analyze the data, and then compare with the random distribution, or find some way of correlating PINK1 levels with the loss of MMP.

3) I would qualify this statement on page 16: "Together this establishes that TOM but not TIM23 is required for PINK1 stabilization and activation on the OMM". Here, the authors show that TIM23 knockdown causes PINK1 activation, but that does not mean TIM23 may not be required for PINK1 stabilization induced by loss of MMP. In fact, Akabane et al (2023) found that knockdown of TIM23 indeed reduced PINK1 activation under conditions of depolarization, and Eldeeb et al (2024) obtained similar results with CRISPRi against TIM50. Please modify the conclusion accordingly and discuss.

4) The authors claim to be surprised by the effect of TOM5 as a facilitator, but a previous study already reported TOMM5 as a facilitator using a genome-wide screen for mitophagy (Hosino et al 2019, Nature, Fig. 1c), although granted they didn't discuss it in the text. This group also mostly studied the role of the nucleotide exchanger ANT1 in mitophagy, which should be discussed in the context of glycolytic vs mitochondrial ATP generation. Furthermore, given the recent publication of the structure of the PINK1-TOM complex by the Komander lab (Callegari et al Science 2025), the authors may want to revisit their interpretation of the TOMs knockout results.

5) While the authors clearly demonstrate that PINK1 accumulation induced by PAM disruption is mediated via loss of MMP. However, they do not address the model whereby expression of delta-OTC induces PINK1 despite no loss of MMP accumulation (Jin & Youle 2013, Burman et al 2017). They should at least discuss this result in the context of their own.

From: "w.teale@embojournal.org" <w.teale@embojournal.org>
Date: Wednesday, April 16, 2025 at 9:09:26 AM
To: "Narendra, Derek (NIH/NINDS) [E]" <derek.narendra@nih.gov>
Subject: [EXTERNAL] Manuscript EMBOJ-2025-120613 - Decision

Dear Derek,

Thank you again for the submission of your manuscript entitled "Novel reporter of the PINK1-Parkin mitophagy pathway identifies its damage sensor in the import gate" and for your patience during the review process. We have now received the reports from the referees, which I copy below.

As you can see from their comments, all referees were enthusiastically supportive of publication of your work. They do, however, raise some minor points that will require your attention before your manuscript can be published in The EMBO Journal. I am also of the opinion that the manuscript would benefit from key questions and discussion points about the nature of the mitochondrial damage sensor being more clearly and prominently elucidated from the start.

We appreciate the enthusiastic support of all referees. We detail below how we addressed the minor points in our revised manuscript.

We have additionally corrected three errors that were discovered on preparing source data and depositing plasmids with Addgene. These do not substantively change the data or conclusions. First, we discovered that the matrix targeted red fluorescent protein reporter we used in multiple experiments is mCherry and not mScarlett, as we mistakenly identified in the initial submission. We had received the incorrect plasmid map, which we discovered on sequencing prior to submitting to Addgene. We have corrected this throughout the text and Figures. Second, in preparing Source Data we realized that the image cropped for Fig. 2G showing immunoblotting of actin contained superimposed boxes outside the cropped region. We have replaced this image with a cropped version of the original image. Finally, in Figure 5C we mistakenly identified the TOMM70 as TOMM70-Halo in the figure. The TOMM70 for this set of images was visualized by immunostaining for endogenous TOMM70. TOMM70-Halo was used for the related CLEM experiment. We apologize for these errors.

We have also expanded our discussion of key questions at the start of our manuscript, following the Editor's suggestion. The relevant paragraph in our Introduction now reads as follows:

"Although many details of the PINK1-Parkin pathway have been worked out, several key questions remain. One of the most pressing is understanding how the PINK1-Parkin response is shaped by individual components of the TOM-TIM23 import precursor pathway and the two driving forces for mitochondrial protein import: the mitochondrial membrane potential (MMP) and (for some but not all precursors) the ATP-dependent presequence translocase-associated motor (PAM) complex¹¹. While it was recognized early that pharmacological disruption of the MMP activates the PINK1-Parkin pathway¹², it remains unclear whether MMP loss is the main trigger of PINK1-Parkin activation in the context of more physiologically relevant sources of mitochondrial damage. For instance, it was recently proposed that protein misfolding in mitochondrial matrix (e.g., due to loss of the quality control protease LONP1) may activate the PINK1-Parkin pathway by disrupting the PAM complex rather than the MMP¹³. In this model, misfolded protein in the mitochondrial matrix competes away the chaperone component of PAM, HSPA9, presumably disrupting PINK1 import and cleavage. However, it has not been established whether the PAM complex is required for endogenous PINK1 import, as required by the model, and whether activation of PINK1-Parkin by matrix protein misfolding is independent of MMP loss at the level of the individual mitochondrion. Finally, while recent studies have established that PINK1 is stabilized in a supercomplex with TOM and TIM23 translocases upon activation, it remains unclear which components of the TOM and TIM23 translocases contribute to PINK1 import and stability⁴⁻⁶. Answers to these questions are critical for understanding how the PINK1-Parkin pathway senses mitochondrial damage. Additionally, an understanding of these mechanisms may reveal how the PINK1-Parkin mitophagy pathway can be tuned pharmacologically for therapeutic benefit."

I would therefore like to invite you to address the comments of all referees in a revised version of the manuscript. I should add that it is The EMBO Journal policy to allow only a single major round of revision and that it is therefore important to resolve the main concerns at this stage. Please contact me if you have any questions, need further input on the referee comments or if you anticipate any problems in addressing any of their points. Please, follow the instructions below when preparing your manuscript for resubmission.

I would also like to point out that as a matter of policy, competing manuscripts published during this period will not be taken into consideration in our assessment of the novelty presented by your study ("scooping" protection). We have extended this 'scooping protection policy' beyond the usual 3 month revision timeline to cover the period required for a full revision to address the essential experimental issues. Please contact me if you see a paper with related content published elsewhere to discuss the appropriate course of action.

When preparing your letter of response to the referees' comments, please bear in mind that this will form part of the Review Process File, and will therefore be available online to the community. For more details on our Transparent Editorial Process, please visit our website: <https://www.embopress.org/page/journal/14602075/authorguide#transparentprocess >

Again, please contact me at any time during revision if you need any help or have further questions.

Thank you very much again for the opportunity to consider your work for publication. I look forward to your revision.

Best regards,

William

William Teale, Ph.D.
Editor
The EMBO Journal

When submitting your revised manuscript, please carefully review the instructions below and include the following items:

- 1) a .docx formatted version of the manuscript text (including legends for main figures, EV figures and tables). Please make sure that the changes are highlighted to be clearly visible.
- 2) individual production quality figure files as .eps, .tif, .jpg (one file per figure).
- 3) a .docx formatted letter INCLUDING the reviewers' reports and your detailed point-by-point response to their comments. As part of the EMBO Press transparent editorial process, the point-by-point response is part of the Review Process File (RPF), which will be published alongside your paper.
- 4) a complete author checklist, which you can download from our author guidelines ([https://wol-prod-cdn.literatumonline.com/pb-assets/embo-site/Author Checklist%20-%20EMBO%20J-1561436015657.xlsx](https://wol-prod-cdn.literatumonline.com/pb-assets/embo-site/Author%20Checklist%20-%20EMBO%20J-1561436015657.xlsx)). Please insert information in the checklist that is also reflected in the manuscript. The completed author checklist will also be part of the RPF.
- 5) Please note that all corresponding authors are required to supply an ORCID ID for their name upon submission of a revised manuscript.

6) We require a 'Data Availability' section after the Materials and Methods. Before submitting your revision, primary datasets produced in this study need to be deposited in an appropriate public database, and the accession numbers and database listed under 'Data Availability'. Please remember to provide a reviewer password if the datasets are not yet public (see <https://www.embopress.org/page/journal/14602075/authorguide#datadeposition>). If no data deposition in external databases is needed for this paper, please then state in this section: This study includes no data deposited in external repositories. Note that the Data Availability Section is restricted to new primary data that are part of this study.

Note - All links should resolve to a page where the data can be accessed.

7) When assembling figures, please refer to our figure preparation guideline in order to ensure proper formatting and readability in print as well as on screen:
<http://bit.ly/EMBOPressFigurePreparationGuideline>

8) For data quantification: please specify the name of the statistical test used to generate error bars and P values, the number (n) of independent experiments (specify technical or biological replicates) underlying each data point and the test used to calculate p-values in each figure legend. The figure legends should contain a basic description of n, P and the test applied. Graphs must include a description of the bars and the error bars (s.d., s.e.m.).

9) We would also encourage you to include the source data for figure panels that show essential data. Numerical data can be provided as individual .xls or .csv files (including a tab describing the data). For 'blots' or microscopy, uncropped images should be submitted (using a zip archive or a single pdf per main figure if multiple images need to be supplied for one panel). Additional information on source data and instruction on how to label the files are available at <https://www.embopress.org/page/journal/14602075/authorguide#sourcedata> >.

10) We replaced Supplementary Information with Expanded View (EV) Figures and Tables that are collapsible/expandable online (see examples in <https://www.embopress.org/doi/10.15252/embj.201695874>). A maximum of 5 EV Figures can be typeset. EV Figures should be cited as 'Figure EV1, Figure EV2" etc. in the text and their respective legends should be included in the main text after the legends of regular figures.

- For the figures that you do NOT wish to display as Expanded View figures, they should be bundled together with their legends in a single PDF file called *Appendix*, which should start with a short Table of Content. Appendix figures should be referred to in the main text as: "Appendix Figure S1, Appendix Figure S2" etc. See detailed instructions regarding expanded view here: <https://www.embopress.org/page/journal/14602075/authorguide#expandedview> >.

12) Our journal encourages inclusion of *data citations in the reference list* to directly cite datasets that were re-used and obtained from public databases. Data citations in the article text are distinct from

normal bibliographical citations and should directly link to the database records from which the data can be accessed. In the main text, data citations are formatted as follows: "Data ref: Smith et al, 2001" or "Data ref: NCBI Sequence Read Archive PRJNA342805, 2017". In the Reference list, data citations must be labeled with "[DATASET]". A data reference must provide the database name, accession number/identifiers and a resolvable link to the landing page from which the data can be accessed at the end of the reference. Further instructions are available at <https://www.embopress.org/page/journal/14602075/authorguide#referencesformat> >.

13) In order to increase the reproducibility and reach of your work, The EMBO Journal includes a table of reagents that were used in the study. Please provide this along with your revisions.

Further instructions for preparing your revised manuscript:

See also figure legend

guidelines: <https://www.embopress.org/page/journal/14602075/authorguide#figureformat>

- a point-by-point response to the referees' comments, with a detailed description of the changes made (as a word file).

- a word file of the manuscript text.

- individual production quality figure files (one file per figure)

- a complete author checklist, which you can download from our author guidelines

(<https://www.embopress.org/page/journal/14602075/authorguide>).

- Expanded View files (replacing Supplementary Information)

Further information is available in our Guide For

Authors: <https://www.embopress.org/page/journal/14602075/authorguide>

We realize that it is difficult to revise to a specific deadline. In the interest of protecting the conceptual advance provided by the work, we recommend a revision within 3 months (15th Jul 2025). Please discuss the revision progress ahead of this time with the editor if you require more time to complete the revisions. Use the link below to submit your revision:

Referee #1:

This manuscript from the Narendra group describes a comprehensive study that aims to answer the question, how PINK1 is actually activated on damaged mitochondria to trigger mitophagy - a mechanism that is particularly important for neuronal survival. PINK1 is a kinase that marks dysfunctional mitochondria and recruits the E3 ligase parkin that ubiquitinates mitochondrial proteins which are then recognized by mitophagy adaptors. Initial studies showed that in healthy mitochondria the vast majority of PINK1 is imported into mitochondria via the TOM-TIM23 pathway to the inner mitochondrial membrane, where it gets cleaved by the rhomboid protease PARL. Proteolytically processed PINK1 is then diffusing back into the cytosol, where it is degraded by the proteasome. On damaged mitochondria, PINK1 import arrests prior to PARL cleavage and the translocation intermediate becomes stabilized at import sites through an unknown mechanism. Stabilized, TOM-associated PINK1 appeared to be the active form of the kinase, as a first step in the downstream cascade phosphorylating ubiquitin on the mitochondrial surface. Substantial evidence has been provided that the collapse of the mitochondrial membrane potential (MMP) is a major trigger for PINK1 stabilization and activation on the mitochondrial surface. However, several papers in recent years reported apparently contradicting results, e.g. on the role of TIM23 in PINK1 stabilization, and also questioned the loss of MMP as a central mechanism for PINK1 activation.

In order to clarify the situation to some extent, Narendra and co-workers established sophisticated genome-wide CRISPRi screens to identify in an unbiased manner 1) factors that are required for activation of PINK1 on damaged mitochondria and 2) gene knock-outs that induce PINK1 activation in the absence of external stressors. Although many papers have addressed similar questions in the past, this study is unprecedentedly comprehensive, perfectly controlled and in part independent from overexpression of parkin. This together makes the work original, unique and very important.

Amongst the most exciting novel findings is that block of glycolysis in the cytosol impairs de novo synthesis of PINK1 and reduces the PINK1 molecules en route to mitochondria and, thus, available as sensors. Moreover, for virtually all mutations inducing PINK1 stabilization the smallest common denominator is the loss of MMP. Sophisticated single cell analysis demonstrates convincingly that although differential mitochondrial defects are induced by the analysed mutations the best correlation to PINK1 stabilization is with the local collapse of MPP within the mitochondrial network. Interestingly, although this study confirms that PINK1 preferentially accumulates in TOM-TIM23 import sites upon loss of MMP, TIM23 per se is not necessary to induce stabilization of PINK1 at the TOM complex.

This work represents a very important step forward in the crowded research field of PINK1/parkin biology. It solves several debates due to apparently conflicting results in the literature by applying a strict regime of controls and complementing experiments from a different angle. The astonishing amount of data and also of experimental setups and cell lines provides a valuable resource for the field. The manuscript impresses with its careful and stringent logical composition with a number of excellent controls and double-checks. This tremendous piece of work should definitely deserve publication in the EMBO Journal and will be exciting to read for a broad audience. Prior to publication I have a few requests to the author that they should address in a convincing manner.

We appreciate the Reviewers overall positive view of the manuscript. We also are grateful for the suggestions which we believe have improved the revised manuscript.

1. The manuscript is extremely dense and quite difficult to access for non-expert readers jumping forth and back between different cell lines and reporter combinations. I strongly encourage to authors to streamline the text, especially in the Results section to improve readability for a broader audience (different in terms of scientific background and mother tongue).

We appreciate the Reviewer suggesting where the text could be clearer. We have made edits throughout the Results section to improve readability for a broad audience—for example, making clearer transitions between experiments with different cell lines.

2. Recent studies on mitochondrial import stress used clogger constructs to block the TOM-TIM23 pathway in cells. Quantitative proteomics analyses show that the number of TOM complexes is several-fold higher than the number of TIM23 complexes. Thus, accumulation of a clogger targeted into the presequence pathway likely leads to a strong decrease of available TIM23 complexes, whereas TOM complexes are still available and the membrane potential is still intact. Can the authors either easily test or thoughtfully speculate what will happen to the PINK1-parkin pathway under these conditions that may reflect real pathological situation as well?

This is an interesting suggestion. We are aware of two recently reported import pathway cloggers validated in mammalian cells: ATP5MG-mCherry-sfGFP and IMMT-DHFR. In our revision experiments, we have found that high (but not moderate) levels of the mitochondrial clogger ATP5MG-mCherry-sfGFP substantially blocks activation of Parkin to degrade MFN2-Halo in response to CCCP. Clogging with IMMT-DHFR, by contrast, had a minimal effect of PINK1-YFP stabilization and Parkin activation, consistent with the results reported by the authors. This would suggest that very high levels of a clogger (probably higher than is physiologically relevant) can inhibit Parkin activation, plausibly by blocking the PINK1 binding site in the TOM translocase.

We added the following to the text:

“Recently, it has been demonstrated that import along the TOM-TIM23 precursor pathway can be blocked in mammalian cells using fusion proteins that clog the mitochondrial import pore in TOM^{46,47}. These fusion proteins contain a N-terminus that is directed to the IMM or matrix and a tightly folded C-terminus in the cytosol that cannot be pulled through the TOM complex. This results in clogging of the TOM pore and limits the import of some precursors. We predicted that clogging a high proportion of the TOM pores might prevent PINK1 activation on the OMM by blocking available TOM binding sites for PINK1 import. We first evaluated the recently described mitochondrial import clogger ATP5MG-mCherry-sfGFP⁴⁶. We transiently expressed ATP5MG-mCherry-sfGFP in a MFN2-Halo reporter line that co-expressed TagBFP2-Parkin, as this combination was compatible with the fluorophores in ATP5MG-mCherry-sfGFP. In cells with the highest ATP5MG-mCherry-sfGFP expression, MFN2-Halo degradation by TagBFP2-Parkin was substantially blocked following CCCP treatment (Fig. S5D). Notably, this was not observed in the population of cells with more modest ATP5MG-mCherry-sfGFP expression (Fig. S5D). A second mitochondrial clogger (IMMT-DHFR) was also tested⁴⁷. IMMT-DHFR was found to have a minimal effect on PINK1 stabilization and activation (Fig. S5E), as also reported previously⁴⁷. Together, these results suggest that clogging of the TOM translocase by other stalled precursors can inhibit PINK1-Parkin activation but likely at only high (super-physiologic) levels.”

The data are reproduced here:

3. The only moment where the authors leave solid ground in the interpretation of their data is when they come up with the hypothesis that the energy provided by the MMP must be sufficient to separate PINK1 from its stable interaction with TOM for handover to TIM23. First, I do not fully understand how this would happen to the separated PINK1 in TIM23 knock-down cells. Second, and more important; I consider such semi-quantitative statement always dangerous. Can the authors provide any calculations or modelling approaches that reveal, if such a scenario is realistic from the bioenergetic perspective? Since the electrophoretic force depends on the electrical field and the number of charges in the moved object, will modifying the charge in the N-terminal region of PINK1 alter the MMP threshold for PINK1 activation?

On reflection, we agree with the Reviewer that our model is too specific about the bioenergetics at play given the evidence provided. In the revised manuscript, we now de-emphasize the specific forces at play in the mechanism.

We have modified the statement in the abstract as follows:

“Together, our findings point to a convergent mechanism of PINK1-Parkin activation by mitochondrial damage: loss of MMP stalls PINK1 import during its transfer from TOM to TIM23.”

We have revised the description of our model in the Introduction as follows:

“Together these suggest a model in which PINK1-Parkin activation is primarily sensitive to loss of the MMP. The MMP, in this model, provides the main driving force for PINK1 import to the IMM. Import of endogenous PINK1 to the IMM additionally requires TIMM23 but not the PAM complex. In the absence of the TIM23 translocase, PINK1 still forms a stable interaction with TOM and activates Parkin on the OMM. TOM (including many but all subunits) is required to hold PINK1 to the mitochondrial surface. This work suggests the damage sensing mechanism for the PINK1-Parkin pathway critically involves the MMP, which provide the primary driving force to complete transfer of PINK1 from TOM to TIM23 during import.”

Finally, the relevant portion of the Discussion was modified as follows:

“Together these findings support the following model for damage-sensing in the PINK1-Parkin mitophagy pathway: the PINK1-Parkin pathway is triggered when MMP across the TIM23 translocase is insufficient to complete PINK1 import during the transfer from TOM to TIM23.”

In response to the Reviewer’s first comment about TIMM23 KD. The thinking behind our original model is that both the import path through TIM23 and the driving force provided by the MMP are needed. The reason KD/KO of TIM23 stabilizes PINK1 even in the presence of the MMP is that the import path is lost. I suspect that we agree with the Reviewer, here, and that the confusion is due to our unclear description in our initial submission.

On the Reviewer’s suggestion to explore computational approaches to estimate binding energy between PINK1 and TOM complex, we spoke with a computational structural biologist, Dr. Lucy Forrest, in our Institute. She did not believe that a computational approach would provide a reliable answer for the free energy of binding between PINK1 and the TOM complex. As an example, she explained that calculations they have attempted for much simpler problems like the binding energy between an ion and transporter have yielded inconsistent results. Therefore, we have not attempted formal calculations and have instead revised our model to avoid semi-quantitative claims of the thermodynamics involved.

Along the Reviewer’s suggestion, we also tested PINK1 mutants that lack positive charges in the MTS. Overall, the findings are consistent with the proposed model. Removing 3 – 7 positive charges from MTS led to greater PINK1-YFP levels in the presence of a physiologic MMP and low doses of CCCP. However, the overall charge of the MTS was not the only factor determining the dynamics of PINK1 stabilization in response to MMP lowering. As the interpretation is not straightforward and will require additional work, we have chosen not to include these data here.

4. I am wondering which role PARL cleavage play in the model favoured by the authors. Because PARL1-KO was not amongst the hits that show PINK1 stabilization without treatment, can one assume the PARL cleavage is actually not necessary to release PINK1 into the cytosol from intact mitochondria?

Consistent with the Reviewer's suggestion, PARL was not among the activators of Parkin identified in the MFN2-Halo screen, in contrast to TIMM23. We speculate that although PINK1 (with its MTS cleaved) is stabilized by PARL KD/KO, the accumulated PINK1 does not form an active ubiquitin kinase on the mitochondrial surface, and, thus, does not efficiently activate Parkin. This interpretation would be consistent with prior reports (PMID: 21115803). To test this, we directly compared PARL vs. TIMM23 KD in HeLa cells. Consistent with our interpretation, we found both KDs stabilize endogenous PINK1, but only TIMM23 KD leads to substantial pS65-Ub accumulation.

The new data included in the Revision are reproduced here:

These data are described in the results as follows:

“These results show that blocking PINK1 import to the IMM, through TIMM23 KD, causes PINK1 to accumulate on the surface of mitochondria, where it can activate Parkin to degrade MFN2 and activate mitophagy. PINK1 stabilization on the OMM has also been reported following disruption of PARL, the protease that cleaves PINK1 following its import to the IMM^{33,44}. However, PARL was not identified as PINK1-Parkin activator in our screen. To explore this further, we directly compared the effect of TIMM23 KD to PARL KD on endogenous PINK1 stabilization and activation. Consistent with the flow cytometry and confocal results, TIMM23 KD stabilized and activated full-length PINK1. PARL KD, by contrast, stabilized an MTS-cleaved form of PINK1 that was not active against its substrate ubiquitin (Appendix Fig. S3F). Consistent with prior reports³³, this suggests that PARL KD stabilizes an inactive form of PINK1, in contrast to TIMM23 KD.”

5. Did the authors consider the possibility that a fraction of PINK1 may also be completely imported into mitochondrial matrix, as some studies claim?

We agree with the Reviewer that our data do not rule out the possibility that a fraction of PINK1 may be completely imported into the mitochondrial matrix. For instance, such a model has been suggested to explain the PINK1-dependent phosphorylation of NDUFA10, a complex I subunit. However, the focus of our study has been on events that occur on the surface of the mitochondria, including phosphorylation of ubiquitin and activation of Parkin. The manuscript already covers a fair bit of ground, and so we are reluctant to address this topic experimentally. We now clarify in the Discussion that our work does not directly address a possible role for PINK1 in the matrix, while also providing some of our view of what would be required for PINK1 targeting to the matrix (e.g., that it must escape lateral release into the IMM from the TIM23 translocase):

“While our work resolves several questions regarding how PINK1 import to the IMM is connected to PINK1-Parkin activation, it is worth noting that our work does not directly address whether PINK1 may also be imported to the mitochondrial matrix, as has been reported previously⁶¹. However, there are some implications of our results for PINK1 import to the mitochondrial matrix. First, our results confirm that majority of PINK1 imported into mitochondria under basal conditions is likely released into the IMM^{33,44}. This is supported by the observation that PINK1 is strongly stabilized by blocking its PARL-dependent cleavage in the IMM or by inhibitors of its degradation by the cytosolic proteasome. Matrix targeted PINK1 should be resistant to these blocks in PINK1 processing. Import of PINK1 to the IMM is thought to follow the route of other IMM-resident transmembrane domain containing proteins¹¹. Import proceeds through a groove in the TIM23 translocase (located in the TIMM17A/B subunit) until it reaches the hydrophobic transmembrane region. This sequence stops transfer of the precursor, and the precursor is laterally released by TIM23 into the IMM. For PINK1 to continue import to the mitochondrial matrix, however, it would need to bypass this stop transfer event in the TIM23 translocase. Matrix import of PINK1 would also be predicted to require the action of the PAM motor, as it does for other matrix directed precursors¹¹, as the whole length of PINK1 (561 residues) and not just the positively charged N-terminus would need to pass through the TIM23 channel. Notably, we did not see accumulation of endogenous PINK1 after PAM disruption. Thus, our results do not reveal a substantial pool of PINK1 in the mitochondrial matrix but also do not rule it out entirely.”

Referee #2:

Thayer et al present a comprehensive set of analyses aimed at furthering our understanding of the PINK1-Parkin pathway and in particular the mechanism(s) by which PINK1 senses mitochondrial dysfunction using a combination of complementary approaches. The authors use a novel putative reporter for PINK1-Parkin activity, MFN2-Halo, to perform a number of complementary genome-wide CRISPRi screens to identify regulators of PINK1-Parkin pathway activity at single-cell level. This revealed multiple hits that they describe as facilitators or activators. Some of these are known or anticipated and some are new. Focussing on hits relating to glycolysis, they presented evidence that glycolytic ATP provides the energy for PINK1 synthesis following mitochondrial perturbation. Interestingly, reduced mitochondrial ATP synthesis (a physiologically relevant mitochondrial disruption) sensitised PINK1 stabilisation to reduced MMP. Then focussing on PINK1-Parkin activators by comparing multiple reporters of PINK1-Parkin activity, proteomics, ultrastructure and MMP, they surmised that multiple perturbations that activate the PINK1-Parkin pathway converge on disrupting the MMP. This led the authors to investigate in detail the relative roles of TOM and TIM import channels. Here, it was noted that while loss of TIM23 blocked import on OMM, loss of TOMM22 and TOMM40 led to PINK1-YFP accumulating on an unidentified compartment near lipid droplets. Ultimately, the results showed that TOM but not TIM23 was required for OMM stabilisation. Further targeted experiments supported that TOMM5 and TOMM20 promote stabilisation, and disruption of the PAM complex likely stabilises PINK1 by affecting MMP.

Overall, the authors present a coherent and thoroughly performed study, originating from unbiased screens using a novel reporter approach, combined with multiple orthogonal assays to analysed in detail the mechanisms on which various screen hits converge. Consistently, these seem to point to loss of membrane potential and the stabilisation of PINK1 in the TOM channel, which is entirely in line with the

predominant view in the field, albeit based on studies using potent mitochondrial poisons and depolarising agents. In this regard, there is little of surprise or controversy here and the observations could be at risk of confirmation bias since impacts on MMP are the most obvious mechanism to investigate. Nevertheless, the experiments are well executed and nicely presented in a cogent narrative. As such, this study is a valuable addition to the field and provides additional insight into the molecular mechanisms of an important quality control process. As ever it still remains to be seen what is happening in vivo and in particular in ageing neurons relevant to Parkinson's disease, but for the present study I have only relatively minor comments for consideration.

We appreciate the Reviewers overall positive view of the manuscript. We also are grateful for the suggestions which we believe have improved the revised manuscript.

Specific points:

1. Just for clarity, if the genome-wide screens were performed in duplicate, what is represented in the Supplementary Tables exactly? It looks like single values are presented for each screening condition. Is this the mean of the duplicates?

What was represented were the log₂ fold-changes (LFC) and -log₁₀ p-values for the comparison of guide counts in the top 30% gate to guide counts in the bottom 30% following the default normalization in the output from the MAGEck analysis. In the revised Dataset 1 we now include the guide counts for the two duplicates for each screen as additional columns. The raw sequencing data has also been uploaded to GEO.

2. Fig. 3F-G. What is shown exactly? what does a single data point represent - measurement of a single mitochondrion or from a single cell or what? How is mean and variance determined?

Please make it clear how the statistics are applied to this and other similar data sets - are they applied to individual cells or independent transductions? Really, it should be the latter.

In these graphs, each data point represents the low or high MMP area measured from a single cell. These measurements were made using a custom script in FIJI that calculates average PINK1-YFP intensity in masks generated from the MitoLite NIR and MTS-mCherry channels. We find the segmentation is not perfect but represents a good balance between speed and accuracy. In the statistical analysis, a paired-test was used (Wilcoxon matched-pairs signed rank test, as the data is not normally distributed) to compare the two mitochondrial populations (MMP high and low) within each cell. We believe this is most intuitive way to represent these data, as it maintains the link between mitochondrial populations in single cells throughout the analysis. Applying the statistics to averages across cells from separate wells breaks this link. In this case, we would no longer be comparing mitochondria within single cells.

However, we understand the reviewers concern that we demonstrate clearly our findings are robust and reproducible from experiment to experiment. To address this concern, we have represented these data a second way in the supplemental figure along the Reviewer's suggestion: we compare the population averages for each well of cells that was imaged (total of 4 – 6 wells per sample imaged in two separate transductions). We do this also for measurements of the Pearson R coefficient over background.

The data separated by replicate are reproduced below. (Of note KD of two additional genes were tested in this manner, LONP1 and HSPD1. These expand on our observation that disrupted proteostasis activates PINK1 by lowering MMP and are responsive to the suggestion by Reviewer 3 to explore this further.)

3. Fig. 4A-C. I guess that the open- vs closed-circle data points represent the two independent transductions as mentioned in the legends. This is a welcome touch, and it would be nice to make this clear in the legend also.

Yes, these represent two independent transductions. We have revised the figure legend as suggested.

4. Fig. 5D. The CLEM here is very nice but the confocal image in D is not very clear and doesn't look like that shown in C. Can this be improved?

The relative lack of clarity for the CLEM in 5D is due to two technical factors involved in the preparation of the sample. The sample was imaged on Aclar rather than coverglass to aid separation of the Epon embedded sample from the support after confocal imaging. Additionally, the Aclar was carbon coated to add a finder grid, which we later found added some autofluorescence in the YFP channel. These two factors limited the resolution and signal to noise of the light microscopy. In response to the Reviewer's comment, we revisited this sample to see if higher quality images could be obtained. Ultimately, however, we concluded that the selected image was the highest quality from the sample. As we believe the correlation is sufficiently clear between the PINK1-YFP and the TEM image to illustrate how the ultrastructure of the general region of PINK1-YFP accumulation is changed, and these experiments are rather time-consuming and involved, we have decided to leave the original correlated image in the figure. While we agree that the aesthetics could be improved here with additional investment, we do not think this would meaningfully strengthen our core findings.

It is the combination of observations in C and D that allow the conclusion that PINK1 accumulates near lipid droplets. Considering this, where do the authors think the PINK1 is exactly? On the vesicles? cytosolic? How does this compare with S5A where there are very few vesicular structures. Since diffraction in the Z axis can confound accurate colocalization, a series of TEM sections would be interesting to see what is above/below the LDs. It would be fantastic to see the localization of PINK1-YFP by immuno-gold EM but this may be beyond the desires of the authors at this stage.

This is an excellent question. In response to the Reviewer, we performed several additional experiments to better understand where PINK1-YFP is localized following disruption of the TOM translocase. Our conclusion is that a pool of PINK1-YFP is directly associated with the lipid droplets. This is based on (1) Airyscan super-resolution light microscopy of PINK1-YFP with the neutral lipid stain LipidTOX, and (2) the experiment suggested by the Reviewer to examine PINK1-YFP localization by immuno-gold EM. We additionally identified that a region of PINK1-YFP previously mapped as an outer mitochondrial membrane localization signal is required for PINK1-YFP association with lipid droplets.

The data are reproduced here:

Figure x

The following has been added to the results text to describe this experiment:

“To more precisely determine where PINK1-YFP accumulates in the absence of TOM, we examined PINK1-YFP localization by super-resolution confocal microscopy and immuno-gold EM. These methods showed that a large portion of PINK1-YFP accumulates directly on lipid droplets: PINK1-YFP formed a halo around lipid droplets stained with LipidTox by superresolution microscopy and immunogold decorated the surface of lipid droplets by TEM (Appendix Fig. S4A and B). To determine whether a particular region of PINK1 is required for lipid droplet localization, we tested a deletion series of PINK1-YFP for lipid droplet localization by confocal microscopy. This showed that lipid droplet localization depended on a helical region (residues 74 - 93) that was previously identified as required for PINK1 OMM localization⁴⁵ (Appendix Fig. S4C). These results suggest that while disruption of either TIM23 or TOM translocases lead to increased cellular PINK1-YFP, the localization is different in each case. PINK1-YFP accumulates on the mitochondria following disruption of TIM23. By contrast, PINK1-YFP accumulates largely in the cytosol and on lipid droplets following disruption of TOM.”

For the Revision we also performed CLEM on the TIMM23 KD as a comparison to the TOMM22 and TOMM40 KD CLEM. We now include these data as Fig. EV3:

Figure EV3

Does this signal co-localise with TOMM70 or cytosolic HSP70s, given the results in Fig. 7.

This is a good point. We observe minimal colocalization between TOMM70 and PINK1-YFP and so conclude that, although a small population of PINK1-YFP is associated with TOMM70 and HSP70s, as revealed by the AP-MS experiment, a large pool of PINK1-YFP is not associated with the TOMM70 receptor.

To make this point clear we have modified the relevant portion of the results text as follows (new sentence underlined):

“Together, these data suggest that TOMM70 may bind some excess unfolded PINK1 that cannot be incorporated into the TOM translocase or degraded in the cytosol. It is likely that most PINK1-YFP is not directly bound to TOMM70, however, as PINK1-YFP does not strongly co-localize with TOMM70-positive mitochondria (Fig. 5C).”

Does this localisation occur with endogenous PINK1, or is this an artefact of exogenous PINK1-YFP?

Our interpretation is that this is an artifact of exogenous PINK1-YFP, as we do not see endogenous PINK1 accumulation following knockdown of TOMM22 or TOMM40, as shown in Fig. 5H of the original and revised manuscript. The YFP tag may block a C-terminal degradation signal or the higher PINK1 expression may overwhelm the ubiquitin proteasome system. This panel and our interpretation is described in the Results text as follows:

“We next examined the effect of TIM23 vs. TOM disruption on endogenous PINK1. Immunoblotting for endogenous PINK1 showed a similar pattern as exogenous PINK-YFP, except that endogenous full length PINK1 did not accumulate following TOM disruption (Fig. 5H, right blot). We hypothesize that the YFP tag may block degradation of full length PINK1, allowing PINK1-YFP but not endogenous PINK1 to accumulate following loss of TOM. As above, PINK1 activity was increased by TIM23 but not TOM disruption. Together these results demonstrate that loss of TIM23 but not TOM causes PINK1 activation.”

5. Also, in 5C, the TOMM22 sgRNA images; the PINK1-YFP zoom images in the separate vs merged channels looks subtly different. Different plane? Please check.

We thank the Reviewer for pointing out this error. The separated image was a maximum projection of three planes. We mistakenly included only one plane for the merge. This has been corrected:

6. With the differing effects of TOMM22/40 vs TIM23 KD, I am curious to know what happens with a double KD, but this may perhaps have limited interpretation.

This is an interesting suggestion. We found that KD of TIMM23 on top of KO of TOMM40, phenocopies TOMM40 KO, as expected. These data are described in the results as follows:

“Notably, disruption of both TIMM23 (by KD) and TOMM40 (by KO) phenocopied TOMM40 KO (Appendix Fig. S5B).”

The relevant panels are reproduced here:

7. MitoLite NIR is a new reagent to me but looks like a useful tool for the field and an alternative to traditional dyes such as TMRE. Since there are very few technical details on the company website, it would be helpful if the authors (recognised experts in the field) could perform a few simple tests to demonstrate the functionality of this reagent compared to other well-known ones, for instance, showing that the dye does not localise to chemically depolarised mitochondria and, importantly, whether its localisation is reversible, i.e., pre-loading energised mitochondria with the dye, it redistributes upon depolarisation.

MitoLite NIR was previously used by the Sesaki lab in (PMID: 33200421), which inspired our use here. We now perform an additional experiment to confirm that is dependent on the MMP in Appendix Figure SX:

We added the following to the text:

“For these experiments, we used the MMP dye MitoLite NIR, which requires a MMP for uptake similar to TMRE but emits in the far-red region³⁷ (Appendix Fig. S3A).”

In these experiments we did note that MitoLite NIR takes longer than TMRE to diffuse away from mitochondria following loss of MMP. However, its use in the present study is to identify mitochondria without MMP at the time of dye addition. We found the two dyes work similarly for this purpose.

8. In general, colour-coding of different conditions is nice and generally helpful. However, the authors may want to change their colour schemes when the coding changes, e.g., in Fig. 4. A-D vs G.

We recognize the concern but also wanted to limit the color palette used to prevent the colors from becoming too distracting. The colors in 4F correspond to those in 4G and indicate which quadrant is being measured. We have placed a box around 4F and G in the Revised figure to help indicate that the colors are being recoded for these two panels.

9. If space allows, the authors could expand a bit on the abstract to make it more accessible to a broader audience.

We have re-written the abstract to give more context and hopefully increase accessibility within the space allowed, as suggested by the Reviewer. It now reads:

“Damaged mitochondria can be cleared from the cell by mitophagy, using a pathway formed by the recessive Parkinson’s disease genes PINK1 and Parkin. Whether the pathway senses diverse forms of mitochondrial damage by a common mechanism, however, remains uncertain. Here, using a novel Parkin reporter in genome-wide screens, we identified that diverse forms of mitochondrial damage converge on loss of mitochondrial membrane potential (MMP) to activate PINK1. Loss of MMP, but not the PAM import motor, blocked progression of PINK1 import through the translocase of the inner membrane (TIM23), causing it to remain bound to the translocase of the outer membrane (TOM). Ablation of TIM23 was sufficient to arrest PINK1 in TOM, irrespective of MMP. Meanwhile, TOM (including subunit TOMM5) was required for PINK1 retention on the mitochondrial surface. The energy-state outside of the mitochondria further modulated the pathway by controlling the rate of new PINK1 synthesis. Together, our findings point to a convergent mechanism of PINK1-Parkin activation by mitochondrial damage: loss of MMP stalls PINK1 import during its transfer from TOM to TIM23.”

10. Finally, the authors may want to comment on their findings in relation to the recent paper from Komander’s group on the structure of PINK1 with TOM/VDAC complexes, particularly with respect to the interaction with TOMM5/7.

We thank the Reviewer for this suggestion. The following has been added to the Discussion:

"While our manuscript was under review a structure of human PINK1 in complex with the TOM translocase was reported⁶⁰. The structural details agree with our functional data in several key respects. First, they confirm extensive contacts between PINK1 and TOMM20 that participate in the stabilization of PINK1 on the OMM. Second, they establish that both direct and indirect contacts form between TOMM5 and PINK1. These include TOMM5 positioning the N-terminal segment of TOMM40 and VDAC2 relative to PINK1. These contacts provide a structural basis for our observation that TOMM5 is required along with TOMM7 for PINK1 stabilization and activation. Third, the structure lacks TOMM70, consistent with our finding that TOMM70 is dispensable for PINK1 activation in cells. Together, these structural findings are highly complementary with the functional data reported here.

Corrections:

- Fig 4C. "PGRAM5" is a misspelling

Thank you for catching this typo. It has been corrected.

- I assume "This identifies maturation of newly important proteins..." in the sentence 4 lines up from the bottom should read 'newly imported proteins'.

Thank you for catching this typo. It has been corrected.

Referee #3:

Thayer et al report a novel reporter of Parkin-PINK1 pathway, based on MFN2 tagged with Halo. They use that reporter to perform a genome-wide CRISPR screen. While several such screen were reported previously, this is the first that looks at a direct substrate of Parkin, and the study is completed by very elegant biochemistry and proteomics. While most of the results are largely confirmatory, the paper brings about a substantial resource for the field and helps clear up several aspects of the mechanism of PINK1 activation. One of the most significant aspects is the consolidation of the idea that loss of MMP is the main driver of PINK1 accumulation, and that knockdown of several genes can cause heterogenous loss of MMP, observed by microscopy or single-cell level, which is why they cause bulk accumulation of PINK1. It is rare that I write this, but I would almost accept the paper as is, except for the minor revisions described below.

We appreciate the Reviewers overall positive view of the manuscript. We also are grateful for the suggestions which we believe have improved the revised manuscript.

1) The requirement for glycolytic ATP has been reported previously by Lee et al (2015, JBC) and McLelland et al (2018, Elife), which should be cited. Lee et al indeed studied in depth the role of ATP generation and inhibition of glycolysis on PINK1 accumulation. Likewise, the impact of galactose on Parkin recruitment was also previously observed by McLelland et al. Please correct accordingly.

We thank the Reviewer for pointing out this oversight. We have added these citations in the Results and credit this previous work in several places where the impact of galactose on PINK1-Parkin activation is discussed and where the role of ATP on PINK1 generation is discussed.

2) I don't understand this sentence on page 11: "Similarly, PINK1-YFP correlated with MTS-mSc but poorly with MMP (Fig. 3G)." Shouldn't it be the opposite, based on the previous sentence and Fig. 3F? In fact PINK1-YFP appears like it is anti-correlated with MMP. Something is weird in the way correlation is reported in Fig. 3G (as well as 4I, 4J, left panels Robs-Rrand)... in the case of depolarization-induced

PINK1 accumulation, MMP and PINK1 levels should be anti-correlated, as shown by the higher levels of PINK1 in MMP low conditions (Fig. 3F and 4I, right panels). The authors should perhaps use a signed correlation coefficient to analyze the data, and then compare with the random distribution, or find some way of correlating PINK1 levels with the loss of MMP.

In these analyses we are correlating the intensity of PINK1-YFP and either MitoLite NIR or Mito-mCherry for all pixels within the cell region of interest (ROI). Even mitochondria with a high MMP have some PINK1-YFP signal. As a result, there is always a positive correlation between these two mitochondrial signals when measured in the whole cell ROI.

We also estimate the correlation between the two signals that occurs by chance, by measuring the correlation after the channels have been randomly transposed relative to each other (averaged across 20 iterations in FIJI). This is done using the default settings of the Colocalization Test in FIJI. We subtract the Pearson coefficient from the average of random these transpositions (which represents the background correlation for the two signals) from the measured Pearson coefficient in the original image to get a correlation above the background that occurs for the two signals by chance. We have rephrased this in our figure axis label as “R over background.”

To directly address the Reviewers concern, we have also repeated this analysis within a mitochondrial (rather than whole cell ROI) ROI defined by the Mito-mCherry signal. As expected, this demonstrated a negative correlation between PINK1-YFP and MitoLite NIR. These data now appear in Appendix Figure S3B:

3) I would qualify this statement on page 16: "Together this establishes that TOM but not TIM23 is required for PINK1 stabilization and activation on the OMM". Here, the authors show that TIM23 knockdown causes PINK1 activation, but that does not mean TIM23 may not be required for PINK1 stabilization induced by loss of MMP. In fact, Akabane et al (2023) found that knockdown of TIM23 indeed reduced PINK1 activation under conditions of depolarization, and Eldeeb et al (2024) obtained similar results with CRISPRi against TIM50. Please modify the conclusion accordingly and discuss.

We have reworded the conclusion highlighted by the Reviewer to be more specific to our findings, as follows:

"Together these results demonstrate that loss of TIM23 but not TOM causes PINK1 activation."

We extend our discussion of this issue in the Discussion as follows:

“This model helps clarify several open questions in the field. Mitochondrial uncoupling with CCCP was shown to be sufficient to activate the PINK1-Parkin pathway over a decade ago^{12,26}, leading us and others to propose that activation of the PINK1-Parkin pathway is due to import block of PINK1 along the precursor pathway⁵⁴. This model, however, was seemingly at odds with recent studies, which found TIMM23 knockdown by transient siRNA transfection (for 2 – 3 days) or small molecule inhibitors does not phenocopy PINK1 stabilization by OXPHOS inhibition^{4,55,56}. In fact, acute KD of TIMM23 was found to inhibit PINK1-Parkin activation by OXPHOS inhibition. In this condition, OMA1 was shown to degrade PINK1 that was no longer shielded by the TIM23 complex, thereby suppressing the PINK1-Parkin pathway. Using methods that allowed for more sustained TIMM23 knockdown or knockout in single cells (assayed at 7 - 8 days after transduction or electroporation), we found that TIMM23 depletion is sufficient to stabilize and activate PINK1. As OMA1 levels are likely depleted by sustained import block, under these conditions, they may no longer be limiting the accumulation of PINK1 in the TOM complex. This clarifies that import block at TIM23 translocase is a key step in PINK1 stabilization on the OMM and reinforces the import block model of PINK1-Parkin activation.”

4) The authors claim to be surprised by the effect of TOM5 as a facilitator, but a previous study already reported TOMM5 as a facilitator using a genome-wide screen for mitophagy (Hosino et al 2019, Nature, Fig. 1c), although granted they didn't discuss it in the text. This group also mostly studied the role of the nucleotide exchanger ANT1 in mitophagy, which should be discussed in the context of glycolytic vs mitochondrial ATP generation.

We apologize for this oversight, which was not intentional. We now mention that TOMM5 was previously identified as a facilitator of mitophagy. We think it is important to note that TOMM5 is not mentioned in the manuscript and only appears in as a label in a scatterplot in the Figure. Further characterization of the role of TOMM5 in the PINK1-Parkin pathway was not reported by the authors.

We have reviewed the Results text as follows:

“TOMM5 was previously identified as an accelerator of PINK1-Parkin mitophagy, as part of a screen, but how TOMM5 modulates the PINK1-Parkin pathway was not further explored⁵⁵.”

This now appears in the Discussion:

“TOMM5 also previously identified as an accelerator of PINK1-Parkin mitophagy in a screen but was not characterized in detail⁵⁵.”

Along the Reviewer's suggestion, we have also added mention of ANT1/2 to the Discussion in the context of describing factors that may alter the response to import efficacy to the MMP. We understand the authors to suggest that ANT1/2 regulates TIM23 import through binding TIMM44 independently of ADP/ATP exchange (and so independently of the effects of ANT on ATP production). Therefore, we thought it fit better here than in discussion of how bioenergetics regulate the PINK1-Parkin pathway.

This now appears in the Discussion:

“The driving force required for PINK1 import through TIM23 may also be tuned through its binding to other proteins in the IMM, such as TIMM44 and ANT1/2⁵⁵.”

Furthermore, given the recent publication of the structure of the PINK1-TOM complex by the Komander lab (Callegari et al Science 2025), the authors may want to revisit their interpretation of the TOMs knockout results.

We now discuss our findings in light of the recent structure from the Komander lab, also in response to the suggestion from Referee #2. The relevant paragraph added to the Discussion is reproduced above.

5) While the authors clearly demonstrate that PINK1 accumulation induced by PAM disruption is mediated via loss of MMP. However, they do not address the model whereby expression of delta-OTC induces PINK1 despite no loss of MMP accumulation (Jin & Youle 2013, Burman et al 2017). They should at least discuss this result in the context of their own.

As the Reviewer suggests, we conclude that disruption of MMP and not PAM is the likely reason for PINK1 stabilization in response to protein misfolding in the mitochondrial matrix. To strengthen this claim in the Revision, we examined the effects of Δ OTC overexpression and perturbation of matrix chaperones/proteases (LONP1, CLPP, CLPX, HSPD1, HSPE1, and TRAP1) on PINK1-YFP stabilization. We felt it was important to also test these factors maintaining the proteostasis of endogenous mitochondrial matrix proteins as this seems more physiologically relevant than overexpression of Δ OTC.

Consistent with our prior findings, PINK1-YFP increase correlated with MMP lowering at the single cell level following KD of matrix chaperones. We quantitatively investigated LONP1 KD and HSPD1 KD also at the single mitochondrion level by confocal microscopy, as these two knockdowns generated very heterogenous populations of mitochondria with respect to MMP within single cells. Similar to our prior observations, PINK1-YFP is stabilized preferentially on mitochondria with low MMP, following these perturbations. These data are reproduced below:

As aggregates of insoluble protein are known to accumulate in the mitochondrial matrix following KD of LONP1, we also performed CLEM to assess the relationship between visible aggregate formation, PINK1-YFP accumulation, and MMP. PINK1-YFP was stabilized on mitochondria with visible matrix aggregates that lost MMP but not those with retained MMP. These data are reproduced below:

To examine the effect of overexpressing the aggregation-prone protein Δ OTC, we first attempted to introduce the cassette for DOX induction of Δ OTC used in the prior publications into our cell line expressing PINK1-YFP. All clones grew slowly, and none showed the typical expression of PINK1-YFP and response to CCCP in the absence of DOX. This may be due to leakiness and toxicity of Δ OTC from this cassette.

As an alternative, we transiently transfected PINK1-YFP cells with Δ OTC. In these experiments, we observed that the PINK1-YFP high population had lower MMP but the effect was more modest than what observed following knockdown of proteases/chaperones maintaining mitochondrial matrix proteostasis. One possible reason for this difference between Δ OTC overexpression vs. KD of matrix chaperone and proteases is that Δ OTC may be prone to aggregation throughout the import path and not only on arrival in the matrix. If Δ OTC were to block TIM23 this may phenocopy TIMM23 knockdown and stabilize PINK1-YFP in the absence (or with more minimal) MMP loss.

These data are reproduced below:

The following text was added to the Results to describe these findings:

“If matrix protein misfolding activates PINK1-YFP through MMP loss, this correlation should be seen following disruption of other matrix chaperones and proteases needed to maintain matrix proteostasis. To test this prediction, we systematically disrupted other proteins that are essential for maintaining proteostasis in the mitochondrial matrix. These included the quality control protease LONP1, subunits of the mtHSP60 chaperonin (HSPD1 and HSPE1), the mtHSP90 chaperone TRAP1, and subunits of the ATP-dependent protease CLPP (CLPX and CLPP). These factors are important for the folding and degradation of endogenous mitochondrial matrix proteins after they complete import. We first assessed knockdown of each on PINK1-YFP and MMP intensity by flow cytometry. Similar to DNAJA3 KD and the other activators assays above (apart from TIMM23), cells fell into two populations with either high MMP/low PINK1 or low MMP/high PINK1 (Fig. EV5B and C), indicating a strong correlation between MMP loss and PINK1-YFP accumulation. To test whether disruption of mitochondrial proteostasis had similar effects on endogenous PINK1 activation, we next assessed the same gene perturbations by immunoblotting (Fig. EV5D). The perturbations that most strongly increased PINK1-YFP levels by flow cytometry also most strongly stabilized and activated endogenous PINK1. The extent of import block, measured by the accumulation of the ATP5A precursor, also correlated with extent of PINK1 activation. Together this suggests that disruption of mitochondrial proteostasis in the matrix activates PINK1 by the same mechanism as observed with the other mitochondrial activators. MMP loss leads to import block of PINK1, causing its stabilization and activation on the mitochondrial surface.

“We next examined select knockdowns by live confocal microscopy. Perturbation of either LONP1 or HSPD1 resulted in single cells containing a heterogeneous population of mitochondria with respect to MMP, similar to disruption of NDUFB1, PMPCB, and DNAJA3 above. As with the other perturbations, PINK1-YFP accumulated preferentially on mitochondria with low MMP in single cells following either LONP1 or HSPD1 KD (Fig. 6I - J and EV5E - G). For LONP1 KD, some cells were fixed immediately after measuring MMP live and processed for CLEM. By TEM mitochondrial aggregates could be directly observed both in mitochondria that retained MMP and mitochondria that had lost MMP (Fig. 6I). PINK1-YFP accumulated preferentially on aggregate-containing mitochondria that had lost MMP. This suggests that aggregate formation is not the direct stimulus for PINK1 accumulation; instead, PINK1 accumulates only after MMP is lost. Together these results demonstrate that disruption of mitochondrial matrix proteostasis activates PINK1 by disruption of the MMP.

“Finally, we tested a mutant form of ornithine carbamoyltransferase (Δ OTC), previously shown to activate the PINK1-Parkin pathway in a MMP independent manner^{50,51}. Again, two populations were seen, but the population with increased PINK1-YFP had an MMP that was only mildly reduced (Fig. EV5H). Δ OTC stress may differ from matrix protein misfolding following loss of proteostasis, as Δ OTC must pass through TIM23 as an aggregation-prone protein. This may disrupt the TIM23 import path (analogous to TIMM23 KD) in addition to lowering MMP, although this will require further investigation to confirm. The Δ OTC system is somewhat artificial, however, and is less likely to reflect physiologic protein misfolding, which may be better modelled by disrupting factors critical for the proteostasis of endogenous mitochondrial proteins.”

Dear Derek,

We have now received re-review reports from three referees, which I have included below. As you will see, you have addressed their concerns satisfactorily. Before I can finally accept the manuscript, there are some remaining editorial points which need to be addressed. In this regard would you please:

- acknowledge the following funding in our online submission system: the Intramural Research Program (ZIA NS003169) of the National Institutes of Health (NIH), National Institute of Neurological Disorders and Stroke (NINDS); and acknowledge the following funding in the main manuscript: the National Institutes of Health (NIH) (1ZIAN003169),
- list references alphabetically, using 'et al.' for author lists with more than ten names,
- rename the conflict of interests statement as the "Disclosure and competing interests statement",
- remove the AC/CrediT section from the text,
- rename EV figure legends as Figure EV1-EV5 instead of Expanded View Figure 1-5,
- update source file titles and legends to Dataset EV1-EV#, legends should be uploaded as a separate tab/sheet in each Excel file,
- provide (where appropriate) exact p values in the legends of figures 1C, E; 2B, 3F, G; 4I, J; 5B, G; 6D, F, H, J; 7A, E; EV4 C, D, E, F; EV5 B, G; S1 D, E, F, I; S3A, B, C, E
- indicate the statistical test used for data analysis in the legends of figures 3H, 5I, 6G and 7G,
- define error bars in the legend of figure S6, and
- adjust the section order as follows: - Sections need to be named and the order should be corrected: Title page - Abstract - Keywords - Introduction - Results - Discussion - Methods - Data Availability - Acknowledgements - Disclosure and Competing Interests Statement - References - Figure Legends - Table(s) - Expanded View Figure Legends.

We include a synopsis of the paper (see <http://emboj.embopress.org/>). Please provide me with a general summary image, a two sentence statement and 3-5 bullet points that capture the key findings of the paper.

I am looking forward to receiving your revised manuscript.

EMBO Press is an editorially independent publishing platform for the development of EMBO scientific publications.

Best wishes,

William

William Teale, PhD
Editor
The EMBO Journal
w.teale@embojournal.org

- a Reagents and Tools Table as part of the Methods section, which can be downloaded from our author guidelines

(<https://www.embopress.org/page/journal/14602075/authorguide#structuredmethods>)

We realize that it is difficult to revise to a specific deadline. In the interest of protecting the conceptual advance provided by the work, we recommend a revision within 3 months (7th Dec 2025). Please discuss the revision progress ahead of this time with the editor if you require more time to complete the revisions. Use the link below to submit your revision:

Referee #1:

The authors have addressed all points raised by the reviewers in an exemplary manner. I would like to congratulate the authors with this outstanding paper!

Referee #2:

The authors have responded to my original comments very well, thoroughly addressing each point and in most places with the addition of new data. My only remaining point for consideration relates to my previous point regarding clearly defining what the data shown in charts relates to in terms of experimental sample breakdown (i.e., mitochondrion/cell/well/transduction etc). The authors clarified my query regarding Fig. 3F-G and added new data representation in Appendix Fig. S3, but the legend to S3 did not clearly describe how these data were derived.

Also, since EMBO J posts reviews and responses, the response to reviewers document contained a few errors in citing specific figures/panel which the authors may want to revise for final posting.

Finally, congratulations to the authors on an excellent study.

Referee #3:

The authors have addressed the minor issues I raised beyond my expectations. The added data on chaperone knockdown (point #5) brings significant additional impact to this study by unambiguously demonstrating that protein misfolding per se does not induce PINK1 accumulation, but mediate its effect via loss of MMP. This paper should be accepted as it is.

All editorial and formatting issues were resolved by the authors.

Dear Derek,

I am pleased to inform you that your manuscript has been accepted for publication in the EMBO Journal.

Congratulations to you and your team!

Best wishes,

William

William Teale, PhD
Editor
The EMBO Journal
w.teale@embojournal.org
